# Subgraph Federated Learning for Local Generalization

**Sungwon Kim**[1,✉]**, Yoonho Lee**[1,✉]**, Yunhak Oh**[1,✉]**, Namkyeong Lee**[1,✉]**,
**Sukwon Yun**[2,✉]**, Junseok Lee**[1,✉]**, Sein Kim**[1,✉]**, Carl Yang**[3,✉] **& Chanyoung Park**[1,✉*]
[1]KAIST, [2]UNC Chapel Hill, [3]Emory University

## Abstract

Federated Learning (FL) on graphs enables collaborative model training to enhance performance without compromising the privacy of each client. However, existing methods often overlook the mutable nature of graph data, which frequently introduces new nodes and leads to shifts in label distribution. Since they focus solely on performing well on each client's local data, they are prone to overfitting to their local distributions (i.e., local overfitting), which hinders their ability to generalize to unseen data with diverse label distributions. In contrast, our proposed method, Fed-LoG, effectively tackles this issue by mitigating local overfitting. Our model generates global synthetic data by condensing the reliable information from each class representation and its structural information across clients. Using these synthetic data as a training set, we alleviate the local overfitting problem by adaptively generalizing the absent knowledge within each local dataset. This enhances the generalization capabilities of local models, enabling them to handle unseen data effectively. Our model outperforms baselines in our proposed experimental settings, which are designed to measure generalization power to unseen data in practical scenarios. Our code is available at `https://github.com/sung-won-kim/FedLoG`

## 1 Introduction

In the realm of Graph Neural Networks (GNNs) (Hamilton, 2020), most systems are designed for a unified, centralized graph. However, real-world applications (Zhang et al., 2021a) frequently involve individual users or institutions maintaining private graphs, isolated due to privacy concerns. Graph Federated Learning (GFL) (Liu et al., 2024) provides a solution by enabling clients to independently train local GNNs on their data. This decentralized training approach allows a central server to aggregate the locally updated weights from multiple clients, creating a unified model that respects privacy constraints. In this paper, among the various settings in GFL, we focus on one of the most challenging aspects—distributed subgraphs (subgraph-FL), where clients manage largely disjoint sets of nodes and their edges.

In real-world scenarios, graph data frequently changes, particularly in social, citation, and e-commerce networks (Sen et al., 2008; McAuley et al., 2015; Shchur et al., 2018). While these changes often result in new label distribution patterns that are distinct from the existing local label distribution within each client, existing subgraph-FL methods (Zhang et al., 2021a; Wu et al., 2021a; Yao et al., 2024; Baek et al., 2023) primarily focus on optimizing models based on the current label distribution within each client (i.e., local optimization). On the other hand, some studies (Zhang et al., 2022; Lee et al., 2022; Guo et al., 2024) demonstrate that client models are particularly prone to local overfitting after local updates, resulting in a significant decrease in the accuracy of minority classes (i.e., tail classes) within the local data. Given these limitations, current approaches face substantial practical challenges, particularly in adapting to new nodes added to the original local graph. This is especially difficult for nodes belonging to *tail classes or unseen classes that are missing from the local graph but exist in other graphs (i.e., missing classes)*. These nodes, which form new connections with existing nodes, often have structural patterns unfamiliar to local clients, leading to substantial discrepancies in both label and structural distributions of the local graphs.

Existing methods in FL (Li & Zhan, 2021; Zhang et al., 2022; Lee et al., 2022; Guo et al., 2024) aim to ensure that local models can make predictions for all classes without bias by mitigating local

---

*Corresponding author (`cy.park@kaist.ac.kr`)

overfitting caused by the local label distribution. Specifically, they propose regularizing the logits of each class in the local models to align more closely with those of the global model. While these methods effectively address local overfitting and manage tail or missing classes, increasing the logits of local tail data risks amplifying noisy data, which is harmful for the class representation of the global model. Beyond FL, another common approach to mitigating the problem of overfitting on training data involves addressing class imbalance. Techniques such as down-sampling (He & Ma, 2013), over-sampling (Chawla et al., 2002; Zhao et al., 2021; He & Bai, 2008), calibration (Niculescu-Mizil & Caruana, 2005; Zadrozny & Elkan, 2001), or constructing expert models for tail data (Menon et al., 2020; Yun et al., 2022) are commonly used. Despite their effectiveness, they require at least one data point to be present for each class, facing challenges when a class is missing in a local client while present in others.

In this paper, we propose to address the local overfitting issue of subgraph-FL by introducing reliable global synthetic data that mitigate class imbalance while addressing missing classes. Specifically, we aggregate knowledge from local data across all clients for each class, and integrate it into the global synthetic data. Subsequently, each client adaptively utilizes the global synthetic data as additional training data to ensure effective learning for all classes, including those that are underrepresented or missing in each client. This strategy helps prevent local overfitting even after local updates and enables accurate class representation (i.e., local generalization). However, there exist two crucial challenges that need consideration:

**C1. Which data across all clients should be aggregated to ensure reliability?** Since clients in FL heavily rely on knowledge from other clients to learn locally absent information, it is crucial for each client to share the most reliable knowledge within its own local graph. Here, data reliability refers to the accuracy and consistency of information sourced from decentralized nodes.

**C2. How can data from other clients be utilized without compromising privacy?** While direct sharing of the data between clients prevents local overfitting, it raises severe privacy concerns. Furthermore, directly using training data from all clients incurs high communication costs.

**Solution to C1. Knowledge from head degree and head class nodes.** Our findings indicate that nodes with a high number of connections (i.e., head degree) and those belonging to the majority class (i.e., head class) provide reliable structural and class-representative information, respectively, which significantly enhances the model's ability to generalize to unseen data. Building on these insights, we aggregate knowledge from clients and filter it based on the "headness" of both their degree and class.

**Solution to C2. Data condensation.** We propose to condense only reliable knowledge into synthetic data to share across the clients. This avoids the direct use of individual client data, mitigating the privacy concerns, while also minimizing the amount of data transferred between the server and local clients, thereby lowering communication costs.

In summary, we propose a subgraph **Fed**erated Learning framework for **Lo**cal **G**eneralization, FedLoG, that generates global synthetic data with a novel reliable knowledge condensation strategy. This approach reduces the risk of noise in class representations, and enables each client to compensate for locally absent knowledge without compromising privacy. By doing so, FedLoG prevents local overfitting and ensures a well-generalized representation of all classes, enabling successful handling of unseen data in our three proposed evaluation settings; **1) Unseen Node:** New nodes with seen classes are added, and they introduce structural changes in local graphs. **2) Missing Class:** New nodes with classes previously absent in the client's graph are added. **3) New Client:** A completely new graph with distinct label distribution and structure is added.

In this paper, we make the following contributions:

- We introduce FedLoG, the first work in subgraph-FL that focuses on preventing local overfitting, including the issue of missing classes, to address the mutable nature of the graph domain. This approach enhances the performance of the global model and improves the generalization power of local models, allowing them to effectively handle unseen data.

- We analyze what constitutes reliable data in graph-based federated learning and propose a method to condense and share this knowledge across clients. This approach not only leverages reliable data effectively but also protects privacy by using condensed synthetic data.

- We propose practical and important evaluation settings on unseen data for subgraph-FL (i.e., Unseen Node, Missing Class, and New Client), enabling measurement of the model's generalization on future data, assessing robustness in mutable graph domains, and demonstrating consistent outperformance over other baselines.

## 2 RELATED WORK

### 2.1 SUBGRAPH FEDERATED LEARNING

Recent works (He et al., 2021; Liu et al., 2024) have introduced FL frameworks that enable collaborative GNN training without sharing graph data. Subgraph-FL aims to leverage disjoint graphs from each local client to collaboratively train a global model for solving downstream tasks. Existing studies (Zhang et al., 2021a; Yao et al., 2024; Wu et al., 2021a; Liu et al., 2024) have attempted to supplement the local absent knowledge among local graphs that each client currently holds. For instance, FedSAGE+ (Zhang et al., 2021a), FedGNN (Wu et al., 2021a), and FedGCN (Yao et al., 2024) request node information from other clients to recover missing neighborhood nodes and compensate for potential edges. FedPUB (Baek et al., 2023) and FedStar (Tan et al., 2023) aim to personalize local models by adapting the global model to specialize in the local data of each client. However, due to the mutable properties of graph domains, subgraph-FL must generalize well not only to the current label distribution but also to new nodes that will emerge in the future. Unlike these approaches (Zhang et al., 2021a; Wu et al., 2021a; Yao et al., 2024; Baek et al., 2023) that only focus on finding missing knowledge relevant to the current state, our model learns representations for all classes and their connection patterns, ensuring better generalization across various future scenarios.

### 2.2 LOCAL OVERFITTING IN FEDERATED LEARNING

Imbalanced data distribution is common in real-world scenarios, and significant efforts (Cui et al., 2019; Menon et al., 2020; Tan et al., 2020; Yun et al., 2022; Li et al., 2022; Ma et al., 2023) have been made to address the resulting deterioration in model performance. Federated learning, the task at hand, inherently faces the data imbalance problem as well. Specifically, the involvement of multiple clients means that each client has its own imbalanced dataset, making local models prone to overfitting to their local data (Zhang et al., 2022; Lee et al., 2022; Guo et al., 2024). Recent works (Chen et al., 2023; Lee et al., 2022; Zhang et al., 2022; Guo et al., 2024) aim to alleviate local overfitting in FL by regularizing local models to be similar to the global model. FedHKD (Chen et al., 2023), FedLC (Zhang et al., 2022), and FedED (Guo et al., 2024) introduce logit calibration, which aligns the logits of each class in the local models more closely with those of the global model. While FedED (Guo et al., 2024) addresses the missing class problem in FL, it does not consider the noisy properties of tail data (Subramonian et al., 2024; Liu et al., 2021; Wu et al., 2021b; Xiao et al., 2021). Our method, FedLoG, addresses local overfitting and ensures reliable representation of all classes by leveraging class-specific knowledge across clients and considering their structural properties. To the best of our knowledge, this is the first work to tackle local overfitting with missing classes in subgraph-FL.

## 3 PRELIMINARIES

**Notations.** We use $\mathcal{G} = (\mathcal{V}, \mathcal{E})$ to denote a graph with the set of nodes $\mathcal{V}$ and the set of edges $\mathcal{E} \subseteq \mathcal{V} \times \mathcal{V}$. The dataset $\mathcal{D} = (\mathcal{G}, Y)$ includes labels $Y$ for the nodes that belong to one of $|\mathcal{C}_{\mathcal{V}}|$ distinct classes, and $\boldsymbol{X}_{\mathcal{V}} \in \mathbb{R}^{|\mathcal{V}| \times d}$ is the feature matrix with $d$ as the feature dimension, where each node $v \in \mathcal{V}$ is associated with a feature vector $\boldsymbol{x}_v \in \mathbb{R}^d$. In subgraph-FL, a server $S$ and $K$ clients manage disjoint subgraphs $\mathcal{G}_k = (\mathcal{V}_k, \mathcal{E}_k)$ for each client $k$. The global set of nodes is $\mathcal{V} = \bigcup_{k=1}^{K} \mathcal{V}_k$ with $\mathcal{V}_i \cap \mathcal{V}_j = \varnothing$ for all $i \neq j$. The local dataset for client $k$ is $\mathcal{D}_k = (\mathcal{G}_k, Y_k)$, and the combined local dataset is $\mathcal{D}^{\text{local}} = \bigcup_{k=1}^{K} \mathcal{D}_k$. Additionally, we generate a global synthetic set $\mathcal{D}^{\text{global}} = (\mathcal{G}^{\text{global}}, Y^{\text{global}})$, where $\mathcal{G}^{\text{global}} = (\mathcal{V}^{\text{global}}, \mathcal{E}_{\varnothing})$ consists of isolated nodes $v_g \in \mathcal{V}^{\text{global}}$ with no edges $\mathcal{E}_{\varnothing}$. $\mathcal{V}^{\text{global}}$ includes $s$ nodes per class, totaling $s \times |\mathcal{C}_{\mathcal{V}}|$ nodes.

**Problem Statement.** We aim to develop a distributed learning framework for collaborative training of a node classifier. Specifically, the classifier $F$ uses optimized parameters $\phi$ to minimize a predefined task loss. The objective is to find global parameters $\phi^*$ that minimizes the aggregated local empirical risk $\mathcal{R}$, defined as: $\phi^* = \arg\min_\phi \mathcal{R}(F(\phi)) = \frac{1}{K} \sum_{k=1}^{K} \mathcal{R}_k(F_k(\phi))$, where $\mathcal{R}_k(F_k(\phi)) := \mathbb{E}_{(\mathcal{G}_k, Y_k) \sim \mathcal{D}^{\text{local}}} [\mathcal{L}_k(F_k(\phi; \mathcal{G}_k), Y_k)]$ and the task-specific loss $\mathcal{L}_k$ is defined as: $\mathcal{L}_k := \frac{1}{|\mathcal{V}_k|} \sum_{v_k \in \mathcal{V}_k} l(\phi; \mathcal{G}_k(v_k), y_{v_k}) + \frac{1}{|\mathcal{V}^{\text{global}}|} \sum_{v_g \in \mathcal{V}^{\text{global}}} l(\phi; v_g, y_{v_g})$. To allow each client to generalize across all classes, including missing classes, we generate global synthetic data $\mathcal{D}^{\text{global}}$ and introduce an additional loss term to take into account this data to prevent local overfitting.

### 3.1 WHICH DATA ARE RELIABLE?

Data reliability refers to the accuracy and consistency of information from decentralized nodes, crucial for training models across varied environments (i.e., clients). Inspired by the robust performance of

GNNs on head class and head degree nodes (Yun et al., 2022; Park et al., 2021; Zhao et al., 2021; Liu et al., 2021), we found that data reliability largely depends on **1)** the extent of data connections (i.e., degree headness) and **2)** the predominance of certain classes (i.e., class headness).

To corroborate our arguments, we measured the target class accuracy of a client (receiver) receiving information (i.e., weights) from other clients (contributors). To check how degree/class headness of the contributors impacts the receiver, we varied the contributors' training sets, adjusting the degree/class headness of the training nodes, while keeping the receiver's training set constant. The global model was constructed by averaging client weights and evaluated on the receiver's local graph.

Figure 1(top) shows receiver accuracy with contributor training sets composed of **1)** 'Only head degree nodes', **2)** 'Only tail degree nodes', and **3)** 'Balanced (head+tail) degree nodes'. The receiver's performance is better when knowledge is received from head degree nodes, indicating the reliability of knowledge from head degree nodes over that from tail degree nodes. Moreover, as the number of clients increases, the performance gap widens, highlighting the accumulated negative impact of noise within tail degree nodes.

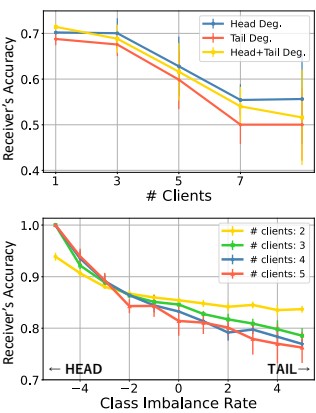

Figure 1(bottom) manipulates label distribution within each contributor's local graph, transforming the target class into a tail or head class by varying the number of training nodes in other classes while keeping the target class constant. When the class headness of the target class within contributors is high (i.e., negative imbalance rates), contributors enhance the receiver's performance. Conversely, with low headness (i.e., positive imbalance rates), the receiver's performance deteriorates as contributors struggle to represent the target class (Yun et al., 2022). This negative impact is magnified as the number of clients increases.

Figure 1: Data Reliability Analysis[1](PubMed used).

In summary, data with 'headness' in both degree and class from other clients (i.e., contributors) helps the target client (i.e., receiver) learn reliable representations, while 'tailness' data negatively impacts the model training due to insufficient or noisy information. Building on our observations, our method, FedLoG, collects knowledge from head degree and head class data across all clients to alleviate locally absent knowledge.

## 4 PROPOSED METHODOLOGY: FEDLOG

Our proposed subgraph-FL framework, FedLoG, works as follows. Figure 2 shows the overall framework of FedLoG.

- **Step 1 – Local Fitting (Section 4.1):** The server initializes the local model parameters of $K$ clients with the parameters of the global model $\phi^{\text{global}}$. Each local model is then trained using local data $\mathcal{D}_k$. Concurrently, head degree and tail degree knowledge are condensed into synthetic nodes within each client, denoted as $\mathcal{V}_{k,\text{head}}$ and $\mathcal{V}_{k,\text{tail}}$.

- **Step 2 – Global Aggregation and Global Synthetic Data Generation (Section 4.2):** After local training, the server aggregates the local models to create the global model $\phi^{\text{global}}$ and generates global synthetic data $\mathcal{D}^{\text{global}}$ by aggregating $\mathcal{V}_{k,\text{head}}$ for all $k$, weighted according to the head classes within each client $k$.

- **Step 3 – Local Fitting (Section 4.1) & Local Generalization (Section 4.3):** At the start of each round, local fitting is performed first. After local fitting, local models are generalized using $\mathcal{D}^{\text{global}}$ which possess both head degree and head class knowledge, adaptively learning the locally absent knowledge.

While the framework starts with Step 1 ($r = 0$), it continues to alternate between Steps 2 and 3 until the final round $R$ is reached ($r \geq 1$). In summary, our method extracts head degree knowledge at the client level and head class knowledge at the server level, then condense them into the global synthetic data, which is utilized to train the local model during Local Generalization to adaptively compensate for the locally absent knowledge within each client. Algorithm 1 outlines the algorithm of FedLoG.

---

[1]Please refer to Appendix A.8 for detailed description of experimental settings.

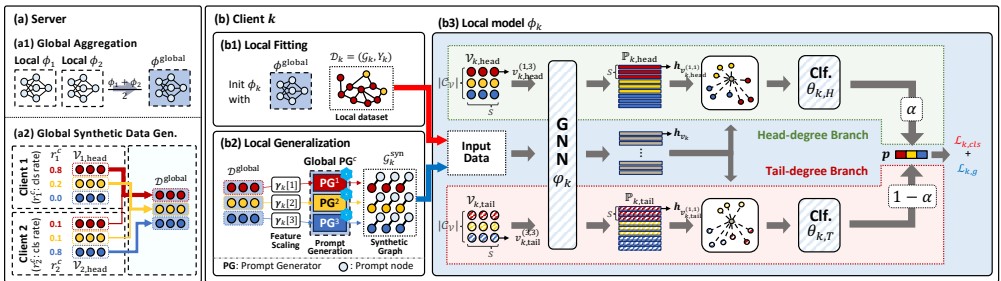

Figure 2: Overview of FedLoG with 2 Clients and 3 Classes.

## 4.1 LOCAL FITTING

The local model for each client $k$ consists of one GNN embedder ($\varphi_k$) and two classifiers ($\theta_{k,H}$ and $\theta_{k,T}$), for the head and tail degree branches, respectively, as shown in Figure 2(b3). Each branch forms a prototypical network (Snell et al., 2017)-based architecture for prediction. While a prototypical network is originally designed for few-shot learning, we repurpose it by designating learnable nodes as prototypes, enabling both prediction and the condensation of class information into these prototypes. Specifically, each branch contains $|\mathcal{C}_\mathcal{V}| \times s$ learnable nodes, each with features of dimension $d$, allocating $s$ nodes per class. Thus, each client $k$ has learnable node sets $\mathcal{V}_{k,\text{head}}$ and $\mathcal{V}_{k,\text{tail}}$ with features $\boldsymbol{X}_{\mathcal{V}_{k,\text{head}}} \in \mathbb{R}^{(|\mathcal{C}_\mathcal{V}| \times s) \times d}$ and $\boldsymbol{X}_{\mathcal{V}_{k,\text{tail}}} \in \mathbb{R}^{(|\mathcal{C}_\mathcal{V}| \times s) \times d}$, respectively. At the client level, we condense knowledge from locally observed nodes into these learnable nodes. Head degree nodes are condensed into $\mathcal{V}_{k,\text{head}}$ and tail degree nodes into $\mathcal{V}_{k,\text{tail}}$, integrating condensation and prediction into a single process. The detailed processes are described as follows (See Figure 2(b1) and (b3)):

**(Initialization)** In the initial round ($r = 0$), we initialize the local model weights for each client $k$, denoted as $\phi_k = \{\varphi_k, \theta_{k,H}, \theta_{k,T}\}$, with the global set of parameters $\phi^{\text{global}} = \{\varphi^{\text{global}}, \theta_H^{\text{global}}, \theta_T^{\text{global}}\}$.

**(Embedding)** Given client $k$'s local graph $\mathcal{G}_k = (\mathcal{V}_k, \mathcal{E}_k)$ where the feature of each node $v_k \in \mathcal{V}_k$ is initialized with $\boldsymbol{h}_{v_k}^{(0)} = \boldsymbol{x}_{v_k}$, a shared GraphSAGE (Hamilton et al., 2017) GNN encoder $\varphi_k$ is employed to embed each local node $v_k \in \mathcal{V}_k$ and learnable nodes $v_{k,\text{head}} \in \mathcal{V}_{k,\text{head}}$ and $v_{k,\text{tail}} \in \mathcal{V}_{k,\text{tail}}$:

$$\boldsymbol{h}_{v_k} = \text{GNN}_{\varphi_k}(v_k, \mathcal{G}_k), \quad \boldsymbol{h}_{v_{k,\text{head}}} = \text{GNN}_{\varphi_k}(v_{k,\text{head}}, \mathcal{G}_{k,\text{head}}), \quad \boldsymbol{h}_{v_{k,\text{tail}}} = \text{GNN}_{\varphi_k}(v_{k,\text{tail}}, \mathcal{G}_{k,\text{tail}}) \quad (1)$$

where $\boldsymbol{h}_{v_k}$, $\boldsymbol{h}_{v_{k,\text{head}}}$, and $\boldsymbol{h}_{v_{k,\text{tail}}}$ are the representations of nodes $v_k$, $v_{k,\text{head}}$, and $v_{k,\text{tail}}$, respectively. $\mathcal{G}_{k,\text{head}} = (\mathcal{V}_{k,\text{head}}, \mathcal{E}_\varnothing)$ and $\mathcal{G}_{k,\text{tail}} = (\mathcal{V}_{k,\text{tail}}, \mathcal{E}_\varnothing)$ denote synthetic graphs constructed by learnable nodes from head and tail degree branches, respectively. It is worth noting that the learnable nodes do not adhere to a specific graph structure but share the same GNN encoder with the local graph, allowing us to condense structural information into the features of the learnable nodes.

After acquiring node representations, we generate model predictions in each branch using class prototypes, which are the representations of learnable nodes. Since the process of generating model predictions is identical in both branches, we only explain the head branch here. In the head branch, prototypes are defined as $\mathbb{P}_{k,\text{head}} = \{\boldsymbol{h}_{v_{k,\text{head}}^{(1,1)}}, \ldots, \boldsymbol{h}_{v_{k,\text{head}}^{(1,s)}}, \ldots, \boldsymbol{h}_{v_{k,\text{head}}^{(|\mathcal{C}_\mathcal{V}|,1)}}, \ldots, \boldsymbol{h}_{v_{k,\text{head}}^{(|\mathcal{C}_\mathcal{V}|,s)}}\}$, with $s$ proto-types per class. To ensure all class information contributes to the final prediction, the target node representations are further updated based on feature differences with all prototypes assigned to each class as follows: $\boldsymbol{h}'_{v_k} = \theta_{k,H}(\boldsymbol{h}_{v_k}, \{(\boldsymbol{h}_{v_k} - \boldsymbol{h}_{v_{k,\text{head}}^{(1,1)}}), \ldots, (\boldsymbol{h}_{v_k} - \boldsymbol{h}_{v_{k,\text{head}}^{(|\mathcal{C}_\mathcal{V}|,s)}})\})$. Please refer to Appendix A.4 for more details on $\theta_{k,H}$. Then, the class probability for target node $v_k$ is given as follows:

$$p(c|\boldsymbol{h}'_{v_k}) = \frac{\exp(-d(\boldsymbol{h}'_{v_k}, \bar{\boldsymbol{h}}_{\mathcal{V}_{k,\text{head}}^c}))}{\sum_{c'=1}^{|\mathcal{C}_\mathcal{V}|} \exp(-d(\boldsymbol{h}'_{v_i}, \bar{\boldsymbol{h}}_{\mathcal{V}_{(k,\text{head})}^{c'}}))}, \quad (2)$$

where $d(\cdot, \cdot)$ is the squared Euclidean distance and $\bar{\boldsymbol{h}}_{\mathcal{V}_{k,\text{head}}^c}$ indicates the average of prototype representations of class $c$, i.e., $\bar{\boldsymbol{h}}_{\mathcal{V}_{k,\text{head}}^c} = \frac{1}{s} \sum_{i=1}^{s} \boldsymbol{h}_{v_{k,\text{head}}^{(c,i)}}$. By updating the target node's representation based on its relationships with all prototypes and minimizing the distance to its correct class prototype, we aim to guide the synthetic learnable node to effectively learn a representation that reflects both the correct class and the broader relationships among all classes.

To obtain final prediction $\boldsymbol{p}$, we combine the class probabilities from both branches $\boldsymbol{p}_{\text{head}}$ and $\boldsymbol{p}_{\text{tail}}$ by weighting them based on the degree value of the target node $v_k$ (i.e., $\deg(v_k)$) as follows:

$$\boldsymbol{p} = \alpha \cdot \boldsymbol{p}_{\text{head}} + (1 - \alpha) \cdot \boldsymbol{p}_{\text{tail}}, \quad (3)$$

where $\alpha = 1/(1 + e^{-(\deg(v_k) - (\lambda+1))})$, and $\lambda$ is the tail degree threshold outlined in the Appendix A.9. Note that the hyperparameter $\alpha$ balances the influence between the head and tail branches based on the node's degree. Specifically, $\alpha$ acts as a weight for the prediction loss contributed by each branch for the target node. For instance, when a target node has a high degree, $\alpha$ becomes large, increasing the influence of the head degree branch on the final predictions. Consequently, knowledge from high-degree nodes is condensed into $\mathcal{V}_{k,\text{head}}$, as the model significantly updates the learnable features $\boldsymbol{X}_{\mathcal{V}_{k,\text{head}}}$ of $\mathcal{V}_{k,\text{head}}$ to minimize prediction loss. Conversely, when a node has a low degree, its knowledge is condensed into $\boldsymbol{X}_{\mathcal{V}_{k,\text{tail}}}$ of $\mathcal{V}_{k,\text{tail}}$ instead. Additionally, the sigmoid-based formulation of $\alpha$ prevents nodes with extremely high degrees from dominating the learnable features, ensuring a balanced contribution across all nodes.

**Training objective for each client** $k$. The prediction loss for each client $k$ is calculated as $\mathcal{L}_{k,cls} = \sum_{v_k \in \mathcal{V}_k} \sum_{c \in \mathcal{C}_\mathcal{V}} -\mathbb{I}(y_{v_k} = c) \log(\boldsymbol{p}[c])$. Furthermore, to ensure the stability of the condensation process, we minimize the $L_2$ norm of the learnable features, denoted as $\mathcal{L}_{k,norm} = \sum_{v \in \mathcal{V}_{k,\text{head}} \cup \mathcal{V}_{k,\text{tail}}} \|\boldsymbol{x}_v\|_2$. Thus, the total loss for model parameters is $\mathcal{L}_k(\phi_k, \boldsymbol{X}_{\mathcal{V}_{k,\text{head}}}, \boldsymbol{X}_{\mathcal{V}_{k,\text{tail}}}) = \mathcal{L}_{k,cls} + \beta \cdot \mathcal{L}_{k,norm}$, where $\beta$ adjusts the extent of regularization. It is important to note that besides updating the local model weights $\phi_k = \{\varphi_k, \theta_{k,H}, \theta_{k,T}\}$ for each client $k$, the learnable features $\boldsymbol{X}_{\mathcal{V}_{k,\text{head}}}$ and $\boldsymbol{X}_{\mathcal{V}_{k,\text{tail}}}$ are also updated.

## 4.2 GLOBAL AGGREGATION AND GLOBAL SYNTHETIC DATA GENERATION

**Global Aggregation.** As shown in Figure 2(a1), after training the $K$ local clients, the server aggregates the local model weights for round $r$ using the weighted average $\phi^{\text{global}} \leftarrow \frac{1}{K} \sum_{k=1}^{K} \phi_k^{(r)}$.

**Global Synthetic Data Generation.** In addition, as shown in Figure 2(a2), the server generates global synthetic data $\mathcal{D}^{\text{global}}$, which will be employed during the Local Generalization phase (Section 4.3) to help mitigate the issue of local overfitting. More specifically, we first generate node features in the global synthetic data $\mathcal{D}^{\text{global}}$ by merging $\mathcal{V}_{k,\text{head}}$ from all $K$ clients. For example, when merging $\mathcal{V}_{k,\text{head}}$ for all $K$ clients regarding a class $c$, the server gives more weight to the synthetic data from client $k$ for whom class $c$ belongs to the head classes within the local data $\mathcal{D}_k$. This is to take into account the expert knowledge of each client regarding the dominant classes within its local data, which is supported by our empirical analysis in Section 3.1. More formally, for each class $c \in \mathcal{C}_\mathcal{V}$, the feature vector of the $i$-th global synthetic node for class $c$, i.e., $\boldsymbol{x}_{v_g^{(c,i)}} \in \mathbb{R}^d$, is generated as follows:

$$\boldsymbol{x}_{v_g^{(c,i)}} = \frac{1}{\sum_{k=1}^{K} r_k^c} \sum_{k=1}^{K} r_k^c \boldsymbol{x}_{v_{k,\text{head}}^{(c,i)}}, \tag{4}$$

where $\boldsymbol{x}_{v_{k,\text{head}}^{(c,i)}} \in \boldsymbol{X}_{\mathcal{V}_{k,\text{head}}}$ and $r_k^c = \frac{|\mathcal{V}_k^c|}{|\mathcal{V}_k|}$ represents the proportion of nodes belonging to class $c$ in the $k$-th client's dataset. By giving more weight to clients with expertise in each head class, the server effectively combines the most reliable knowledge from all clients to create the global synthetic nodes. This process results in $|\mathcal{C}_\mathcal{V}| \times s$ global synthetic nodes $\mathcal{V}^{\text{global}}$ with features $\boldsymbol{X}_{\mathcal{V}^{\text{global}}} = \{\boldsymbol{x}_{v_g^{(1,1)}}, \ldots, \boldsymbol{x}_{v_g^{(1,s)}}, \ldots, \boldsymbol{x}_{v_g^{(|\mathcal{C}_\mathcal{V}|,1)}}, \ldots, \boldsymbol{x}_{v_g^{(|\mathcal{C}_\mathcal{V}|,s)}}\}$. The final global synthetic data is represented as $\mathcal{D}^{\text{global}} = (\mathcal{G}^{\text{global}}, Y^{\text{global}})$, where $\mathcal{G}^{\text{global}} = (\mathcal{V}^{\text{global}}, \mathcal{E}_\varnothing)$, with node features $\boldsymbol{X}_{\mathcal{V}^{\text{global}}}$. At the end of each round $r$, the server distributes the aggregated model weights $\phi^{\text{global}}$ and the generated global synthetic data $\mathcal{D}^{\text{global}}$. In summary, considering both the degree headness through the learnable features $\boldsymbol{X}_{\mathcal{V}_{k,\text{head}}}$ and the class headness through $r_k^c$ allows the global features $\boldsymbol{X}_{\mathcal{V}^{\text{global}}}$ to contain knowledge about both head degree and head class across all clients.

### 4.2.1 DISCUSSIONS ON THE GRAPH STRUCTURES OF GLOBAL SYNTHETIC NODES

Note that even though each global synthetic node contains only features without explicit graph structures, these features still implicitly capture the original graph structure. This is because these features are learned using a graph embedder $\varphi_k$ that is shared across both synthetic and original nodes. As a result, the structural information of the original graph is condensed into the features of the synthetic nodes.

### 4.3 LOCAL GENERALIZATION

At the beginning of each round ($r \geq 1$), each client $k$ initializes its local model $\phi_k$ with the distributed global model parameters $\phi^{\text{global}}$ followed by the local update of $\phi_k$ based on its local data $D_k$ (i.e.,

Local Fitting described in Section 4.1). Then, as shown in Figure 2(b2) and (b3), we additionally train the local model with the global synthetic data $\mathcal{D}^{\text{global}}$, enabling the model to generalize to locally absent knowledge (i.e., Local Generalization), such as tail and missing classes. Since each client has different locally absent knowledge, we first adaptively customize the global synthetic data for the current state of the local model through two strategies, i.e., 1) feature scaling and 2) prompt generation, and then train the local model with the customized data.

**Strategy 1) Feature Scaling.** When the local model is strongly biased toward the local distribution, it tends to assign high logits to the dominant class, making it difficult to predict tail or missing classes. To address this issue, we apply strong perturbation to the training data of the dominant class to help balance predictions across all classes, allowing the model to effectively learn absent knowledge within the local data. To achieve this, we use feature scaling for the perturbation on the global synthetic data $\boldsymbol{X}_{\mathcal{V}^{\text{global}}} = \left\{ \boldsymbol{x}_{v_g^{(1,1)}}, \ldots, \boldsymbol{x}_{v_g^{(1,s)}}, \ldots, \boldsymbol{x}_{v_g^{(|\mathcal{C}_{\mathcal{V}}|,1)}}, \ldots, \boldsymbol{x}_{v_g^{(|\mathcal{C}_{\mathcal{V}}|,s)}} \right\}$ as follows:

$$\hat{\boldsymbol{x}}_{v_g^{(c,i)}} = \boldsymbol{x}_{v_g^{(c,i)}} + \boldsymbol{\gamma}_k[c] \cdot (\bar{\boldsymbol{x}}_{\mathcal{V}^{\text{global}}} - \boldsymbol{x}_{v_g^{(c,i)}}), \text{ where } \bar{\boldsymbol{x}}_{\mathcal{V}^{\text{global}}} = \frac{1}{\left| \mathcal{V}^{\text{global}} \right|} \sum_{\boldsymbol{x}_{v_g} \in \boldsymbol{X}_{\mathcal{V}^{\text{global}}}} \boldsymbol{x}_{v_g}, \quad (5)$$

where $\boldsymbol{\gamma}_k \in \mathbb{R}^{|\mathcal{C}_{\mathcal{V}}|}$ is the class-wise adaptive factor that adjusts the strength of the perturbation by moving the global synthetic data of class $c$ to the average of global synthetic data, making it harder to predict. Note that when $\boldsymbol{\gamma}_k[c] = 1$, the global synthetic data for class $c$ is completely replaced with the average of the global synthetic data, whereas the global synthetic data for class $c$ remains unchanged when $\boldsymbol{\gamma}_k[c] = 0$. During training, we dynamically modify the factor by incrementing it by 0.001 whenever the local model's accuracy for class $c$ exceeds the threshold at the end of the round, thereby increasing the perturbation of the corresponding class.

**Strategy 2) Prompt Generation.** While we have focused on condensing knowledge into the global synthetic data, we have not yet addressed how to train our GNN encoder on this data. Recall that the GNN encoder $\varphi_k$ is shared across both the synthetic and original nodes. However, since global synthetic nodes only contain features without an explicit graph structure while the original nodes involve a graph structure, the parameters of the shared GNN encoder would be differently affected by them, leading to discrepancies in the gradient matrices.

Our main idea to solve this issue is to ensure that synthetic nodes are trained in the same environment as original nodes within the local graph. Specifically, the objective of the prompt generator is to make the synthetic graph—which consists of a target node and its corresponding prompt node derived from the target node's features—produce the same gradient matrix as when the target node is predicted using its true $h$-hop subgraph within the local graph. Details on pretraining the prompt generators are in Appendix A.7. More formally, we generate a prompt node for each feature-scaled global synthetic node using the class-specific prompt generator $\mathbf{PG}^c$ corresponding to class $c$ (Figure 2(b2)) as follows: $\boldsymbol{x}_{v_{v_g}^p} = \mathbf{PG}^{c=y_{v_g}}(\hat{\boldsymbol{x}}_{v_g})$, where $\boldsymbol{x}_{v_{v_g}^p}$ is the feature of the prompt node $v_{v_g}^p$ corresponding to the node $v_g$. We then construct a synthetic graph $\mathcal{G}_k^{\text{syn}} = \bigcup_{v_g \in \mathcal{V}^{\text{global}}} \mathcal{G}_{k,v_g}^{\text{syn}}$ for client $k$, where each $\mathcal{G}_{k,v_g}^{\text{syn}}$ consists of a global synthetic node $v_g$ and its prompt node $v_{v_g}^p$, connected to each other.

**Training with Customized Data.** Using the synthetic graph $\mathcal{G}_k^{\text{syn}}$, the prediction for the target synthetic nodes within $\mathcal{V}^{\text{global}}$ follows Eqs. 1-3 using the same local model $\phi_k$ as for normal nodes $\mathcal{V}_k$, with $\alpha$ set to 0.5. The prediction loss for $\mathcal{D}^{\text{global}}$ is $\mathcal{L}_{k,g} = \sum_{v_g \in \mathcal{V}^{\text{global}}} \sum_{c \in \mathcal{C}_{\mathcal{V}}} -\mathbb{I}(y_{v_g} = c) \log(\boldsymbol{p}[c])$. At the end of each round, we adjust the adaptive factor $\boldsymbol{\gamma}_k[c] \forall c$ based on the class prediction accuracy.

In summary, the final loss of the local model is: $\mathcal{L}_k = \mathcal{L}_{k,cls} + \mathcal{L}_{k,g} + \beta \cdot \mathcal{L}_{k,norm}$.

# 5 EXPERIMENTS

## 5.1 EXPERIMENTAL SETTINGS

**Datasets.** We conduct experiments on five real-world graph datasets. Distributed subgraphs are constructed by dividing each dataset into a certain number of clients using the METIS graph partitioning algorithm (Karypis & Kumar, 1997). The datasets used are Cora (Sen et al., 2008), CiteSeer (Sen et al., 2008), PubMed (Sen et al., 2008), Amazon Computer (McAuley et al., 2015), and Amazon Photo (Shchur et al., 2018). For more details, see Appendix A.12.

**Baseline Methods. 1) Local:** Refers to local training without any weight sharing. **2) FedAvg** (McMahan et al., 2017): The most widely-used FL baseline. **3) FedSAGE+** (Zhang et al., 2021a), **4) FedGCN** (Yao et al., 2024) and **5) FedPUB** (Baek et al., 2023): subgraph-FL baselines that primarily

Table 1: Model performance on Seen Graph settings. Mean accuracy with std. over 3 runs.

| | | Cora | | | CiteSeer | | | PubMed | | | Amazon Photo | | | Amazon Computers | | |
|---|---|---|---|---|---|---|---|---|---|---|---|---|---|---|---|---|
| | **Methods** | 3 Clients | 5 Clients | 10 Clients | 3 Clients | 5 Clients | 10 Clients | 3 Clients | 5 Clients | 10 Clients | 3 Clients | 5 Clients | 10 Clients | 3 Clients | 5 Clients | 10 Clients |
| **Seen Graph** | **Local** | 0.7357 (0.0030) | 0.7325 (0.0066) | 0.8039 (0.0008) | 0.6674 (0.0069) | 0.6647 (0.0045) | 0.7128 (0.0035) | 0.8445 (0.0003) | 0.8108 (0.0000) | 0.8024 (0.0011) | 0.6724 (0.0003) | 0.7959 (0.0106) | 0.7562 (0.0137) | 0.6523 (0.0221) | 0.5764 (0.0001) | 0.6645 (0.0051) |
| | **FedAvg** | 0.8416 (0.0044) | 0.6332 (0.0166) | 0.7162 (0.0382) | 0.7426 (0.0024) | 0.7498 (0.0049) | 0.7252 (0.0035) | 0.7126 (0.0000) | 0.8640 (0.0024) | 0.8586 (0.0010) | 0.7668 (0.0414) | 0.5695 (0.0483) | 0.5669 (0.0974) | 0.5626 (0.0715) | 0.4195 (0.0173) | 0.4858 (0.0187) |
| | **FedSAGE+** | 0.7560 (0.0237) | 0.4156 (0.0034) | 0.3522 (0.1196) | 0.7505 (0.0150) | 0.5167 (0.0389) | 0.4929 (0.0075) | 0.8980 (0.0001) | 0.9091 (0.0025) | 0.9041 (0.0012) | 0.9239 (0.0083) | 0.6670 (0.0206) | 0.6246 (0.0585) | 0.7539 (0.0062) | 0.6934 (0.0006) | 0.6656 (0.0082) |
| | **FedGCN** | 0.8226 (0.0062) | 0.8124 (0.0158) | 0.7243 (0.0172) | 0.7376 (0.0111) | 0.7649 (0.0010) | 0.7123 (0.0122) | 0.7127 (0.0000) | 0.8504 (0.0011) | 0.8441 (0.0070) | 0.7398 (0.0036) | 0.5717 (0.0583) | 0.5627 (0.0957) | 0.5782 (0.0623) | 0.4217 (0.0243) | 0.4908 (0.0183) |
| | **FedPUB** | 0.8476 (0.0021) | 0.8448 (0.0009) | **0.8622** (0.0059) | 0.7455 (0.0065) | 0.7694 (0.0074) | 0.7505 (0.0081) | 0.9064 (0.0016) | 0.9069 (0.0019) | 0.9092 (0.0019) | 0.9399 (0.0020) | 0.9122 (0.0016) | 0.8983 (0.0052) | 0.8339 (0.0142) | 0.8202 (0.0141) | 0.8181 (0.0124) |
| | **FedNTD** | 0.8452 (0.0067) | 0.8526 (0.0024) | 0.6984 (0.0030) | 0.7455 (0.0069) | **0.7826** (0.0047) | 0.7146 (0.0079) | 0.9049 (0.0002) | 0.9065 (0.0009) | 0.9061 (0.0012) | 0.9378 (0.0029) | 0.9166 (0.0021) | 0.9119 (0.0036) | 0.8492 (0.0107) | 0.8619 (0.0034) | 0.8707 (0.0055) |
| | **FedED** | 0.8542 (0.0084) | 0.8398 (0.0024) | 0.6779 (0.0343) | 0.7305 (0.0086) | 0.7624 (0.0050) | 0.6251 (0.0149) | 0.9080 (0.0006) | 0.9086 (0.0027) | 0.8985 (0.0025) | 0.9463 (0.0014) | 0.9101 (0.0027) | 0.8950 (0.0059) | 0.8623 (0.0136) | 0.8722 (0.0035) | 0.8356 (0.0158) |
| | **FedLoG** | **0.8601** (0.0118) | **0.8575** (0.0074) | 0.8451 (0.0103) | **0.7663** (0.0086) | 0.7728 (0.0049) | **0.7624** (0.0063) | **0.9180** (0.0005) | **0.9129** (0.0015) | **0.9115** (0.0043) | **0.9653** (0.0020) | **0.9496** (0.0037) | **0.9305** (0.0049) | **0.9073** (0.0012) | **0.8986** (0.0014) | **0.8742** (0.0107) |

address missing knowledge within the current local label distribution. To ensure a fair comparison, we also evaluate our method against **6) FedNTD** (Lee et al., 2022) and **7) FedED** (Guo et al., 2024), which address local overfitting in FL. For more details, see Appendix A.13.

**Evaluation Protocol.** We perform FL for 100 rounds. Node classification accuracy is measured on the client side and averaged across all clients over three runs. More details are in Appendix A.14.

## 5.2 Experiment Results

**Q1. How does FedLoG perform in conventional FL settings?** Table 1 presents the evaluation of models on graphs that were used for training. The label distributions of the test nodes match the training label distribution of each client. We refer to this conventional setting as **Seen Graph**, where models are evaluated on test nodes within the same graph structure as the training nodes (i.e., transductive setting (Kipf & Welling, 2016)). The overall performance of FedLoG on the 'Seen Graph' outperforms that of other baselines, demonstrating its strong performance in conventional settings.

**Q2. Does FedLoG generalize to unseen data after local updates?** In this section, we introduce practical and novel test settings for subgraph-FL, aiming to assess the model's ability to generalize to potential unseen data in real-world scenarios. By proposing these new settings, we emphasize the importance of evaluating models beyond conventional FL settings, and demonstrate that our method consistently achieves superior performance compared to other baselines. We introduce three practical scenarios for unseen data:

**1) Unseen Node (Table 2(a)).** Each client has new nodes with seen classes added to its local graph, which introduce structural changes. We perform evaluations on the new nodes to assess how well the FL framework adapts to these structural changes. **2) Missing Class (Table 2(b)).** Each client has new nodes with missing classes added to its local graph. We evaluate the performance on new nodes representing missing classes for each client, assessing how effectively the FL framework enables the local model to learn previously absent knowledge. **3) New Client (Table 2(c)).** A new client that has never participated in the FL framework emerges. This client has a distinct label distribution and graph structure. We assess how well the FL framework generalizes to accommodate this new client, ensuring robust performance across diverse scenarios. To do so, we perform evaluations on the unseen graph of the new client using each trained local model without the new client being involved in the training, and then report the mean accuracy over all clients. These approaches help simulate real-world scenarios where clients have incomplete information. Detailed settings are described in Appendix A.10 and A.16.

In Table 2, we observe that FedLoG outperforms baselines by preventing local overfitting and effectively addressing unseen data. Specifically, in the Unseen Node and Missing Class settings, FedLoG shows superior performance on the added nodes, even with missing classes. For the Missing Class, which requires extensive knowledge from other clients, Local and personalized FL models like FedPUB (Baek et al., 2023) fail to predict the missing classes as they optimize for the training label distribution. Although FedSAGE+ (Zhang et al., 2021a) and FedGCN (Yao et al., 2024) attempt to compensate for missing neighbors, they are not always effective because the missing class is not always within the neighbors. Moreover, FedNTD (Lee et al., 2022) and FedED (Guo et al., 2024) address local overfitting and achieve relatively high performance in missing class prediction. However, they regularize the local model logits to match the global model, risking noisy information from tail data and resulting in inconsistent performance across different settings. In contrast, FedLoG alleviates local overfitting by using reliable class representations and structural information across clients, reducing the emphasis on noisy information. Thus, FedLoG successfully addresses unseen data, ensuring robust performance even with unseen graph structures (New Client) due to its generalization ability across all classes and structural features.

Table 2: Model performance in FL settings on unseen data. Mean accuracy with std. over 3 runs.

**(a) Unseen Node**

| Methods | Cora 3 Clients | Cora 5 Clients | Cora 10 Clients | CiteSeer 3 Clients | CiteSeer 5 Clients | CiteSeer 10 Clients | PubMed 3 Clients | PubMed 5 Clients | PubMed 10 Clients | Amazon Photo 3 Clients | Amazon Photo 5 Clients | Amazon Photo 10 Clients | Amazon Computers 3 Clients | Amazon Computers 5 Clients | Amazon Computers 10 Clients |
|---|---|---|---|---|---|---|---|---|---|---|---|---|---|---|---|
| Local | 0.1250 (0.0030) | 0.2957 (0.0079) | 0.2854 (0.0263) | 0.4443 (0.0131) | 0.3471 (0.0020) | 0.5177 (0.0052) | 0.7510 (0.0010) | 0.7292 (0.0000) | 0.7489 (0.0013) | 0.1333 (0.0000) | 0.1900 (0.0000) | 0.3958 (0.0000) | 0.1687 (0.0000) | 0.2488 (0.0000) | 0.3890 (0.0043) |
| FedAvg | 0.5403 (0.0797) | 0.5198 (0.0179) | 0.4139 (0.1308) | 0.6585 (0.0220) | 0.6098 (0.0301) | 0.6199 (0.0084) | 0.6154 (0.0010) | 0.8189 (0.0120) | 0.8070 (0.0043) | 0.1782 (0.0419) | 0.2125 (0.0116) | 0.3727 (0.0221) | 0.2275 (0.1017) | 0.3095 (0.0179) | 0.4177 (0.0335) |
| FedSAGE+ | 0.5653 (0.0546) | 0.4265 (0.0062) | 0.3836 (0.0705) | 0.6572 (0.0093) | 0.4023 (0.0339) | 0.6154 (0.0034) | 0.8944 (0.0048) | 0.8921 (0.0089) | 0.8926 (0.0051) | 0.4781 (0.0093) | 0.2298 (0.0394) | 0.4607 (0.0391) | 0.3462 (0.0325) | 0.3555 (0.0196) | 0.3596 (0.0024) |
| FedGCN | 0.3689 (0.0646) | 0.5877 (0.0018) | 0.5075 (0.0001) | 0.6232 (0.0243) | 0.6530 (0.1095) | 0.6139 (0.0119) | 0.6154 (0.0000) | 0.7759 (0.0058) | 0.8188 (0.0049) | 0.2565 (0.0067) | 0.2604 (0.0077) | 0.3708 (0.0304) | 0.2086 (0.0597) | 0.3084 (0.0140) | 0.4103 (0.0342) |
| FedPUB | 0.5529 (0.0246) | 0.5192 (0.0064) | 0.4767 (0.0286) | 0.6798 (0.0334) | 0.6691 (0.0057) | 0.6938 (0.0245) | 0.8878 (0.0003) | 0.8822 (0.0056) | 0.8836 (0.0043) | 0.4085 (0.0118) | 0.3890 (0.0494) | 0.5033 (0.0155) | 0.4414 (0.0225) | 0.5025 (0.0466) | 0.5253 (0.0471) |
| FedNTD | 0.6355 (0.0195) | 0.5880 (0.0041) | 0.3913 (0.1235) | 0.7057 (0.0173) | 0.7014 (0.0614) | 0.6151 (0.0155) | 0.8939 (0.0068) | 0.8852 (0.0044) | 0.8816 (0.0031) | 0.4042 (0.0155) | 0.5833 (0.0043) | 0.5286 (0.0030) | 0.5056 (0.0440) | 0.6034 (0.0309) | 0.6482 (0.0226) |
| FedED | 0.7338 (0.0294) | 0.5514 (0.0117) | 0.3916 (0.1184) | 0.6646 (0.0658) | 0.6148 (0.0097) | 0.5381 (0.0781) | 0.9008 (0.0027) | 0.8884 (0.0036) | 0.8730 (0.0077) | 0.6227 (0.0429) | 0.4265 (0.0675) | 0.4629 (0.0137) | 0.4582 (0.0176) | 0.5408 (0.0223) | 0.4940 (0.0275) |
| **FedLoG** | **0.7341** (0.0273) | **0.7413** (0.0316) | **0.7406** (0.0527) | **0.7624** (0.0522) | **0.7415** (0.0142) | **0.8044** (0.0078) | **0.9044** (0.0021) | **0.8956** (0.0035) | **0.8965** (0.0061) | **0.7065** (0.0715) | **0.7077** (0.0571) | **0.7176** (0.0277) | **0.7677** (0.0237) | **0.8156** (0.0356) | **0.6735** (0.0292) |

**(b) Missing Class**

| Methods | Cora 3 Clients | Cora 5 Clients | Cora 10 Clients | CiteSeer 3 Clients | CiteSeer 5 Clients | CiteSeer 10 Clients | PubMed 3 Clients | PubMed 5 Clients | PubMed 10 Clients | Amazon Photo 3 Clients | Amazon Photo 5 Clients | Amazon Photo 10 Clients | Amazon Computers 3 Clients | Amazon Computers 5 Clients | Amazon Computers 10 Clients |
|---|---|---|---|---|---|---|---|---|---|---|---|---|---|---|---|
| Local | 0.0000 (0.0000) | 0.0000 (0.0000) | 0.0000 (0.0000) | 0.0000 (0.0000) | 0.0000 (0.0000) | 0.0000 (0.0000) | 0.0000 (0.0000) | 0.0000 (0.0000) | 0.0000 (0.0000) | 0.0000 (0.0000) | 0.0000 (0.0000) | 0.0000 (0.0000) | 0.0000 (0.0000) | 0.0000 (0.0000) | 0.0000 (0.0000) |
| FedAvg | 0.3900 (0.1104) | 0.1119 (0.0202) | 0.0652 (0.0568) | 0.2022 (0.0751) | 0.1914 (0.0140) | 0.3189 (0.0218) | 0.0000 (0.0000) | 0.0013 (0.0013) | 0.0020 (0.0010) | 0.0000 (0.0000) | 0.0000 (0.0000) | 0.0085 (0.0148) | 0.0000 (0.0000) | 0.0000 (0.0000) | 0.0073 (0.0127) |
| FedSAGE+ | 0.5000 (0.0457) | 0.1393 (0.0317) | 0.0287 (0.0111) | 0.5581 (0.0524) | 0.1622 (0.0470) | 0.3701 (0.0528) | 0.0000 (0.0000) | 0.0015 (0.0013) | 0.0034 (0.0004) | 0.0000 (0.0000) | 0.0000 (0.0000) | 0.0036 (0.0051) | 0.0000 (0.0000) | 0.0000 (0.0000) | 0.0000 (0.0000) |
| FedGCN | 0.0702 (0.0713) | 0.2123 (0.0197) | 0.0549 (0.0091) | 0.1648 (0.0187) | 0.1702 (0.0833) | 0.0833 (0.0584) | 0.0000 (0.0000) | 0.0000 (0.0000) | 0.0006 (0.0006) | 0.0000 (0.0000) | 0.0156 (0.0271) | 0.0097 (0.0169) | 0.0000 (0.0000) | 0.0000 (0.0000) | 0.0085 (0.0148) |
| FedPUB | 0.0000 (0.0000) | 0.0000 (0.0000) | 0.0000 (0.0000) | 0.0012 (0.0021) | 0.0053 (0.0026) | 0.0000 (0.0000) | 0.0000 (0.0000) | 0.0000 (0.0000) | 0.0002 (0.0003) | 0.0000 (0.0000) | 0.0000 (0.0000) | 0.0000 (0.0000) | 0.0000 (0.0000) | 0.0000 (0.0000) | 0.0000 (0.0000) |
| FedNTD | 0.3714 (0.1273) | 0.1895 (0.0098) | 0.0336 (0.0317) | 0.4257 (0.0077) | 0.2438 (0.0476) | 0.2878 (0.0459) | 0.0003 (0.0002) | 0.0256 (0.0256) | 0.0512 (0.0050) | 0.0000 (0.0000) | 0.0019 (0.0009) | 0.0061 (0.0064) | 0.0038 (0.0015) | 0.0008 (0.0008) | 0.0104 (0.0024) |
| FedED | 0.5305 (0.1078) | 0.1080 (0.0158) | 0.0350 (0.0305) | 0.3184 (0.0121) | 0.1534 (0.2660) | 0.1039 (0.1785) | 0.0056 (0.0005) | 0.0097 (0.0097) | 0.0050 (0.0022) | 0.2192 (0.0395) | 0.1796 (0.0634) | 0.0004 (0.0007) | 0.1162 (0.0297) | 0.0005 (0.0005) | 0.0004 (0.0007) |
| **FedLoG** | **0.6472** (0.0811) | **0.4948** (0.0930) | **0.4037** (0.0619) | **0.6142** (0.0292) | **0.5922** (0.1037) | **0.5958** (0.0608) | **0.1070** (0.0623) | **0.1700** (0.1236) | **0.2290** (0.0308) | **0.4795** (0.0949) | **0.5525** (0.1464) | **0.1328** (0.0562) | **0.3580** (0.1256) | **0.2175** (0.0638) | **0.0412** (0.0327) |

**(c) New Client**

| Methods | Cora 3 Clients | Cora 5 Clients | Cora 10 Clients | CiteSeer 3 Clients | CiteSeer 5 Clients | CiteSeer 10 Clients | PubMed 3 Clients | PubMed 5 Clients | PubMed 10 Clients | Amazon Photo 3 Clients | Amazon Photo 5 Clients | Amazon Photo 10 Clients | Amazon Computers 3 Clients | Amazon Computers 5 Clients | Amazon Computers 10 Clients |
|---|---|---|---|---|---|---|---|---|---|---|---|---|---|---|---|
| Local | 0.0995 (0.0084) | 0.1488 (0.0059) | 0.1778 (0.0284) | 0.1435 (0.0113) | 0.1968 (0.0006) | 0.1337 (0.0000) | 0.3570 (0.0000) | 0.3947 (0.0000) | 0.3936 (0.0001) | 0.0313 (0.0001) | 0.0965 (0.0000) | 0.1169 (0.0089) | 0.1571 (0.0000) | 0.1974 (0.0000) | 0.2816 (0.0111) |
| FedAvg | 0.3583 (0.0206) | 0.2713 (0.0057) | 0.3924 (0.1880) | 0.2572 (0.0222) | 0.2859 (0.0348) | 0.2976 (0.0138) | 0.3333 (0.0000) | 0.5243 (0.0000) | 0.5175 (0.0240) | 0.2597 (0.0118) | 0.0853 (0.0000) | 0.1661 (0.0172) | 0.1978 (0.0000) | 0.2924 (0.0637) | 0.4799 (0.0904) |
| FedSAGE+ | 0.2411 (0.0109) | 0.3250 (0.0226) | 0.4129 (0.1052) | 0.3630 (0.0385) | 0.0646 (0.0099) | 0.1048 (0.0322) | 0.3834 (0.0022) | 0.5616 (0.0126) | 0.5279 (0.0000) | 0.2782 (0.0086) | 0.0900 (0.0021) | 0.1472 (0.0549) | 0.2132 (0.3683) | 0.3718 (0.1044) | 0.2756 (0.0062) |
| FedGCN | 0.3449 (0.0494) | 0.3320 (0.0052) | 0.4825 (0.0189) | 0.2572 (0.0072) | 0.4548 (0.1900) | 0.2144 (0.0566) | 0.3333 (0.0000) | 0.4830 (0.0148) | 0.4890 (0.0000) | 0.2597 (0.0550) | 0.0659 (0.0226) | 0.1605 (0.0169) | 0.2066 (0.0781) | 0.3009 (0.0990) | 0.4841 (0.0586) |
| FedPUB | 0.3990 (0.0239) | 0.2258 (0.0153) | 0.4031 (0.0087) | 0.3929 (0.0485) | 0.3408 (0.0113) | 0.3930 (0.0296) | 0.4112 (0.0007) | 0.6036 (0.0067) | 0.5743 (0.0008) | 0.3171 (0.0120) | 0.1075 (0.0102) | 0.2540 (0.0078) | 0.5244 (0.0276) | 0.6764 (0.0289) | 0.5543 (0.0586) |
| FedNTD | 0.3805 (0.0328) | 0.3169 (0.0010) | 0.3705 (0.1879) | 0.4321 (0.0479) | 0.5288 (0.0940) | 0.3057 (0.0166) | 0.4153 (0.0039) | 0.6321 (0.0005) | 0.6026 (0.0106) | 0.4617 (0.1349) | 0.1473 (0.0006) | 0.3081 (0.0361) | 0.0038 (0.0602) | 0.7146 (0.0101) | 0.7873 (0.0232) |
| FedED | 0.4527 (0.0353) | 0.2537 (0.0165) | 0.3194 (0.1364) | 0.3303 (0.1068) | 0.4053 (0.0240) | 0.1346 (0.0136) | 0.4842 (0.0226) | 0.6352 (0.0019) | 0.5969 (0.0131) | 0.5451 (0.0586) | 0.1563 (0.0098) | 0.2622 (0.0027) | 0.1162 (0.0712) | 0.7147 (0.0067) | 0.5228 (0.0610) |
| **FedLoG** | **0.5047** (0.0884) | **0.4439** (0.0455) | **0.6055** (0.0914) | **0.5973** (0.1623) | **0.5647** (0.0179) | **0.6487** (0.1143) | **0.6053** (0.1293) | **0.7091** (0.0557) | **0.7546** (0.0107) | **0.5605** (0.1052) | **0.5083** (0.1794) | **0.5574** (0.0368) | **0.7386** (0.0190) | **0.9164** (0.0071) | **0.8029** (0.0866) |

Figure 3: Impact of headness of class/degree for various scenarios (Amazon Clothing - 3 Clients).

**Q3: Do the headness of degree and class really help other clients?** We evaluate the importance of the headness of degree/class under various scenarios, both of which are expected to enhance data reliability. As FedLoG additionally trains the clients using global synthetic data, we measure the impact by varying the knowledge condensed into global synthetic data. Specifically, we compare four different test settings for constructing global synthetic data, i.e., using **1)** head class & head degree nodes (HH), **2)** head class & tail degree nodes (HT), **3)** tail class & head degree nodes (TH), and **4)** tail class & tail degree nodes (TT). Detailed descriptions are provided in Appendix A.15.

Figure 3 shows test accuracy curves illustrating the impact of each test setting on performance and stability. Data reliability varies with global synthetic data knowledge; HH knowledge is the most reliable. Class headness significantly affects reliability, evident in the performance gap between head and tail classes. Degree headness impacts stability; tail degree settings show more fluctuations. Thus, using HH knowledge is crucial for maintaining reliability and stable outcomes.

**Q4. Does each module effectively address the local overfitting problem?** Figure 4 shows the results of ablation studies under the Missing Class setting: w/o LG denotes excluding the Local Generalization (LG) phase, w/o PG denotes without Prompt Generation, and w/o FS denotes without Feature Scaling. We validate that the Local Generalization phase is crucial for addressing the absent knowledge, while other modules (i.e., PG and FS) impact performance and stability. Refer to Appendix A.11.4 for details.

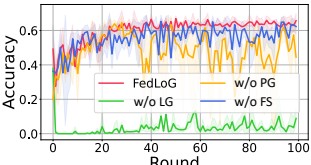

Figure 4: Ablation Studies (CiteSeer - 3 Clients).

Further experiments, including an assessment of Feature Scaling's adaptive effects, hyperparameter analysis, and evaluation on unseen data in an open set, are provided in Appendix A.11. Communication overhead and time complexity analysis are described Appendix A.3 and A.5, respectively.

## 6 PRIVACY ANALYSIS

**Q1. Does utilizing the class distribution of the clients pose a privacy problem?** The class distribution does not include individual data but merely represents the proportion of each class, which is far less sensitive than the raw data. As the class distribution is general statistical information that indicates the trends within a group rather than specific data about individual users, we argue that it is very difficult for an attacker to infer specific data of individual nodes from the class distribution.

However, in case privacy concerns remain, we can add noise to the class rate to make it difficult to determine the exact class distribution. We experimented with two methods of adding noise to the class rate: **1)** adding class-wise Gaussian noise with $\mu$ as 0 and $\sigma$ as $a \times r_k^c$, where $a$ is chosen from $[0.01, 0.1, 0.5]$ and $r_k^c = \frac{|\mathcal{V}_k^c|}{|\mathcal{V}_k|}$ , and **2)** performing random permutation of the elements in the class rate vector. To maintain the trend while applying random permutation, we permuted only the elements within the head classes and within the tail classes.

In Table 3, both Gaussian Noise (GN) and Random Permutation (RP) methods, which result in class rates roughly similar to the original, showed no significant difference in performance, except for the GN ($a = 0.5$) setting that highly deteriorates the trend of the class rate. This indicates that Fed-LoG does not require an exact class distribution as long as the general trend is maintained, allowing us to protect privacy more rigorously.

Table 3: Class rate with noises (Cora - 3 clients).

|  | No Noise | GN ($a = 0.01$) | GN ($a = 0.1$) | GN ($a = 0.5$) | RP |
|---|---|---|---|---|---|
| **Seen Graph** | 0.8601 (0.0118) | 0.8530 (0.0089) | 0.8542 (0.0080) | 0.8560 (0.0010) | 0.8631 (0.0010) |
| **Unseen Node** | 0.7341 (0.0273) | 0.7127 (0.0191) | 0.7217 (0.0318) | 0.7057 (0.0123) | 0.7351 (0.0054) |
| **Missing Class** | 0.6472 (0.0811) | 0.6244 (0.05275) | 0.6032 (0.0457) | 0.6277 (0.0148) | 0.6328 (0.0586) |
| **New Client** | 0.5047 (0.0884) | 0.5278 (0.0326) | 0.5297 (0.0523) | 0.4883 (0.0232) | 0.5199 (0.0421) |

**Q2. Can synthetic data be specified to match the original data's features?** Since the synthetic data is generated by condensing the original nodes within each client's graph, there may be a potential privacy risk. One way to assess this risk is to compare the feature distributions of the original nodes (i.e., $\mathcal{V}_k$) and the synthetic nodes (i.e.,

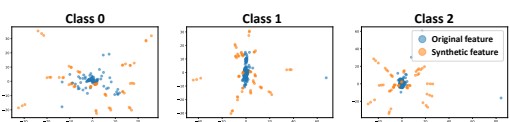

Figure 5: 2D PCA visualization of feature distributions for the same class in the CiteSeer dataset.

$\mathcal{V}_{k,\text{head}}$ and $\mathcal{V}_{k,\text{tail}}$). If these distributions overlap, it suggests that the original features can be reconstructed from the synthetic data, which entails privacy risk. In Figure 5, we present a 2-dimensional PCA visualization of both the original feature matrix (blue) and the synthetic feature matrix (orange) for the same class in the CiteSeer dataset. The clear difference between the two distributions shows that sharing synthetic nodes poses minimal privacy risk.

Moreover, privacy risk is further reduced because the synthetic data represents an aggregation of all training nodes, without being tied to any specific node (i.e., condensation). This aggregation also incorporates structural information into features, resulting in a distribution that diverges from the original feature space. Since the synthetic nodes lack an explicit graph structure, their feature space distills structural information differently from the original nodes, leading to a distinct distribution.

**Q3. How does FedLoG provide protection against gradient inversion attacks?** FedLoG enhances protection against gradient inversion attacks (Zhu et al., 2019), where adversaries attempt to reconstruct the original data from gradients uploaded to the server. This protection is primarily due to each client being trained not only on its local data but also on global synthetic data. The inclusion of this synthetic data introduces noise into the gradients, making it difficult for adversaries to extract information solely from the original data. Additionally, each client applies different levels of feature scaling to the global synthetic data (Section 4.3), and these scaling factors are never shared. This variation further obscures the gradients derived from the synthetic data, making it even harder for adversaries to accurately invert the gradients and reconstruct the original data.

## 7 CONCLUSION

In this study, we address the challenges of local overfitting and unseen data (i.e., unseen node, missing class, and new client) in subgraph-FL with our proposed method, FedLoG. Our model generates global synthetic data by condensing reliable information from each class representation and its structural information across clients, enabling adaptive generalization of absent knowledge within local datasets without directly using data from other clients. This approach enhances the generalization capabilities of local models, allowing them to handle unseen data effectively while also mitigating privacy concerns. Our experimental results demonstrate that FedLoG outperforms existing baselines, proving its efficacy in novel practical scenarios for generalizing to unseen data.

## ACKNOWLEDGEMENTS

This work was supported by Institute of Information & Communications Technology Planning & Evaluation (IITP) and National Research Foundation of Korea (NRF) grant funded by the Korea government (MSIT) (RS-2022-II220157 and RS-2024-00406985) and National Research Foundation of Korea (NRF) funded by Ministry of Science and ICT (NRF-2022M3J6A1063021).

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

# Supplementary Material for
# Subgraph Federated Learning for Local Generalization

# A APPENDIX

## A.1 ALGORITHM

---

**Algorithm 1** FedLoG: The Overall Algorithm

---

 **Server** $\mathcal{S}$
1: **if** round $== 0$ **then**
2:  **Initialize** global model $\phi^{\text{global}}$
3: **else**
4:  **Aggregate** local models: $\phi^{\text{global}} \leftarrow \frac{1}{K} \sum_k \phi_k$
5:  **Generate** global synthetic data $\mathcal{D}^{\text{global}}$:
6:   Combine $\mathcal{V}_{k,\text{head}}$ based on class proportions      $\triangleright$ Equation 4
7: **end if**

 **Local Client** $k$
8: **if** round $== 0$ **then**
9:  **Initialize** local model: $\phi_k \leftarrow \phi^{\text{global}}$
10:  **Local Fitting:** Update $\phi_k, \mathcal{V}_{k,\text{head}}, \mathcal{V}_{k,\text{tail}}$ using $\mathcal{D}_k$
11: **else**
12:  **Update** local model: $\phi_k \leftarrow \phi^{\text{global}}$
13:  **Local Fitting:** Update $\phi_k, \mathcal{V}_{k,\text{head}}, \mathcal{V}_{k,\text{tail}}$ using $\mathcal{D}_k$
14:  **Local Generalization:**
15:   **1. Download** $\mathcal{D}^{\text{global}}, \mathbf{PG}^c \forall c$ from server
16:   **2. Feature Scaling:** Adapt $\mathcal{D}^{\text{global}}$ features locally    $\triangleright$ Equation 5
17:   **3. Prompt Generation:** Generate prompt nodes of $\mathcal{D}^{\text{global}}$ via $\mathbf{PG}^c$
18:   **4. Update** $\phi_k, \mathcal{V}_{k,\text{head}}, \mathcal{V}_{k,\text{tail}}$ using adapted $\mathcal{D}^{\text{global}}$
19: **end if**

 **Pretrain Prompt Generators (PGs)**
20: **Local Client** $k$**:** Pretrain $\mathbf{PG}_k$ using $\mathcal{D}_k$
21: **Server** $\mathcal{S}$**: Aggregate** $\mathbf{PG}^c \leftarrow \frac{1}{\sum_k r_k^c} \sum_k r_k^c \mathbf{PG}_k$ for each class $c$  $\triangleright$ Weighted by $r_k^c = \frac{|\mathcal{V}_k^c|}{|\mathcal{V}_k|}$

---

## A.2 NOTATIONS

Table 4: Summary of the notations. For simplicity, we describe the notation based on a head-branch.

| Notation | Description |
|---|---|
| **General Notations** | |
| $S$ | Server |
| $K$ | Number of clients |
| $\mathcal{G}$ | Graph |
| $\mathcal{V}$ | Set of nodes |
| $\mathcal{E}$ | Set of edges |
| $\mathcal{D}$ | Dataset consisting of $\mathcal{G}$ and $Y$ |
| $Y$ | Label set for the nodes |
| $\boldsymbol{X}_{\mathcal{V}}$ | Feature matrix of a set of nodes $\mathcal{V}$ |
| $\boldsymbol{x}_v$ | Feature vector of a node $v \in \mathcal{V}$ |
| $\phi$ | Set of parameters of a global model |
| $|\mathcal{C}_{\mathcal{V}}|$ | Number of classes within a set of nodes $\mathcal{V}$ |
| $r$ | Current round |
| $\boldsymbol{h}_v$ | Representation of a node $v$ |
| $\alpha$ | Weight of prediction between head and tail branches |
| **Local Client $k$ Notations** | |
| $\mathcal{G}_k$ | Local graph for client $k$ |
| $\mathcal{V}_k$ | Set of nodes within a local graph for client $k$ |
| $\mathcal{E}_k$ | Set of edges within a local graph for client $k$ |
| $\mathcal{D}_k$ | Local dataset for client $k$ |
| $\mathcal{D}^{\text{local}}$ | Combined local datasets, $\mathcal{D}^{\text{local}} = \bigcup_{k=1}^{K} \mathcal{D}_k$ |
| $Y_k$ | Label set for the nodes within a local graph for client $k$ |
| $\phi_k$ | Set of parameters of a local model for client $k$ |
| $\varphi_k$ | Parameters of a GNN embedder for client $k$ |
| $\theta_{k,H}$ | Parameters of a head-branch classifier for client $k$ |
| $\theta_{k,T}$ | Parameters of a tail-branch classifier for client $k$ |
| $\mathcal{V}_{k,\text{head}}$ | Set of synthetic nodes within a head-degree branch of client $k$ |
| $v_{k,\text{head}}^{(c,s)}$ | $s$-th synthetic node for class $c$ in a head-branch for client $k$ |
| $\boldsymbol{X}_{\mathcal{V}_{k,\text{head}}}$ | Feature matrix of a set of nodes $\mathcal{V}_{k,\text{head}}$ |
| $\boldsymbol{x}_{k,\text{head}}^{(c,s)}$ | Feature of the $s$-th synthetic node for class $c$ in a head-branch for client $k$ |
| $\boldsymbol{h}_{k,\text{head}}^{(c,s)}$ | Representation of the $s$-th synthetic node for class $c$ in a head-branch for client $k$ |
| $\mathbb{P}_{k,\text{head}}$ | Set of prototype representations (i.e., representations of synthetic nodes) in a head-branch for client $k$ |
| $\bar{\boldsymbol{h}}_{\mathcal{V}_{k,\text{head}}^c}$ | Average of prototype representations of class $c$ in a head-branch for client $k$ |
| $r_k^c$ | Proportion of nodes labeled $c$ in client $k$'s dataset $\mathcal{D}_k$ |
| $\textbf{PG}_k$ | Pretrained prompt generator for client $k$ (regardless of specific class) |
| $\boldsymbol{\gamma}_k$ | Class-wise adaptive factor for client $k$ |
| $\mathcal{G}_k^{\text{syn}}$ | Synthetic graph set consisting of graphs, each containing the global synthetic nodes (which are adapted locally) neighboring with their generated prompt nodes. |
| $v_k^p$ | Generated prompt node of node $v_k$ |
| $\hat{\boldsymbol{x}}_{v_k^p}$ | Generated feature of generated prompt node $v_k^p$ for the input feature $\boldsymbol{x}_{v_k}$ of node $v_k$ |
| $\bar{\boldsymbol{x}}_{\mathcal{N}_{v_k}}$ | Average of features of the $h$-hop neighbors of node $v_k$ within $\mathcal{G}_k$ |
| **Global Synthetic Node Notations** | |
| $\mathcal{D}^{\text{global}}$ | Global synthetic dataset |
| $\mathcal{G}^{\text{global}}$ | Global synthetic graph |
| $\mathcal{V}^{\text{global}}$ | Set of global synthetic nodes |
| $Y^{\text{global}}$ | Label set for the global synthetic nodes |
| $\boldsymbol{X}_{\mathcal{V}^{\text{global}}}$ | Feature matrix of the global synthetic nodes $\mathcal{V}^{\text{global}}$ |
| $\boldsymbol{x}_{v_g^{(c,s)}}$ | Feature of the $s$-th global synthetic node $v_g$ for class $c$ |
| $\textbf{PG}^c$ | Class-specific prompt generator for class $c$ (by aggregating $\textbf{NG}_k$ for all $k$ in a class-wise manner) |
| **Hyperparameter Notations** | |
| $s$ | Hyperparameter for assigning the number of synthetic nodes per class |
| $\lambda$ | Hyperparameter for adjusting tail degree threshold |
| $\beta$ | Hyperparameter for adjusting the extent of regularization of the features of synthetic nodes |

A.3 COMMUNICATION OVERHEAD

In this section, we provide comparisons of communication overhead across different baselines. FedLoG uploads and downloads both the synthetic data and the model parameters at the end of each round. For the Cora dataset with a setting of 3 clients and $s = 20$, our model has 1,081,926 parameters to share with the server, resulting in 4 bytes×1,081,926=4.32 MB (excluding the prompt generator, which is only trained once at the first round). Additionally, the synthetic data has 182,000 parameters ($s \times |\mathcal{C}_\mathcal{V}| \times d$, where $|\mathcal{C}_\mathcal{V}|$ denotes the number of classes and $d$ denotes the dimension of the features), amounting to 0.72 MB. In summary, our model requires 10.08 MB ($2 \times (4.32 + 0.72)$) for upload and download each round. Below are comparisons of communication overhead between models over 100 rounds:

|  | FedAvg | FedSAGE+ | FedGCN | FedPUB | FedNTD | FedED | FedLoG |
|---|---|---|---|---|---|---|---|
| **Cost (MB)** | 393.11 | 1543.58 | 393.11 | 786.03 | 393.11 | 393.11 | 1011.14 |

Table 5: Comparison of Communication Cost (MB) Across Different Models

Although FedLoG relatively requires higher communication overhead compared to other baselines, it shows faster convergence due to its utilization of reliable class representation, leading to a stable training process. Below are comparisons of communication overhead until each model reaches the same accuracy (i.e., 0.8 on the Cora dataset with 3 clients).

| Model | FedAvg | FedSAGE+ | FedGCN | FedPUB | FedNTD | FedED | FedLoG |
|---|---|---|---|---|---|---|---|
| **Rounds to Reach 0.8** | 58 | 100 (Fails to reach) | 57 | 29 | 19 | 39 | 10 |
| **Cost (MB)** | 228.00 | 1543.58 | 224.07 | 227.95 | 72.79 | 149.41 | 101.11 |

Table 6: Rounds to reach 0.8 accuracy and corresponding communication cost in MB across different models.

Despite FedLoG's higher communication overhead per round, its faster convergence results in a lower overall communication overhead to achieve the same accuracy compared to other baselines. This demonstrates the efficiency and stability of FedLoG's training process, making it an effective approach despite the initially higher communication cost per round.

A.4 DETAILED PROCESS OF THE CLASSIFIERS $\theta_{k,H}$ AND $\theta_{k,T}$

In this section, we provide the detailed process of the classifiers $\theta_{k,H}$ and $\theta_{k,T}$. Specifically, we describe Eq. 2:

$$\boldsymbol{h}'_{v_k} = \theta_{k,H}\left(\boldsymbol{h}_{v_k}, \left\{\left(\boldsymbol{h}_{v_k} - \boldsymbol{h}_{v^{(1,1)}_{k,\text{head}}}\right), \ldots, \left(\boldsymbol{h}_{v_k} - \boldsymbol{h}_{v^{(|\mathcal{C}_\mathcal{V}|,s)}_{k,\text{head}}}\right)\right\}\right),$$

which corresponds to the head-degree classifier in the local model. Since both the head-degree and tail-degree classifiers have the same architecture, we focus on describing the head-degree classifier.

The primary objective of the classifier is to ensure that all prototypes in $\mathbb{P}_{k,\text{head}}$ contribute to the final prediction of the target node, allowing the prediction loss to be influenced by all prototypes. To achieve this, we first generate a message $\boldsymbol{m}_{kj}$ from each prototype node $v_j \in \mathbb{P}_{k,\text{head}}$ to the target node $v_k$, based on the distance $d_{kj}$ between them. This is computed using $\text{MLP}_{\text{msg}}$ as follows:

$$\boldsymbol{m}_{kj} = \text{MLP}_{\text{msg}}([\boldsymbol{h}_{v_k} \parallel \boldsymbol{n}_{v_k} \parallel d_{kj}]), \tag{6}$$

where the distance $d_{kj}$ is calculated as:

$$d_{kj} = \|\Delta \boldsymbol{r}_{kj}\|^2, \quad \Delta \boldsymbol{r}_{kj} = [\boldsymbol{h}_{v_k} \parallel \boldsymbol{n}_{v_k}] - [\boldsymbol{h}_{v_j} \parallel \boldsymbol{h}_{v_j}], \tag{7}$$

and $\parallel$ denotes concatenation. Here, $\boldsymbol{n}_{v_k} = \frac{1}{|\mathcal{N}_{v_k}|}\sum_{v_o \in \mathcal{N}_{v_k}} \boldsymbol{h}_{v_o}$ is the average embedding of the 1-hop neighbors of the target node $v_k$ within the graph $\mathcal{G}_k$.

The neighbor information is used because the embeddings of neighbors from head-degree and tail-degree nodes differ, enabling each branch to leverage degree-specific knowledge. The target node

embedding $\boldsymbol{h}_{v_k}$ is then updated by applying a learned transformation to the representation differences $\Delta \boldsymbol{r}_{kj}$, followed by aggregation:

$$\boldsymbol{t}_{kj} = [\boldsymbol{h}_{v_k} - \boldsymbol{h}_{v_j}] \odot \mathrm{MLP}_{\mathrm{trans}}(\boldsymbol{m}_{kj}), \tag{8}$$

$$\boldsymbol{t}'_k = \frac{1}{|\mathbb{P}_{k,\mathrm{head}}|} \sum_j \boldsymbol{t}_{kj}, \tag{9}$$

$$\boldsymbol{h}'_{v_k} = \boldsymbol{h}_{v_k} + \boldsymbol{t}'_k, \tag{10}$$

where $\mathrm{MLP}_{\mathrm{trans}}$ transforms the message $\boldsymbol{m}_{kj}$ into a scalar value.

In summary, the classifiers $\theta_{k,H}$ and $\theta_{k,T}$ update the embedding of the target node $\boldsymbol{h}_k$ by reflecting the interactions between all different prototypes in $\mathbb{P}_{k,\mathrm{head}}$, ensuring that the final prediction and its loss are influenced by all prototypes (i.e., learnable synthetic nodes).

## A.5 TIME COMPLEXITIES

In this section, we assess the time complexity of FedLoG and demonstrate its efficiency in computational requirements. Notably, each classifier, including the head and tail branches, shares the same time complexity as a multi-layer perceptron (MLP), specifically $O(d^2)$, where $d$ represents the feature dimension of its input. This ensures that the branches have minimal computational overhead, even when processing high-dimensional features. We provide detailed time complexity calculations for each module as follows:

**Classifier.** *Pairwise Distance between Prototypes (Eq. 7).* The naive time complexity is $O(PF)$, where $P$ is the number of prototypes (i.e., $|C_v| \times s$) and $F$ is the dimension of the input ($2 \times d$, where $d$ denotes the dimension of its inputs). Since $P$ is small enough to be negligible, the complexity reduces to $O(d)$.

*Distance-Based Message Generation (Eq. 6).* The naive time complexity is $O(PF^2)$, where $F$ is the input dimension (i.e., $2 \times d + 1$). With $P$ being negligible, this results in a complexity of $O(d^2)$, which is the same as that of an MLP.

*Updating the Target Node's Representation (Eqs. 8- 10).* Eq. 8 includes an MLP and elementwise operations with subtraction, giving a total complexity of $O(d^2)$. Eqs. 9 and 10 involve only simple additions and are therefore negligible. Thus, the total time complexity for this update is $O(d^2)$.

**Prompt Generator.** In the pretraining phase, the prompt generator requires $O(|\mathcal{V}|d)$ complexity for Eq. 11, and $O(|\mathcal{E}|d + |\mathcal{V}|d^2)$ for Eq. 12. Therefore, the total time complexity of the prompt generator is $O(|\mathcal{E}|d + |\mathcal{V}|d^2)$, which is the same as the GNN encoder. However, it is worth noting that, during inference, it requires only $O(d^2)$, which has the same time complexity as the MLP.

**Feature Scaling.** For feature scaling in Eq. 5, the time complexity is $O(d)$ since the operation involves only simple element-wise additions.

Consequently, the total time complexity of the classifiers, including the Prompt Generator and Feature Scaling, is $O(d^2)$. This is significantly lighter than the complexity of the GNN encoder, $O(|\mathcal{E}|d + |\mathcal{V}|d^2)$, where $|\mathcal{E}|$ and $|\mathcal{V}|$ denote the number of edges and nodes, respectively.

**Graph Encoder.** As we utilize GraphSAGE for the GNN encoder, it requires $O(|\mathcal{E}|d + |\mathcal{V}|d^2)$ for both forward and backward passes.

**Overall Model Complexity.** To sum up, our model requires $O(|\mathcal{E}|d + |\mathcal{V}|d^2)$ complexity, which is the same as a GNN encoder. Importantly, each classifier has the same time complexity as an MLP (i.e., $O(d^2)$), which has little influence on the total complexity of our architecture. This highlights the efficiency of FedLoG in handling computational demands, even for large-scale graphs with high-dimensional data.

## A.6 RELATED WORKS

### A.6.1 IMPROVING GENERALIZATION IN FEDERATED LEARNING

One of the core challenges in Federated Learning (FL) is achieving strong generalization across heterogeneous and biased client datasets. Clients often have non-i.i.d. data distributions or class

imbalances, which make it difficult to train robust and generalized models. To address these challenges, various approaches (Chen & Chao, 2021; Li et al., 2023b;a; Ye et al., 2023) have been proposed, focusing on improving the generalization capabilities of both global and local models.

FedRoD (Chen & Chao, 2021) bridges the gap between generic FL and personalized FL by leveraging a class-balanced loss and empirical risk minimization. While this approach improves generic FL, it depends on the presence of at least one data point for each class within each client. This reliance makes it less effective in scenarios where certain classes are entirely absent in some clients, a common challenge in federated learning (i.e., the Missing Class setting).

FedETF (Li et al., 2023b) addresses classifier biases by enhancing the generalization of the global model and enabling personalized adaptation through local fine-tuning. To improve generalization, FedETF employs a balanced feature loss weighted by the number of samples in each class. However, its generalization phase does not adequately handle the Missing Class scenario, where certain classes have no samples at all. Furthermore, its reliance on local fine-tuning exacerbates the local overfitting problem, making local models more prone to overfitting their biased data distributions and struggling to generalize to unseen data, such as missing classes.

FedLAW (Li et al., 2023a) enhances the generalization of global models by introducing a learnable weighted aggregation mechanism, where the L1 norm of the aggregation weights is constrained to be less than 1. Additionally, it incorporates the concept of client coherence to identify clients that positively contribute to generalization. Similarly, FedDisco (Ye et al., 2023) proposes a weighted aggregation method based on the discrepancy between local and global category distributions, further improving the performance of the global model. While both FedLAW and FedDisco primarily focus on enhancing the generalization of the global model, our work takes a different approach by addressing the local overfitting problem. Specifically, we aim to improve the generalization of local models, which are prone to overfitting their local data distributions after a few local updates from the global model, even when the global model itself is well generalized.

### A.6.2 Synthetic-based Federated Learning

Generating synthetic data using aggregated knowledge from clients has emerged as a promising approach to compensate for the limitations of local training data. This method facilitates data augmentation while addressing challenges such as class imbalance and limited data availability.

MixUp-based synthetic data generation methods (Yoon et al., 2021; You et al., 2024; Oh et al., 2020; Shin et al., 2020) augment training datasets by mixing data samples with privacy-preserving techniques. However, these approaches operate in the raw feature space, which poses significant risks of privacy leakage, particularly when the local data size is small.

Alternatively, GAN-based (Goodfellow et al., 2020) methods, such as FedGAN (Rasouli et al., 2020) and FedDPGAN (Zhang et al., 2021b), leverage generative models to create synthetic data. These methods aim to generalize a global generator to produce synthetic data that can mitigate data imbalance while preserving privacy. However, they incur high computational costs, which limits their practical applicability.

Recently, condensation-based methods (Kim & Choi, 2022; Liu et al., 2022; Wang et al., 2024) have been proposed to alleviate the impact of data heterogeneity. FedDC (Kim & Choi, 2022) condenses synthetic data based on local data and fine-tunes the global model at the server level to ensure stable convergence. FedMK (Liu et al., 2022) generates synthetic data by condensing private data into meta-knowledge, which is used as an additional training set to accelerate convergence. FedAF (Wang et al., 2024) introduces an aggregation-free paradigm, where the server directly trains the global model using condensed synthetic data.

Key distinctions of our approach compared to existing synthetic-based methods are as follows:

**No reliance on raw features.** MixUp-based (Yoon et al., 2021; You et al., 2024; Oh et al., 2020; Shin et al., 2020) and condensation-based methods (Kim & Choi, 2022; Liu et al., 2022; Wang et al., 2024) generate synthetic data by augmenting or condensing data at the raw feature level of the input, which can lead to privacy leakage, particularly when the original data is limited. In contrast, our synthetic data has a distinct feature distribution from the original data, arising from differences in the embedding approach used for synthetic and original data, particularly due to the presence or absence of the explicit structure. Furthermore, our method leverages not only the original data but also global synthetic data as an additional training set for condensation. This design significantly

reduces privacy risks, especially when the local data size is small. Moreover, we only share a subset of the synthetic data (i.e., synthetic data within the head-degree branch), which not only excludes complete information about the local data to enhance privacy but is also specifically designed to capture reliable information relevant to the graph domain.

**Handling the local overfitting problem.** Our method effectively addresses the local overfitting problem, which is one of the most challenging issues in federated learning. Local overfitting occurs after a few local updates with the distributed global model, causing the local model to severely struggle in predicting unseen data that involves unseen distributions, particularly for missing classes. MixUp-based approaches (Yoon et al., 2021; You et al., 2024; Oh et al., 2020; Shin et al., 2020) still depend on local data for augmenting the training set, which limits their ability to generate data for missing classes. In contrast, our method generates global synthetic data even in scenarios where local data for certain classes is completely absent. This is achieved without relying on raw feature-based MixUp, ensuring both privacy and flexibility.

**Optimizing training with synthetic data.** We extend beyond the generation of synthetic data by investigating how to train it effectively. Since our synthetic data has a different feature distribution from the original data but is utilized as training data (i.e., local generalization), it is essential to explore how to optimize the model training process with global synthetic data. To address this, we propose the *Feature Scaling* and *Prompt Generator* phases, as detailed in Section 4.3, to minimize the training-effect gap between original nodes and synthetic nodes.

## A.7 DETAILED PROCESS OF PRETRAINING THE PROMPT GENERATOR

In this section, we explain the process of pretraining local prompt generators and how they are aggregated on the server to produce unbiased prompt nodes for each class. Specifically, the primary goal of the prompt generator is to ensure that the synthetic graph—comprising a target node and its corresponding prompt node derived from the target node's features—produces a similar gradient matrix as when the target node is predicted using its true $h$-hop subgraph within the local graph.

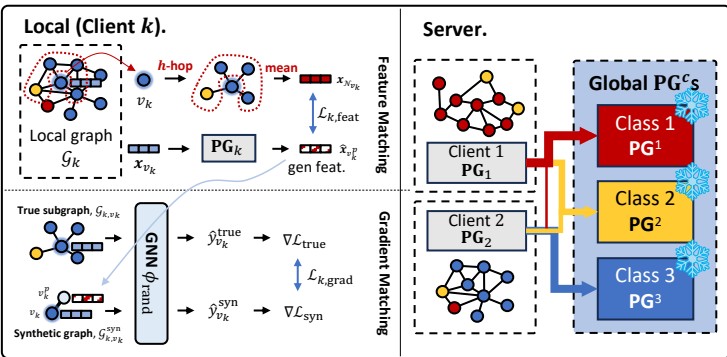

Figure 6: Overview of Pretraining the Prompt Generator.

**Training Local Prompt Generators.** Each client $k$ trains its own prompt generator $\mathbf{PG}_k$, aiming to generate a synthetic prompt node that optimizes the GNN's training effectiveness on feature-only data (Figure 6(left)). Let $\mathcal{G}_{k,v_k}^{\text{syn}}$ be the generated synthetic graph consisting of the target node $v_k$ and its generated prompt node $v_k^p$. The objective is to approximate the true $h$-hop subgraph around $v_k$ within the local graph $\mathcal{G}_k$ (denoted as $\mathcal{G}_{k,v_k} = (\mathcal{V}_{k,v_k}, \mathcal{E}_{k,v_k}) \subseteq \mathcal{G}_k$) in a compact form within $\mathcal{G}_{k,v_k}^{\text{syn}}$ (Jin et al., 2021; 2022).

To achieve this, the prompt generator applies feature matching to ensure that the generated prompt node has features similar to those of the true neighbors of the target node. In essence, the synthetic prompt node acts as a "compressed summary" of the surrounding structural information derived from the target node's features. The input to the generator $\mathbf{PG}_k$ is the feature vector of the target node $\boldsymbol{x}_{v_k}$, and it outputs a synthetic prompt feature $\hat{\boldsymbol{x}}_{v_k^p} = \mathbf{PG}_k(\boldsymbol{x}_{v_k}) \in \mathbb{R}^d$. To align the synthetic prompt node with the average features of real $h$-hop neighbors within the local graph, we minimize the following

feature-matching loss:

$$\mathcal{L}_{k,\text{feat}} = \frac{1}{|\mathcal{V}_k|} \sum_{v_k \in \mathcal{V}_k} \|\hat{\boldsymbol{x}}_{v_k^p} - \bar{\boldsymbol{x}}_{\mathcal{N}_{v_k}}\|_2^2, \text{ where } \bar{\boldsymbol{x}}_{\mathcal{N}_{v_k}} = \frac{1}{|\mathcal{V}_{k,v_k}| - 1} \sum_{v \in \mathcal{V}_{k,v_k} \backslash v_k} \boldsymbol{x}_v. \quad (11)$$

To ensure that the training effect on the synthetic graph $\mathcal{G}_{k,v_k}^{\text{syn}}$ resembles that on the true $h$-hop graph $\mathcal{G}_{k,v_k}$, gradient matching is applied. This approach minimizes the difference between gradients of the GNN when trained on the true $h$-hop graph versus the synthetic graph, aligning the parameter updates and thus making the learning process similar. The gradient-matching loss is defined as follows:

$$\mathcal{L}_{k,\text{grad}} = \frac{1}{N} \sum_{n=1}^{N} \left\| \nabla_{\phi_{\text{rand}}^{(n)}} l(\phi_{\text{rand}}^{(n)}; \mathcal{G}_{k,v_k}^{\text{syn}}(v_k), y_{v_k}) - \nabla_{\phi_{\text{rand}}^{(n)}} l(\phi_{\text{rand}}^{(n)}; \mathcal{G}_{k,v_k}(v_k), y_{v_k}) \right\|_2^2, \quad (12)$$

where $N$ is the number of randomly initialized weights $\phi_{\text{rand}}^{(n)}$ (for $n = 1, 2, \ldots, N$) used to optimize for the target-node classification task.

The combined loss for optimizing the local prompt generator is:

$$\mathcal{L}_{\mathbf{PG}_k} = \mathcal{L}_{k,\text{feat}} + \mathcal{L}_{k,\text{grad}}. \quad (13)$$

Thus, the generator produces a single synthetic prompt node feature $\hat{\boldsymbol{x}}_{v_k^p} \in \mathbb{R}^d$, which serves two main purposes: (1) it captures essential structural information from the target node's features, and (2) it enhances the GNN's learning effect on feature-only data.

We pretrain $\mathbf{PG}_k$ for all $k \in \{1, \ldots, K\}$ over $P$ (i.e., 100) epochs using the training sets within each local dataset, resulting in a collection $\mathbb{PG} = \{\mathbf{PG}_1, \ldots, \mathbf{PG}_K\}$. We set $N$ to 20.

## A.8 DETAILED PROCESS OF EVALUATING DATA RELIABILITY

In this section, we detail the process of evaluating data reliability as outlined in Section 3.1. We define 'data reliability' as the accuracy and consistency of information from decentralized nodes. Specifically, we assess which data within the local dataset positively or negatively impacts other clients in the FL framework.

Inspired by the robust performance of GNNs for head degree and head degree data (Yun et al., 2022; Park et al., 2021; Zhao et al., 2021; Liu et al., 2021), we design experiments to evaluate data reliability from two perspectives: 1) headness of degree and 2) headness of class. We use the PubMed dataset for validation.

We set the base settings for both perspectives. In the FL framework, we assign two roles to each client. The 'receiver' is the client who receives information about the target class from other clients. This client is trained using the same training data across all settings for this section, ensuring a fair comparison to validate the impact from other clients. 'Contributors' are the clients who share knowledge from their own data with the 'receiver'. Their training sets (i.e., information shared through the FL framework) vary for each setting, such as adjusting the proportion of head/tail degree nodes or class imbalance rate. In a global setting with $K$ clients in FL, we assign one client as the 'receiver' and the others as 'contributors' (i.e., $K - 1$ clients).

To assess how degree or class headness affects data reliability, we measure the target class accuracy of the 'receiver' when varying the training sets of 'contributors'. This helps identify whether headness or tailness of data positively or negatively impacts the 'receiver'. We construct the global model by averaging the weights from each client and then evaluate the global model on the 'receiver's' local graph following FedAvg (McMahan et al., 2017).

**Detailed Process for Headness of Degree Perspective**  We divide head degree and tail degree using the tail degree threshold $\lambda$ set to 3, as justified in Appendix A.9. Nodes with degrees less than or equal to 3 are considered tail degree nodes, while those with degrees greater than 3 are head degree nodes. We only vary the training dataset of the 'contributors'. We create three different training sets for each 'contributor': 1) Head degree nodes only (Head degree), 2) Tail degree nodes only (Tail degree), and 3) Balanced degree nodes (Balanced degree). Each training set contains the same number of nodes, but their headness differs according to the setting. The 'Head degree' setting includes only

head degree nodes with the target class, the 'Tail degree' setting includes only tail degree nodes, and the 'Balanced degree' setting includes an equal mix of head and tail degree nodes.

We use the FedAvg (McMahan et al., 2017) framework for the federated learning setting with 100 rounds. At the final round, we evaluate the accuracy of the target class within the 'receiver's' local data using the global model. We average the performances across all classes and report the mean of three seeds results.

**Impact of Class Headness on the Data Reliability**   We define the 'imbalance rate' using the proportion of the number of the target class within each 'contributor's local data. We fix the training nodes of the target class for each 'contributor', and varies the number of training nodes for other classes which are not the target class. Let $n_c$ be the number of nodes per class, and let the number of training nodes of the target class be $n_t$. We then assign $n_k$ number of training nodes for each non-target class:

$$n_k = n_t + \frac{r_{\text{imb}}}{10} \times \min(n_c \forall c \in C) \tag{14}$$

where $\min(n_c \forall c \in C)$ is the minimum number of nodes across all other classes $c$ within the set $C$, and $r_{\text{imb}}$ is the imbalance rate can be defined as:

$$r_{\text{imb}} = 10 \times \frac{n_k - n_t}{\min(n_c \forall c \in C)}.$$

Thus, if the $r_{\text{imb}}$ has a negative value (i.e., $n_t > n_k$), it means the target class becomes a head class within the local data. Conversely, when the $r_{\text{imb}}$ has a positive value, the target class becomes a tail class. As the value of $r_{\text{imb}}$ increases, the tailness of the target class gets higher. We set $r_{\text{imb}}$ in the range from -5 to +5, and we average the performances of the 'receiver' at the final round across all classes and report the mean of three seed results.

## A.9   Criteria for Threshold Degree Value for Tail-Degree Nodes

Recent methods (Yun et al., 2022; Liu et al., 2021) addressing the degree long-tail problem consider nodes with degrees less than or equal to 5 as tail degree nodes, while those with degrees greater than 5 are considered head degree nodes.

As shown in Figure 7, we illustrate the number of nodes belonging to 1) head class & head degree (HH), 2) head class & tail degree (HT), 3) tail class & head degree (TH), and 4) tail class & tail degree (TT) as we vary the threshold value $\lambda$ within the global graph. We use the Cora dataset for validation.

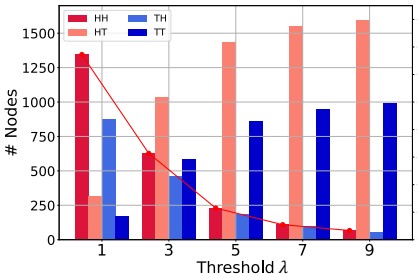

Figure 7: The number of nodes for HH/HT/TH/TT at threshold $\lambda$ (Cora dataset used).

When the threshold $\lambda$ increases, the number of HH nodes significantly decreases, reducing the amount of knowledge that can be condensed into the global synthetic data. In this work, we set $\lambda$ to 3 to utilize a sufficient amount of HH knowledge while filtering out noisy information from tail degree nodes.

## A.10   Detailed Process of Evaluating Unseen Data

In this section, we provide a detailed description of our proposed 'Unseen Data' test settings (i.e., 'Unseen Node', 'Missing Class', and 'New Client'). To evaluate realistic scenarios, we define two different settings for evaluating unseen data: **1)** Closed set nodes setting (Closed set) and **2)** Open set nodes setting (Open set). The results in Table 2 are evaluated on the closed set nodes setting.

### A.10.1   Closed Set

Following recent work (Baek et al., 2023), we partition the global graph into several subgraphs using the Metis graph partitioning algorithm (Karypis & Kumar, 1997). For the 'New Client' setting, we generate an additional subgraph, resulting in the partitioning of the global graph into $k + 1$ subgraphs, where $k$ denotes the number of clients. Due to the properties of the Metis algorithm, the extra subgraph has a distinct label distribution, as the algorithm minimizes the number of edges between partitions, leading to the formation of distinct communities.

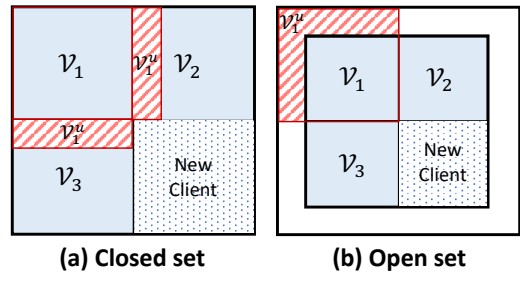

Figure 8: Overview of Unseen Data settings ($K = 3$).

The closed set setting includes unseen data for the 'Unseen Node' and 'Missing Class' settings from other clients. Specifically, in Figure 8(a), the global set of nodes is $\mathcal{V} = \bigcup_{k=1}^{K} \mathcal{V}_k$, with $\mathcal{V}_i \cap \mathcal{V}_j = \varnothing$ for all $i \neq j$. We construct the 'Unseen Node' and 'Missing Class' nodes for client $k$ by expanding the $h$-hop subgraph from the local graph $\mathcal{G}_k$. Since we allocate all nodes within the global node set $\mathcal{V}$ to the clients, the nodes within the $h$-hop subgraph (i.e., $\mathcal{V}_k^u$) inevitably overlap with those of other clients. Although nodes may overlap, no edges are shared between different clients. Unseen nodes from other clients establish new connections with the local data.

For the 'Missing Class' setting, we select the missing classes for each client and then exclude the nodes corresponding to those classes (i.e., $\mathcal{V}_k^{uc}$) within each local graph $\mathcal{G}_k$. To maintain the overall context of the local graph, we select the missing class from tail classes, which have the smallest portion within each local graph. If the number of nodes corresponding to the missing classes is insufficient, we add additional missing classes for those clients. Excluded nodes $\mathcal{V}_k^{uc}$ are included in $\mathcal{V}_k^u$.

When evaluating the 'Missing Class' at test time, we expand the local graph $\mathcal{G}_k$ to the range of $h$-hop, and within the evolved graph structure, the local model predicts the labels of nodes in $\mathcal{V}_k^{uc}$. For 'Unseen Node', the local model predicts the labels of nodes in $\mathcal{V}_k^u \setminus \mathcal{V}_k^{uc}$.

For real-world case for the closed set setting, consider Store-A, which uses a model tailored to the purchasing habits of its regular customers. This model may struggle to adapt to the distinct buying patterns of customers from Store-B. These new patterns could create unfamiliar 'also-bought' connections between products within Store-A, especially if they involve new products that Store-A has never sold before. However, these customers can visit Store-A at any time, forming new relationships with existing nodes, reflecting a real-world scenario. This complexity increases the difficulty in effectively integrating and addressing new nodes in the model. In addition, we provide the data statistics for each setting in Appendix A.16.

### A.10.2 OPEN SET

In real-world scenarios, unseen data outside the global nodes $\mathcal{V}$ in the FL system can emerge and form new relationships with existing nodes. We define this setting as Open Set, where the unseen nodes are $\mathcal{V}_k^u \cap \mathcal{V} = \varnothing$. To create this setting, we randomly crop 20% of the global graph before partitioning it into $k + 1$ subgraphs, denoting the cropped node set as $\mathcal{V}_{\text{crop}}$.

Similar to the Closed Set, we exclude nodes corresponding to locally assigned missing classes within each local graph. At test time, for the 'Unseen Node' and 'Missing Class' settings, we reconstruct the structure between cropped nodes $\mathcal{V}_{\text{crop}}$ and local nodes $\mathcal{V}_k$. Within the reconstructed graph, we evaluate the nodes in $\mathcal{V}_{\text{crop}}$ that belong to the missing classes for the 'Missing Class' setting and those having locally trained classes for the 'Unseen Node' setting.

In Table 9, we provide the experimental results on the open set in Appendix A.11.3.

### A.11 ADDITIONAL EXPERIMENTS

### A.11.1 IMPACT OF THE HYPERPARAMETERS

In Figure 9, we analyze the impact of hyperparameters such as the number of synthetic data for each class ($s$) and the tail degree threshold ($\lambda$).

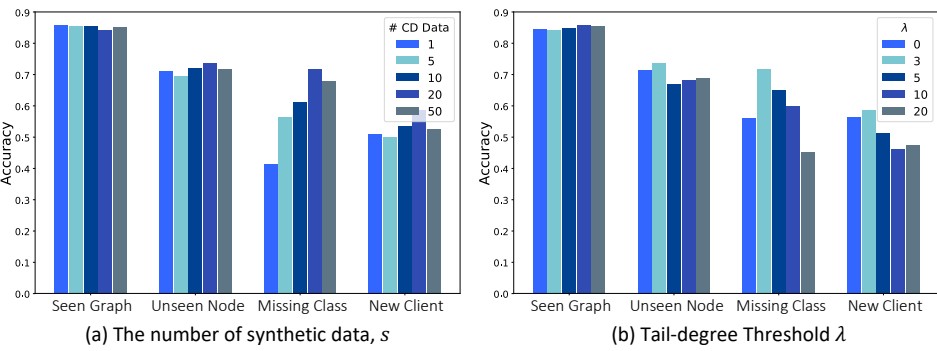

Figure 9: Hyperparameter analysis.

**The Number of Synthetic Data,** $s$    For generating the global synthetic data, sets of learnable nodes $\mathcal{V}_{k,\text{head}}$ and $\mathcal{V}_{k,\text{tail}}$ are constructed during the Local Fitting phase within each client. We assign $s$ learnable synthetic nodes per class and vary $s$ to assess its impact on global synthetic nodes.

As shown in Figure 9(a), we vary $s$ within the range $[1, 5, 10, 20, 50]$ and evaluate the model's performance on the same test data using the Cora dataset with 3 clients. Notably, $s$ significantly impacts the 'Unseen Data' settings, particularly the 'Missing Class' setting, which relies heavily on global synthetic data. A larger number of synthetic data condenses diverse knowledge expressions. However, too many synthetic data points complicate modeling the interaction between the target node and each synthetic nodes (i.e., prototypes), as all prototypes participate in the final prediction described in Section 4.1. Consequently, accuracy for 'Unseen Data'—including 'Unseen Node', 'Missing Class', and 'New Client'—improves with more synthetic data, but an excessive number (e.g., $s = 50$) can reduce performance. Conversely, the performance of the 'Seen Graph' settings shows robustness to the number of synthetic data compared to the 'Unseen Data' settings because the dependency on knowledge from other clients is lower for test data with the same distribution as the training data.

**Tail-Degree Threshold,** $\lambda$    We evaluate the impact of the tail-degree threshold $\lambda$ on performance. Varying $\lambda$ within the range $[0, 3, 5, 10, 20]$, we use the CiteSeer dataset with 3 clients for the evaluation. As shown in Figure 9(b), the tail-degree threshold $\lambda$ significantly impacts the 'Unseen Data' settings as it directly influences the knowledge condensed into the global synthetic data. Increasing $\lambda$ filters out more knowledge from tail-degree nodes, condensing primarily head-degree node knowledge. However, as illustrated in Figure 7 in Section A.9, the number of HH nodes significantly decreases with a higher $\lambda$, reducing the amount of knowledge to be condensed into the global synthetic data. Thus, setting $\lambda$ to 3 yields the best performance, effectively filtering out tail-degree knowledge while ensuring a sufficient amount of HH nodes.

**Degree-Based Branch Weight for Prediction,** $\alpha$    The primary objective of each branch is to distill knowledge from the input data into learnable synthetic data. To achieve this, we designed a prototypical network-based branch that uses learnable synthetic data to represent class-specific knowledge. By adjusting the weight (i.e., alpha) of each branch's final prediction based on the target node's degree, we guide the gradient flow from head degree nodes primarily towards the head branch. This approach enables head degree knowledge to be distilled within the head branch's synthetic data, and similarly, tail-degree knowledge within the tail branch.

As shown in Section 3.1, tail-degree knowledge negatively affects the performance of other clients. This effect is illustrated in Figure 3, where we see that synthetic data generated from head-branch knowledge (i.e., HH: head class/head degree and TH: tail class/head degree) outperforms that generated from tail-branch knowledge (i.e., HT: head class/tail degree and TT: tail class/tail degree). Specifically, the performance hierarchy (HH > HT and TH > TT) suggests that the head branch holds more reliable knowledge from the input data, particularly head degree knowledge, as discussed in Section 3.1. This demonstrates that each branch effectively captures distinct types of knowledge, successfully separating head and tail degree information from the input data.

Table 7: Impact of degree-based branch weight $\alpha$ on performance (Cora dataset used).

| | 3 Clients | | | | 5 Clients | | | | 10 Clients | | | |
|---|---|---|---|---|---|---|---|---|---|---|---|---|
| | SG | UN | MC | NC | SG | UN | MC | NC | SG | UN | MC | NC |
| FedLoG ($\alpha = 0.5$) | **0.8613** (0.0108) | 0.7154 (0.0239) | 0.5668 (0.0330) | 0.4852 (0.0329) | 0.8519 (0.0065) | 0.7397 (0.0016) | 0.4837 (0.0331) | 0.3982 (0.0055) | 0.8377 (0.0089) | 0.7160 (0.0770) | 0.3769 (0.1721) | 0.5285 (0.0986) |
| FedLoG | 0.8601 (0.0118) | **0.7341** (0.0273) | **0.6472** (0.0811) | **0.5047** (0.0884) | **0.8575** (0.0074) | **0.7413** (0.0316) | **0.4948** (0.0930) | **0.4439** (0.0455) | **0.8451** (0.0103) | **0.7406** (0.0527) | **0.4037** (0.0619) | **0.6055** (0.0914) |
| Improvement (%p) | −0.12 | +1.87 | +8.04 | +1.95 | +0.56 | +0.16 | +1.11 | +4.57 | +0.74 | +2.46 | +2.68 | +7.70 |

**SG**: Seen Graph, **UN**: Unseen Node, **MC**: Missing Class, **NC**: New Client

To provide additional clarity, we conducted an ablation study to evaluate the effectiveness of weight averaging based on the target node's degree. In this ablation, we fixed the $\alpha$ value in Eq. 3 at 0.5, preventing degree-specific knowledge from being divided across branches. This configuration results in a simple ensemble of the two branches without considering the degree. The results of this ablation study are shown in Table 7.

The performance on the unseen data settings (i.e., **UN**, **MC**, and **NC**) differs significantly from that on the **SG** (Seen Graph) setting. Specifically, eliminating degree-based weight averaging leads to a significant performance decrease in unseen data settings. This difference arises because, in unseen data settings, the model relies more heavily on the reliability of global synthetic data. Consequently, weighting predictions from each branch based on the target node's degree effectively extracts reliable knowledge into the head branch while preserving tail-specific knowledge within the tail branch.

### A.11.2 ASSESSING THE ADAPTIVE IMPACT OF FEATURE SCALING ON LOCAL CLIENTS

FedLoG shares the same global synthetic data with clients at the end of each round, but clients have distinct absent knowledge due to different label distributions. Clients in FedLoG adaptively utilize the global synthetic data by adjusting its perturbation strength in a class-wise manner, as described in Section 4.3. We verify that the adaptive factor optimally adjusts the perturbation strength for local clients. Table 8 shows the value of the adaptive factor for class $c$ (i.e., $\gamma[c]$) in Client 1 at the last round $R$. The adaptive factor for the head class (i.e., class 3) is higher than that for tail (i.e., class 1) and missing classes (i.e., class 5), showing that it effectively adjusts the perturbation strength based on each client's current learning status.

Table 8: $\gamma[c]$ values at the final round $R$ (CiteSeer - 3 Clients).

| Class | 1 | 2 | 3 | 4 | 5 | 6 |
|---|---|---|---|---|---|---|
| # Nodes | 5 | 27 | **129** | 16 | **0** | 25 |
| $\gamma[c]$ | 0.288 | 0.289 | **0.297** | 0.282 | **0.282** | 0.286 |

### A.11.3 EXPERIMENTAL RESULTS ON THE OPEN SET

We evaluate the Unseen Node and Missing Class in the Open Set settings to validate the model's ability to generalize to nodes never seen at the global level. The results are provided in Table 9. Similar to the Closed Set, our method, FedLoG, outperforms the baselines across most settings. However, in the Unseen Node setting on the PubMed dataset, some baselines show better performance than our method. We attribute this to the PubMed dataset providing a sufficient number of training data for each class, allowing methods to generalize well within each class's local data. Conversely, in the Missing Class setting, the baselines fail to generalize due to the absence of local data for the missing classes. In contrast, our model effectively generalizes to all classes, including missing classes, demonstrating its robustness on various real-world scenarios.

### A.11.4 ABLATION STUDY

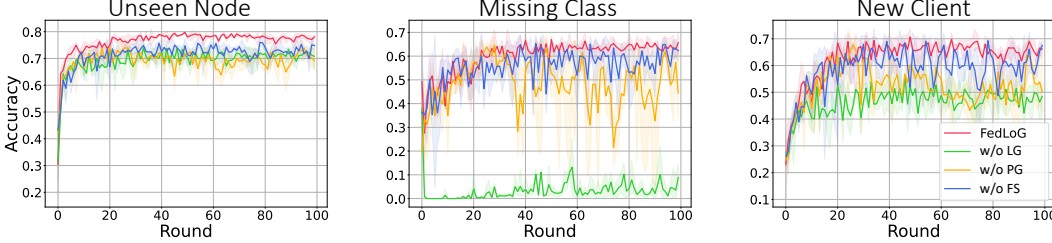

Figure 10: Ablation studies (CiteSeer - 3 Clients).

Table 9: Performance on Unseen Node and Missing Class in the Open Set setting.

**(a) Unseen Node**

| Methods | Cora 3 Clients | Cora 5 Clients | Cora 10 Clients | CiteSeer 3 Clients | CiteSeer 5 Clients | CiteSeer 10 Clients | PubMed 3 Clients | PubMed 5 Clients | PubMed 10 Clients | Amazon Photo 3 Clients | Amazon Photo 5 Clients | Amazon Photo 10 Clients | Amazon Computers 3 Clients | Amazon Computers 5 Clients | Amazon Computers 10 Clients |
|---|---|---|---|---|---|---|---|---|---|---|---|---|---|---|---|
| Local | 0.1250 (0.0030) | 0.2957 (0.0077) | 0.2854 (0.0263) | 0.4443 (0.0131) | 0.3471 (0.0020) | 0.5177 (0.0052) | 0.7510 (0.0010) | 0.7292 (0.0000) | 0.7489 (0.0013) | 0.1333 (0.0000) | 0.1900 (0.0039) | 0.3958 (0.0211) | 0.1687 (0.0001) | 0.2891 (0.0000) | 0.3890 (0.0043) |
| FedAvg | 0.6696 (0.0232) | 0.5939 (0.0215) | 0.4243 (0.1304) | 0.6055 (0.0033) | 0.7126 (0.0210) | 0.5255 (0.0119) | 0.8679 (0.0059) | 0.7192 (0.0170) | 0.6793 (0.0127) | 0.2481 (0.0455) | 0.2491 (0.0671) | 0.2692 (0.0304) | 0.3480 (0.0428) | 0.2980 (0.0198) | 0.2617 (0.0074) |
| FedSAGE+ | 0.6362 (0.0764) | 0.5050 (0.0047) | 0.3953 (0.0527) | 0.4090 (0.0155) | 0.2667 (0.0160) | 0.3945 (0.0571) | 0.9035 (0.0028) | 0.8820 (0.0015) | 0.8312 (0.0124) | 0.3117 (0.0071) | 0.2529 (0.0198) | 0.3651 (0.0183) | 0.4205 (0.0073) | 0.6028 (0.0055) | 0.3404 (0.0189) |
| FedGCN | 0.6840 (0.0083) | 0.6299 (0.0022) | 0.4389 (0.1433) | 0.6148 (0.0124) | 0.6500 (0.0319) | 0.5767 (0.0254) | 0.8571 (0.0027) | 0.7138 (0.0114) | 0.6558 (0.0044) | 0.2329 (0.0448) | 0.2411 (0.0512) | 0.2617 (0.0238) | 0.3519 (0.0506) | 0.2923 (0.0235) | 0.2621 (0.0041) |
| FedPUB | 0.6772 (0.0039) | 0.5971 (0.0117) | 0.4717 (0.0114) | 0.6097 (0.0264) | 0.7222 (0.0087) | 0.5958 (0.0049) | 0.8842 (0.0114) | 0.8954 (0.0022) | 0.8864 (0.0023) | 0.4842 (0.0204) | 0.5109 (0.0331) | 0.3790 (0.0359) | 0.4886 (0.0123) | 0.5068 (0.0286) | 0.4574 (0.0125) |
| FedNTD | 0.7066 (0.0241) | 0.6402 (0.0076) | 0.4245 (0.1274) | 0.6443 (0.0105) | 0.7639 (0.0082) | 0.5664 (0.0168) | 0.8953 (0.0052) | 0.8769 (0.0033) | 0.8789 (0.0013) | 0.5516 (0.0283) | 0.6196 (0.0098) | 0.4903 (0.0165) | 0.4183 (0.0033) | 0.6707 (0.0252) | 0.6778 (0.0102) |
| FedED | 0.6904 (0.0163) | 0.5453 (0.0185) | 0.3024 (0.0038) | 0.5985 (0.0330) | 0.6568 (0.0060) | 0.4448 (0.0232) | 0.8978 (0.0047) | 0.8771 (0.0043) | 0.8805 (0.0028) | 0.6491 (0.0119) | 0.5872 (0.0291) | 0.2581 (0.0374) | 0.4326 (0.0291) | 0.7420 (0.0201) | 0.5751 (0.0291) |
| FedLoG | **0.7224** (0.0102) | **0.7163** (0.0089) | **0.6203** (0.0089) | **0.6363** (0.0153) | **0.7645** (0.0141) | **0.6634** (0.0235) | 0.8627 (0.0078) | 0.8622 (0.0058) | 0.8627 (0.0062) | **0.8754** (0.0049) | **0.8275** (0.0340) | **0.6576** (0.0202) | **0.7759** (0.0475) | **0.8625** (0.0180) | **0.7163** (0.0279) |

**(b) Missing Class**

| Methods | Cora 3 Clients | Cora 5 Clients | Cora 10 Clients | CiteSeer 3 Clients | CiteSeer 5 Clients | CiteSeer 10 Clients | PubMed 3 Clients | PubMed 5 Clients | PubMed 10 Clients | Amazon Photo 3 Clients | Amazon Photo 5 Clients | Amazon Photo 10 Clients | Amazon Computers 3 Clients | Amazon Computers 5 Clients | Amazon Computers 10 Clients |
|---|---|---|---|---|---|---|---|---|---|---|---|---|---|---|---|
| Local | 0.0000 (0.0000) | 0.0000 (0.0000) | 0.0000 (0.0000) | 0.0000 (0.0000) | 0.0000 (0.0000) | 0.0000 (0.0000) | 0.0000 (0.0000) | 0.0000 (0.0000) | 0.0000 (0.0000) | 0.0000 (0.0000) | 0.0000 (0.0000) | 0.0000 (0.0000) | 0.0001 (0.0000) | 0.0000 (0.0000) | 0.0000 (0.0000) |
| FedAvg | 0.0000 (0.0000) | 0.2091 (0.0291) | 0.0000 (0.0317) | 0.1801 (0.0405) | 0.4269 (0.0517) | 0.1490 (0.0387) | 0.2771 (0.0207) | 0.0499 (0.0133) | 0.0166 (0.0064) | 0.0000 (0.0000) | 0.0000 (0.0000) | 0.0000 (0.0000) | 0.0000 (0.0000) | 0.0000 (0.0000) | 0.0000 (0.0000) |
| FedSAGE+ | 0.2030 (0.0883) | 0.2774 (0.0528) | 0.0244 (0.0326) | 0.3243 (0.2832) | 0.4155 (0.0220) | 0.2007 (0.1161) | 0.0495 (0.016) | 0.0733 (0.0160) | 0.1166 (0.0217) | 0.0000 (0.0000) | 0.0000 (0.0000) | 0.0175 (0.0304) | 0.0000 (0.0000) | 0.0000 (0.0000) | 0.0189 (0.0327) |
| FedGCN | 0.0000 (0.0000) | 0.2940 (0.0280) | 0.0579 (0.0520) | 0.0961 (0.0364) | 0.3562 (0.1246) | 0.1831 (0.0253) | 0.2035 (0.0165) | 0.0478 (0.0058) | 0.0049 (0.0012) | 0.0000 (0.0000) | 0.0000 (0.0000) | 0.0000 (0.0000) | 0.0000 (0.0000) | 0.0000 (0.0000) | 0.0177 (0.0085) |
| FedPUB | 0.0000 (0.0000) | 0.0082 (0.0094) | 0.0352 (0.0000) | 0.0000 (0.0000) | 0.0251 (0.0220) | 0.0070 (0.0060) | 0.0318 (0.0000) | 0.0002 (0.0000) | 0.0100 (0.0087) | 0.0000 (0.0000) | 0.0000 (0.0000) | 0.0000 (0.0000) | 0.0000 (0.0000) | 0.0000 (0.0000) | 0.0000 (0.0000) |
| FedNTD | 0.1182 (0.0686) | 0.2650 (0.0189) | 0.0457 (0.0402) | 0.4054 (0.0504) | 0.5822 (0.0504) | 0.2054 (0.0166) | 0.0733 (0.0638) | 0.2688 (0.0263) | 0.3895 (0.0263) | 0.0292 (0.0479) | 0.0385 (0.0376) | 0.1558 (0.0305) | 0.1708 (0.0137) | 0.0088 (0.0047) | 0.0017 (0.0014) |
| FedED | 0.1333 (0.0844) | 0.1449 (0.013) | 0.0370 (0.0339) | 0.2523 (0.0375) | 0.2922 (0.0364) | 0.1197 (0.0412) | 0.1487 (0.0460) | 0.1118 (0.0696) | 0.1604 (0.0067) | 0.0503 (0.0820) | 0.0000 (0.0000) | 0.0029 (0.0050) | 0.0613 (0.0434) | 0.1232 (0.0973) | 0.0091 (0.0463) |
| FedLoG | **0.4273** (0.0567) | **0.5528** (0.0569) | **0.2649** (0.0174) | **0.4234** (0.0324) | **0.5342** (0.0449) | **0.4484** (0.0205) | **0.5697** (0.2118) | **0.4758** (0.1292) | **0.5697** (0.2169) | **0.3333** (0.0142) | **0.5423** (0.1803) | **0.4397** (0.1365) | **0.7648** (0.0985) | **0.2548** (0.0284) | **0.2929** (0.0348) |

We perform an ablation study on **1)** Local Generalization (w/o LG), **2)** Prompt Generation (w/o PG), and **3)** Feature Scaling (w/o FS). As these modules are all directly related to addressing unseen data, we depict the test accuracy curves in Unseen Data settings to easily verify the effectiveness of each module.

**Local Generalization**  Local Generalization is an essential phase to prevent local overfitting after the local updates of each client within the FL framework. The Local Generalization phase enables clients to learn locally absent knowledge from the global synthetic data, allowing them to generalize all classes even if they don't have any data for certain classes within their local data (i.e., missing class). As shown in Figure 10, our method without the Local Generalization phase fails to generalize the missing class, which means Local Generalization is crucial for addressing the absent knowledge. Furthermore, for the Unseen Node and New Client settings, the performance deteriorates when we omit the Local Generalization phase.

**Prompt Generation**  We evaluate the effectiveness of the prompt generators $\mathbf{PG}^c \forall [c]$. The prompt generators generate the prompt nodes of the global synthetic data $\mathcal{D}_g$, which contain the $h$-hop neighbor information for the target nodes and also contribute to training by mimicking the true $h$-hop subgraphs' gradient. We perform the ablation study for the prompt generators by omitting the generation of prompt nodes for the global synthetic data, which means we train them without any generated prompts. In Figure 10, without prompt nodes, there is a discrepancy in the training mechanism of the GNN between isolated nodes and nodes within the graph structure, leading to a performance decrease for all settings. Furthermore, the learning curves fluctuate when training the global synthetic data without prompt generation, indicating that using only the features of synthetic nodes negatively affects stability.

**Feature Scaling**  Feature Scaling helps each client learn all classes adaptively. Feature Scaling adjusts the strength of the perturbation of the global synthetic data for each client depending on the class prediction ability for all classes at the current round. Thus, Feature Scaling affects the stability of learning for each client. In Figure 10, we can verify the effectiveness of Feature Scaling, as the learning curves are more fluctuating than the original FedLoG method, and the performance is decreased.

## A.12  DATASETS

**Cora (Sen et al., 2008):**  The Cora dataset consists of 2,708 scientific publications classified into one of seven classes. The citation network contains 5,429 links. Each publication in the dataset is described by a 1,433-dimensional binary vector, indicating the absence/presence of a word from a dictionary.

**CiteSeer (Sen et al., 2008):** The CiteSeer dataset comprises 3,327 scientific publications classified into one of six classes. The citation network consists of 4,732 links. Each publication is described by a 3,703-dimensional binary vector.

**PubMed (Sen et al., 2008):** The PubMed dataset includes 19,717 scientific publications from the PubMed database pertaining to diabetes, classified into one of three classes. The citation network comprises 44,338 links. Each publication is described by a TF/IDF-weighted word vector from a dictionary with a size of 500.

**Amazon Computers (McAuley et al., 2015):** The Amazon Computers dataset is a subset of the Amazon co-purchase graph. It consists of 13,752 nodes (products) and 245,861 edges (co-purchase relationships). Each product is described by a 767-dimensional feature vector, and the task is to classify products into 10 classes.

**Amazon Photos (Shchur et al., 2018):** The Amazon Photos dataset is another subset of the Amazon co-purchase graph. It consists of 7,650 nodes (products) and 143,663 edges (co-purchase relationships). Each product is described by a 745-dimensional feature vector, and the task is to classify products into 8 classes.

### A.12.1 DATASET STATISTICS

Table 10: Dataset Statistics

| Dataset | Nodes | Edges | Features | Classes | Description |
|---|---|---|---|---|---|
| Cora | 2,708 | 5,429 | 1,433 | 7 | Scientific publications |
| CiteSeer | 3,327 | 4,732 | 3,703 | 6 | Scientific publications |
| PubMed | 19,717 | 44,338 | 500 | 3 | Scientific publications |
| Amazon Computers | 13,752 | 245,861 | 767 | 10 | Amazon co-purchase |
| Amazon Photos | 7,650 | 143,663 | 745 | 8 | Amazon co-purchase |

### A.13 BASELINES

In this section, we provide details for the baselines and the URLs of the official codes, where available.

**Local.** This is a non-FL baseline where each local model is trained independently using the GCN embedder without any weight sharing.

**FedAvg. (McMahan et al., 2017)** This FL baseline involves clients sending their local model weights to the server, which then averages these weights based on the number of training samples at each client. The aggregated model is then distributed back to the clients. In our implementation, we use GCN as the graph embedder.

**FedSAGE+. (Zhang et al., 2021a)** This subgraph-FL baseline involves clients using GraphSAGE as an embedder and a missing neighbor generator, trained using a graph mending technique. The neighbor generator creates missing neighbors based on their number and features. With the neighbor generator, local models are trained with compensated neighbors and then their weights are aggregated on the server using FedAvg-based FL aggregation.

**FedGCN. (Yao et al., 2024)** This subgraph-FL baseline involves clients who collect $h$-hop averaged neighbor node features from other clients at the beginning of training to address missing information. The server then collects local model weights for FedAvg-based FL aggregation.

**FedPUB. (Baek et al., 2023)** This subgraph-FL baseline proposes weight aggregation based on the similarity between clients. It identifies highly correlated clients with similar community graph structures by using the functional embeddings of local GNNs, which are computed using random graphs as inputs to determine similarities.

**FedNTD. (Lee et al., 2022)**    This FL baseline is designed to tackle the challenge of overfitting in local models due to non-IID data across clients. It performs local-side distillation only for non-true classes to prevent forgetting global knowledge corresponding to regions outside the local distribution. In our implementation, we use GCN as the graph embedder.

**FedED. (Guo et al., 2024)**    This FL baseline is designed to tackle the challenge of overfitting in local models and addresses the issue of local missing classes. Similar to our task, it addresses the missing class problem in FL by adding a loss term that regularizes the logits of missing classes to be similar to those of the global model. In our implementation, we use GCN as the graph embedder.

Table 11: Baselines and their corresponding code repositories. * We utilized the FedAvg code implemented in the official FedPub code.

| Baseline | URL / Note |
| --- | --- |
| FedAvg* | `https://github.com/JinheonBaek/FED-PUB/` |
| FedSAGE | `https://github.com/zkhku/fedsage` |
| FedGCN | `https://github.com/yh-yao/FedGCN` |
| FedPub | `https://github.com/JinheonBaek/FED-PUB/` |
| FedNTD | `https://github.com/Lee-Gihun/FedNTD` |
| FedED | Self-implemented due to absence of official code. |

### A.14 IMPLEMENTATION DETAILS

In this section, we provide implementation details of FedLoG.

**Model Architecture.**    In our experiments, we use a 2-layer GraphSAGE (Hamilton et al., 2017) implementation ($\varphi_E$) with a dropout rate of 0.5, a hidden dimension of 128, and an output dimension of 64. The model parameters with learnable features $X_{\mathcal{V}_{k,\text{head}}}$ and $X_{\mathcal{V}_{k,\text{tail}}}$ are optimized with Adam (Kingma & Ba, 2014) using a learning rate of 0.001. The classifiers $\theta_H$ and $\theta_T$ consist of 2 main learnable functions (i.e., **MLP**$_{\text{msg}}$ and **MLP**$_{\text{trans}}$) as follows:

- **Message generating function (MLP$_{\text{msg}}$):** Two linear layers with SiLU activation (Inputs $\rightarrow$ Linear $(2 \times 64 \rightarrow 64) \rightarrow$ SiLU $\rightarrow$ Linear $(64 \rightarrow 64) \rightarrow$ SiLU $\rightarrow$ Outputs).
- **Message embedding function (MLP$_{\text{trans}}$):** Three linear layers with SiLU activation (Inputs $\rightarrow$ Linear $(64 \rightarrow 64) \rightarrow$ SiLU $\rightarrow$ Linear $(64 \rightarrow 64) \rightarrow$ Linear $(64 \rightarrow 1) \rightarrow$ Outputs).

In all experiments, we utilize 2-layer classifiers.

**Training Details.**    Our method is implemented on Python 3.10, PyTorch 2.0.1, and Torch-geometric 2.4.0. All experiments are conducted using four 24GB NVIDIA GeForce RTX 4090 GPUs. For all experiments, we set the number of rounds ($R$) to 100 and the number of local epochs to 1. This setting is applied consistently across all baselines.

**Evaluation Details.**    For the evaluation under the Seen Node setting, we assess the test nodes using the model that achieves the best validation performance across all rounds (R) in the Seen Node setting. This model is then used to evaluate performance in the other unseen data settings: Unseen Node, Missing Class, and New Client, by testing on the corresponding test nodes for each setting.

**Hyperparameters.**    We set the number of learnable nodes $s$ to 20, the tail-degree threshold $\gamma$ to 3, and select the regularization parameter $\beta$ to values in the range of [0.01, 0.1, 1].

### A.15 DETAILED PROCESS OF GENERATING HH/HT/TH/TT GLOBAL SYNTHETIC DATA

In this section, we describe the process of generating global synthetic data using **1)** head class & head degree nodes (HH), **2)** head class & tail degree nodes (HT), **3)** tail class & head degree nodes (TH), and **4)** tail class & tail degree nodes (TT). FedLoG has two branches, each generating $\mathcal{V}_{k,\text{head}}$ and $\mathcal{V}_{k,\text{tail}}$, which contain knowledge from head degree nodes and tail degree nodes, respectively.

**HH.** As described in Section 4.2, we generate HH global synthetic data by merging the head degree condensed nodes $\mathcal{V}_{k,\text{head}}$ from all clients, weighted by the proportion of head classes for each client. In Figure 2(d), for each class $c \in \mathcal{C}$, the feature vector of the $i$-th global synthetic node for class $c$, $\boldsymbol{x}_{v_g^{(c,i)}}$, is defined as:

$$\boldsymbol{x}_{v_g^{(c,i)}} = \frac{1}{\sum_{k=1}^{K} r_k^c} \sum_{k=1}^{K} r_k^c \boldsymbol{x}_{v_{k,\text{head}}^{(c,i)}},$$

where $r_k^c = \frac{|\mathcal{V}_k^c|}{|\mathcal{V}_k|}$ represents the proportion of nodes labeled $c$ in the $k$-th client's dataset.

**HT.** In generating HT global synthetic data, we substitute $\mathcal{V}_{k,\text{head}}$ with $\mathcal{V}_{k,\text{tail}}$. Thus, for each class $c \in \mathcal{C}$, the feature vector of the $i$-th global synthetic node for class $c$, $\boldsymbol{x}_{v_g^{(c,i)}}$, is defined as:

$$\boldsymbol{x}_{v_g^{(c,i)}} = \frac{1}{\sum_{k=1}^{K} r_k^c} \sum_{k=1}^{K} r_k^c \boldsymbol{x}_{v_{k,\text{tail}}^{(c,i)}},$$

**TH.** For generating TH global synthetic data, we aim to give more weight to the tail classes. To achieve this, we adjust the weights inversely proportional to $r_k^c$, ensuring that tail classes (with lower $r_k^c$) receive higher weights. The new equation is given by:

$$\boldsymbol{x}_{v_g^{(c,i)}} = \frac{1}{\sum_{k=1}^{K} \alpha_k^c} \sum_{k=1}^{K} \alpha_k^c \boldsymbol{x}_{v_{k,\text{head}}^{(c,i)}}, \text{ where } \alpha_k^c = \frac{\sum_{j=1}^{K} r_j^c}{r_k^c + \epsilon} \tag{15}$$

Here, $\epsilon$ is a very small positive value added to prevent division by zero. In this revised equation, $\alpha_k^c$ assigns higher weights to classes with smaller $r_k^c$ values, thereby giving more importance to the tail classes.

**TT.** Finally, we generate TT global synthetic data using:

$$\boldsymbol{x}_{v_g^{(c,i)}} = \frac{1}{\sum_{k=1}^{K} \alpha_k^c} \sum_{k=1}^{K} \alpha_k^c \boldsymbol{x}_{v_{k,\text{tail}}^{(c,i)}}, \text{ where } \alpha_k^c = \frac{\sum_{j=1}^{K} r_j^c}{r_k^c + \epsilon} \tag{16}$$

## A.16 EXPERIMENTAL DATASET STATISTICS

In this section, we provide the experimental dataset statistics for all testing settings for three clients, allowing for an easy verification of the data distribution of each client and the New Client. In the 'Global' row, we sum up the statistics from all local clients.

Table 12: Cora Dataset Statistics (Closed Set)

| Dataset | Class | Seen Graph | | | Unseen Node | Missing Class |
| | | Train | Valid | Test | Test | Test |
|---|---|---|---|---|---|---|
| Global | 0 | 49 | 37 | 32 | 268 | 86 |
| | 1 | 82 | 45 | 75 | 258 | 0 |
| | 2 | 140 | 110 | 133 | 217 | 220 |
| | 3 | 242 | 206 | 171 | 605 | 0 |
| | 4 | 120 | 82 | 90 | 268 | 0 |
| | 5 | 45 | 41 | 32 | 120 | 29 |
| | 6 | 53 | 42 | 27 | 9 | 59 |
| Client 0 | 0 | 8 | 12 | 4 | 121 | 0 |
| | 1 | 7 | 4 | 3 | 124 | 0 |
| | 2 | 4 | 2 | 3 | 190 | 0 |
| | 3 | 208 | 168 | 131 | 125 | 0 |
| | 4 | 33 | 22 | 22 | 96 | 0 |
| | 5 | 0 | 0 | 0 | 0 | 29 |
| | 6 | 0 | 1 | 0 | 1 | 23 |
| Client 1 | 0 | 0 | 0 | 0 | 0 | 86 |
| | 1 | 7 | 1 | 3 | 117 | 0 |
| | 2 | 136 | 108 | 130 | 27 | 0 |
| | 3 | 6 | 15 | 14 | 202 | 0 |
| | 4 | 74 | 49 | 60 | 74 | 0 |
| | 5 | 3 | 4 | 2 | 44 | 0 |
| | 6 | 0 | 0 | 0 | 0 | 36 |
| Client 2 | 0 | 41 | 25 | 28 | 147 | 0 |
| | 1 | 68 | 40 | 69 | 17 | 0 |
| | 2 | 0 | 0 | 0 | 0 | 220 |
| | 3 | 28 | 23 | 26 | 278 | 0 |
| | 4 | 13 | 11 | 8 | 98 | 0 |
| | 5 | 42 | 37 | 30 | 76 | 0 |
| | 6 | 53 | 41 | 27 | 8 | 0 |
| New Client | 0 | - | - | 222 | - | - |
| | 1 | - | - | 12 | - | - |
| | 2 | - | - | 2 | - | - |
| | 3 | - | - | 107 | - | - |
| | 4 | - | - | 87 | - | - |
| | 5 | - | - | 166 | - | - |
| | 6 | - | - | 9 | - | - |

Table 13: CiteSeer Dataset Statistics (Closed Set)

| Dataset | Class | Seen Graph | | | Unseen Node | Missing Class |
|---|---|---|---|---|---|---|
| | | Train | Valid | Test | Test | Test |
| Global | 0 | 41 | 24 | 25 | 72 | 0 |
| | 1 | 90 | 75 | 90 | 185 | 0 |
| | 2 | 196 | 163 | 137 | 424 | 93 |
| | 3 | 116 | 85 | 77 | 210 | 0 |
| | 4 | 154 | 96 | 110 | 132 | 121 |
| | 5 | 37 | 25 | 26 | 88 | 53 |
| Client 0 | 0 | 19 | 5 | 11 | 26 | 0 |
| | 1 | 52 | 50 | 58 | 59 | 0 |
| | 2 | 67 | 52 | 32 | 296 | 0 |
| | 3 | 73 | 52 | 44 | 76 | 0 |
| | 4 | 7 | 1 | 7 | 60 | 0 |
| | 5 | 0 | 0 | 0 | 0 | 53 |
| Client 1 | 0 | 5 | 5 | 2 | 18 | 0 |
| | 1 | 27 | 16 | 26 | 79 | 0 |
| | 2 | 129 | 111 | 105 | 128 | 0 |
| | 3 | 16 | 15 | 14 | 66 | 0 |
| | 4 | 0 | 0 | 0 | 0 | 121 |
| | 5 | 25 | 13 | 17 | 44 | 0 |
| Client 2 | 0 | 17 | 14 | 12 | 28 | 0 |
| | 1 | 11 | 9 | 6 | 47 | 0 |
| | 2 | 0 | 0 | 0 | 0 | 93 |
| | 3 | 27 | 18 | 19 | 68 | 0 |
| | 4 | 147 | 95 | 103 | 72 | 0 |
| | 5 | 12 | 12 | 9 | 44 | 0 |
| New Client | 0 | - | - | 35 | - | - |
| | 1 | - | - | 53 | - | - |
| | 2 | - | - | 12 | - | - |
| | 3 | - | - | 110 | - | - |
| | 4 | - | - | 93 | - | - |
| | 5 | - | - | 211 | - | - |

Table 14: PubMed Dataset Statistics (Closed Set)

| Dataset | Class | Seen Graph | | | Unseen Node | Missing Class |
| | | Train | Valid | Test | Test | Test |
|---|---|---|---|---|---|---|
| Global | 0 | 1271 | 983 | 949 | 221 | 168 |
| | 1 | 1176 | 897 | 951 | 572 | 1003 |
| | 2 | 2972 | 2171 | 2141 | 658 | 0 |
| Client 0 | 0 | 277 | 222 | 209 | 93 | 0 |
| | 1 | 0 | 0 | 0 | 0 | 346 |
| | 2 | 1642 | 1227 | 1189 | 193 | 0 |
| Client 1 | 0 | 994 | 761 | 740 | 128 | 0 |
| | 1 | 0 | 0 | 0 | 0 | 657 |
| | 2 | 594 | 408 | 414 | 213 | 0 |
| Client 2 | 0 | 0 | 0 | 0 | 0 | 168 |
| | 1 | 1176 | 897 | 951 | 572 | 0 |
| | 2 | 736 | 536 | 538 | 252 | 0 |
| New Client | 0 | - | - | 787 | - | - |
| | 1 | - | - | 3500 | - | - |
| | 2 | - | - | 591 | - | - |

Table 15: Photos Dataset Statistics (Closed Set)

| Dataset | Class | Seen Graph | | | Unseen Node | Missing Class |
|---------|-------|-------|-------|------|-------------|---------------|
| | | Train | Valid | Test | Test | Test |
| Global | 0 | 150 | 102 | 108 | 61 | 28 |
| | 1 | 592 | 502 | 468 | 881 | 0 |
| | 2 | 266 | 219 | 197 | 61 | 20 |
| | 3 | 358 | 241 | 258 | 329 | 0 |
| | 4 | 300 | 258 | 250 | 327 | 0 |
| | 5 | 332 | 217 | 247 | 0 | 25 |
| | 6 | 214 | 146 | 128 | 907 | 0 |
| | 7 | 39 | 17 | 27 | 307 | 0 |
| Client 0 | 0 | 0 | 0 | 0 | 0 | 28 |
| | 1 | 71 | 50 | 49 | 345 | 0 |
| | 2 | 261 | 215 | 196 | 3 | 0 |
| | 3 | 49 | 26 | 33 | 135 | 0 |
| | 4 | 5 | 6 | 10 | 186 | 0 |
| | 5 | 332 | 217 | 247 | 0 | 0 |
| | 6 | 9 | 5 | 7 | 96 | 0 |
| | 7 | 9 | 5 | 9 | 59 | 0 |
| Client 1 | 0 | 146 | 100 | 107 | 7 | 0 |
| | 1 | 394 | 369 | 333 | 226 | 0 |
| | 2 | 0 | 0 | 0 | 0 | 20 |
| | 3 | 11 | 8 | 8 | 101 | 0 |
| | 4 | 3 | 1 | 0 | 67 | 0 |
| | 5 | 0 | 0 | 0 | 0 | 18 |
| | 6 | 197 | 132 | 113 | 69 | 0 |
| | 7 | 1 | 0 | 3 | 40 | 0 |
| Client 2 | 0 | 4 | 2 | 1 | 54 | 0 |
| | 1 | 127 | 83 | 86 | 310 | 0 |
| | 2 | 5 | 4 | 1 | 58 | 0 |
| | 3 | 298 | 207 | 217 | 93 | 0 |
| | 4 | 292 | 251 | 240 | 74 | 0 |
| | 5 | 0 | 0 | 0 | 0 | 7 |
| | 6 | 8 | 9 | 8 | 742 | 0 |
| | 7 | 29 | 12 | 15 | 208 | 0 |
| New Client | 0 | - | - | 0 | - | - |
| | 1 | - | - | 72 | - | - |
| | 2 | - | - | 3 | - | - |
| | 3 | - | - | 43 | - | - |
| | 4 | - | - | 64 | - | - |
| | 5 | - | - | 1 | - | - |
| | 6 | - | - | 1412 | - | - |
| | 7 | - | - | 248 | - | - |

Table 16: Computers Dataset Statistics (Closed Set)

| Dataset | Class | Seen Graph | | | Unseen Node | Missing Class |
|---|---|---|---|---|---|---|
| | | Train | Valid | Test | Test | Test |
| Global | 0 | 168 | 120 | 115 | 146 | 140 |
| | 1 | 297 | 221 | 201 | 1604 | 0 |
| | 2 | 558 | 442 | 410 | 2 | 479 |
| | 3 | 91 | 63 | 66 | 765 | 0 |
| | 4 | 1399 | 1094 | 1103 | 3397 | 0 |
| | 5 | 129 | 69 | 98 | 0 | 60 |
| | 6 | 182 | 134 | 164 | 132 | 93 |
| | 7 | 343 | 220 | 232 | 4 | 167 |
| | 8 | 748 | 597 | 580 | 1321 | 0 |
| | 9 | 119 | 80 | 79 | 105 | 22 |
| Client 0 | 0 | 164 | 118 | 114 | 32 | 0 |
| | 1 | 98 | 65 | 71 | 398 | 0 |
| | 2 | 558 | 442 | 410 | 2 | 0 |
| | 3 | 66 | 42 | 46 | 206 | 0 |
| | 4 | 41 | 25 | 30 | 813 | 0 |
| | 5 | 0 | 0 | 0 | 0 | 46 |
| | 6 | 0 | 0 | 0 | 0 | 93 |
| | 7 | 343 | 220 | 232 | 4 | 0 |
| | 8 | 12 | 10 | 8 | 447 | 0 |
| | 9 | 108 | 59 | 61 | 25 | 0 |
| Client 1 | 0 | 4 | 2 | 1 | 114 | 0 |
| | 1 | 125 | 84 | 80 | 573 | 0 |
| | 2 | 0 | 0 | 0 | 0 | 262 |
| | 3 | 7 | 6 | 5 | 235 | 0 |
| | 4 | 139 | 123 | 117 | 1504 | 0 |
| | 5 | 129 | 69 | 98 | 0 | 0 |
| | 6 | 181 | 133 | 164 | 4 | 0 |
| | 7 | 0 | 0 | 0 | 0 | 143 |
| | 8 | 707 | 561 | 552 | 178 | 0 |
| | 9 | 11 | 21 | 18 | 80 | 0 |
| Client 2 | 0 | 0 | 0 | 0 | 0 | 140 |
| | 1 | 74 | 72 | 50 | 633 | 0 |
| | 2 | 0 | 0 | 0 | 0 | 217 |
| | 3 | 18 | 15 | 15 | 324 | 0 |
| | 4 | 1219 | 946 | 956 | 1080 | 0 |
| | 5 | 0 | 0 | 0 | 0 | 14 |
| | 6 | 1 | 1 | 0 | 128 | 0 |
| | 7 | 0 | 0 | 0 | 0 | 24 |
| | 8 | 29 | 26 | 20 | 696 | 0 |
| | 9 | 0 | 0 | 0 | 0 | 22 |
| New Client | 0 | - | - | 30 | - | - |
| | 1 | - | - | 1374 | - | - |
| | 2 | - | - | 2 | - | - |
| | 3 | - | - | 302 | - | - |
| | 4 | - | - | 1360 | - | - |
| | 5 | - | - | 1 | - | - |
| | 6 | - | - | 1 | - | - |
| | 7 | - | - | 1 | - | - |
| | 8 | - | - | 167 | - | - |
| | 9 | - | - | 5 | - | - |

