# OpenReview forum: "Subgraph Federated Learning for Local Generalization"
_ICLR.cc/2025/Conference — ICLR 2025 Oral_

### Official Review · Reviewer_T9gM · 2024-10-30

**Soundness:** 4
**Presentation:** 4
**Contribution:** 4
**Rating:** 10
**Confidence:** 4

**Summary:**

This paper introduces FedLoG, a novel framework addressing local overfitting in subgraph federated learning. The key innovation lies in its approach to generating global synthetic data by condensing reliable information from head degree and head class nodes across clients. The framework employs a dual-branch architecture to process head and tail degree nodes differently, with adaptive weighting based on node degrees. Through feature scaling and prompt generation mechanisms, FedLoG enables each client to adaptively learn locally absent knowledge. The method is evaluated under three practical scenarios: unseen nodes, missing classes, and new clients. Experimental results demonstrate FedLoG's effectiveness in preventing local overfitting while maintaining privacy and improving generalization capabilities across various graph datasets.

**Strengths:**

Originality:
The paper introduces FedLoG, a framework that innovatively addresses the problem of local overfitting in subgraph federated learning by focusing on the adaptability and generalization of models across clients. The originality lies in its method of using global synthetic data generated from the most reliable node types—head degree and head class nodes—thus tackling the challenge of missing classes and mutable graph structures in a novel way.

Quality:
The research is characterized by its high quality, demonstrated through a comprehensive experimental setup that includes diverse real-world datasets and practical scenarios like unseen nodes, missing classes, and new clients. The thoroughness of the ablation studies and the detailed analysis of the results contribute to a robust validation of the proposed method, ensuring that the findings are both reliable and applicable.

Clarity:
The paper is clearly written, with well-organized sections that effectively convey complex ideas. The authors provide detailed explanations of their methodology, supported by illustrative diagrams and thorough descriptions of experimental setups. This clarity facilitates a deeper understanding of the framework's mechanisms and its impact on federated learning.

Significance:
The significance of this work is underscored by its potential to enhance federated learning applications where privacy and adaptability are crucial. By improving the generalization capabilities of local models and addressing the mutable nature of graph data, the paper contributes significantly to advancing federated learning technologies. Its approach to mitigating local overfitting while respecting privacy constraints is particularly relevant in today's data-sensitive environment, offering practical benefits and paving the way for future research in this area.

**Weaknesses:**

1） The paper seems to lack a thorough discussion on the complexity of the proposed FedLoG framework. Given the dual-branch architecture that separately handles head and tail degree nodes, alongside the mechanisms for feature scaling and prompt generation, a detailed analysis of computational and implementation complexities would be beneficial. Understanding these complexities is crucial for assessing the feasibility and scalability of the method in practical, large-scale applications.

2）Another potential limitation is the assumption that each client has distinct strengths in terms of node types or class distributions. While this assumption aids in leveraging reliable data from head nodes, it may not hold in scenarios where client data distributions are highly homogenous or overlapping. This could limit the framework's effectiveness and generalization in such environments.

**Questions:**

The datasets used in this paper are all publicly available (i.e. they are not task-oriented) and have been specifically processed to fit the described scenarios. While this approach provides a controlled environment for testing, is there potential for future work to incorporate specialized datasets that more naturally exhibit the challenges addressed by FedLoG? Such datasets could offer deeper insights into the framework's performance and applicability in real-world contexts where these issues are more prevalent.

---

> ### Author Response · Authors · 2024-11-20
> **(1/N) Author Rebuttal**
>
> Thank you for your valuable feedback and thorough review!
>
> > 1） The paper seems to lack a thorough discussion on the complexity of the proposed FedLoG framework. Given the dual-branch architecture that separately handles head and tail degree nodes, alongside the mechanisms for feature scaling and prompt generation, a detailed analysis of computational and implementation complexities would be beneficial. Understanding these complexities is crucial for assessing the feasibility and scalability of the method in practical, large-scale applications.
>
>
> **1) Regarding the time complexities,**
>
> __We would like to emphasize that each classifier, including the Prompt Generator and Feature Scaling modules, has the same time complexity as a standard MLP.__  Specifically, the time complexity of our classifiers is $O(d^2)$, which matches that of an MLP model. Below, we provide the detailed calculations for each module:
>
> - **Pairwise Distance between Prototypes (Equation (7))**:
>   The naive time complexity is $O(PF)$, where $P$ is the number of prototypes ($|C_v| \times s$) and $F$ is the dimension of the input ($2 \times d$, where $d$ denotes the dimension of the representation $h_v$). Since $P$ is small enough to be negligible, the complexity becomes $O(d)$.
>
> - **Distance-Based Message Generation (Equation (6))**:
>   The naive time complexity is $O(PF^2)$, where $F$ is the input dimension ($2d + 1$). With $P$ being negligible, this results in a complexity of $O(d^2)$, which is the same as that of an MLP.
>
> - **Updating the Target Node’s Representation (Equations (8-10))**:
>   Equation (8) includes an MLP and elementwise operations with subtraction, giving a total complexity of $O(d^2)$. Equations (9) and (10) involve only simple additions and are therefore negligible. Thus, the total time complexity for this update is $O(d^2)$.
>
> - **Prompt Generator**:
>   In the pretraining phase, the prompt generator requires $O(|V|d)$ complexity for Equation (11), and $O(|\mathcal{E}|d + |V|d^2)$ for Equation (12). Therefore, the total time complexity of the prompt generator is $O(|\mathcal{E}|d + |V|d^2)$, which is the same as the GNN encoder. However, it is worth noting that, during inference, it requires only $O(d^2)$, which has the same time complexity as the MLP.
>
> - **Feature Scaling**:
>  For feature scaling in Equation (5), the time complexity is $O(d)$ since the operation involves only simple element-wise additions.
>
> Consequently, the total time complexity of the classifiers, including the Prompt Generator and Feature Scaling, is $O(d^2)$. This is significantly lighter than the complexity of the GNN encoder, $O(|\mathcal{E}|d + |V|d^2)$, where $|\mathcal{E}|$ and $|V|$ denote the number of edges and nodes, respectively.
>
> - **Graph Encoder**:
>   As we utilize GraphSAGE for the GNN encoder, it requires $O(|\mathcal{E}|d + |V|d^2)$ for both forward and backward time.
>
> To sum up, our model requires $O(|\mathcal{E}|d + |V|d^2)$ complexity, which is the same as a GNN encoder. Again, it is worth noting that each classifier has the same time complexity as an MLP (i.e., $O(d^2)$), which has little influence on the total complexity of our architecture.
>
> You can refer to the updated PDF in Appendix A.5, where the revisions are highlighted in red.
>
> __2) Regarding communication overhead,__
>
> We have provided an analysis of communication overhead in Appendix A.3. Over 100 rounds, our method requires 0.65x to 2.57x the communication cost compared to other baselines, meaning our method does not always incur the highest communication cost. As shown in Table 6 in Appendix A.3, our method demonstrates faster convergence due to its use of reliable class representations, resulting in lower communication costs to achieve the same accuracy. For example, with the Cora dataset and 3 clients, our method achieves relatively low cost (second best) among the baselines.
>
> Thank you for your suggestion. We will ensure to include this complexity analysis, as it will significantly enhance the quality of our paper.

---

> ### Author Response · Authors · 2024-11-20
> **(N/N) Author Rebuttal**
>
> > 2）Another potential limitation is the assumption that each client has distinct strengths in terms of node types or class distributions. While this assumption aids in leveraging reliable data from head nodes, it may not hold in scenarios where client data distributions are highly homogenous or overlapping. This could limit the framework's effectiveness and generalization in such environments.
>
> Thank you for highlighting a potential limitation of our work. We are happy to validate whether our method remains effective in highly homogeneous environments.
>
> To compare i.i.d. and non-i.i.d. partitioning (assuming the same class distribution for each client under i.i.d. settings, but different class distribution for each client under non-i.i.d. settings), we partition the global graph based on the Dirichlet distribution by adjusting the alpha value (which is distinct from the alpha in the paper that weights the prediction across branches). The alpha value in a Dirichlet distribution determines the level of balance of the class distribution among the partitions: a small alpha value results in highly imbalanced class distributions, while a large alpha value leads to more balanced class distributions across clients. We provide results on the Cora dataset with 3 clients for alpha values in [0.05, 0.1, 0.5, 1.0, 10, 100, 1000] in Table R1.
>
> |  **[Table R1]**      $\alpha=$         | 0.05  | 0.1   | 0.5   | 1.0   | 10    | 100   | 1000  |
> |----------------------|-------|-------|-------|-------|-------|-------|-------|
> | **FedAvg**           | 0.9083| 0.7965| 0.9179| 0.8525| 0.7992| 0.8308| 0.8235|
> | **FedSAGE+**         | 0.6347| 0.8102| 0.7197| 0.8508| 0.8166| 0.8213| 0.8392|
> | **FedGCN**           | 0.8083| 0.8116| 0.8093| 0.8103| 0.8087| 0.8080| 0.8070|
> | **FedPUB**           | 0.9126| 0.8200| 0.9197| **0.8902**| 0.8031| 0.8422| 0.8353|
> | **FedNTD**           | 0.8711| 0.8180| 0.9094| 0.8852| 0.8108| 0.8403| 0.8333|
> | **FedED**            | 0.8768| 0.8082| 0.9026| 0.8820| 0.8012| 0.8194| 0.8294|
> | **FedLoG**           | **0.9255**| **0.8219**| **0.9231**| 0.8869| **0.8301**| **0.8498**| **0.8569**|
>
> As shown in Table R1, FedLoG continues to outperform baselines from non-i.i.d. (i.e., alpha values 0.05 to 1.0) to i.i.d. (i.e., alpha values 10 to 1000) settings. We argue that FedLoG successfully distills reliable knowledge from each local distribution into the synthetic data. This allows clients to effectively learn absent knowledge through Local Generalization (Section 4.3), regardless of the balance or imbalance extent across the clients.
>
> ---
>
> > Q1) The datasets used in this paper are all publicly available (i.e. they are not task-oriented) and have been specifically processed to fit the described scenarios. While this approach provides a controlled environment for testing, is there potential for future work to incorporate specialized datasets that more naturally exhibit the challenges addressed by FedLoG? Such datasets could offer deeper insights into the framework's performance and applicability in real-world contexts where these issues are more prevalent.
>
> Thank you for your thoughtful question.
>
> We acknowledge that incorporating specialized datasets, which more naturally exhibit the dynamic and imbalanced conditions of real-world graphs, could further validate FedLoG's performance and applicability. Exploring such datasets could be a valuable direction for future work, particularly to investigate how well FedLoG generalizes and adapts to evolving graph structures and distributions beyond the controlled settings in this study.
>
> A potential example of a specialized dataset could involve hospital data, where each institution specializes in specific diseases, resulting in unique patient distributions. When a patient with unfamiliar characteristics (i.e., Unseen Data and Missing Class settings) visits a new institution, the system must adapt without being influenced by local biases. This scenario highlights the importance of FedLoG’s capability to effectively mitigate local overfitting. Exploring and validating FedLoG's performance in such scenarios as part of future work could provide meaningful insights into its practical applicability.
>
> Thank you for raising this important discussion point!

---

> > ### Comment · Reviewer_T9gM · 2024-11-20
> >
> > Thank you very much for answering all my confusions, I couldn't find any more shortcomings with this version and therefore intend to boost its score to 10. Good luck!

---

> ### Author Response · Authors · 2024-11-20
>
> Thank you for your positive score! Your feedback inspires us to continue striving for high-quality research. We sincerely appreciate your recognition of our efforts.

---

### Official Review · Reviewer_sNMQ · 2024-11-01

**Soundness:** 3
**Presentation:** 3
**Contribution:** 3
**Rating:** 6
**Confidence:** 3

**Summary:**

This paper proposes a novel personalization method to solve the local overfitting problem of the subgraph federated learning. Specifically, they propose a two-branch prototypical network for handling high-degree and low-degree knowledge of each node,  and a global data generation strategy for tackling the limitation of tail nodes in each client.

**Strengths:**

1.	Sufficient comparison of experimental results demonstrates the performance advantages of the proposed method.

2.	The figures and language are clear.

**Weaknesses:**

1. The local fitting process comprises two branches of modules and prototypical inference networks, which may bring expensive inference computational costs.

2. The ablation study about the effectiveness of dividing the inference process into two head and tail branches is required. In my opinion, this design works appearing more using the ensemble of two branches, instead of considering the head or tail degree of nodes. In fact, the effect of degree is only considered by using the weighted average of predictions, $\alpha$. The design of the two branches themselves are not involved to the degree at all.

3.	The construction of global synthetic data requires the uploading of local graph nodes, which not only brings extra communication costs but also causes privacy leakage.

4. This technique seems to apply a mixup of data in FL, which has been used in many previous works, e.g., [1,2]. The novelty may be limited.

[1] FedMix Approximation of Mixup under Mean Augmented Federated Learning NIPS 2022

[2] FedMDO: Privacy-Preserving Federated Learning via Mixup Differential Objective TCSVT 2024

5. Some key steps should be included in the maintext. For example, how to generate the edges for the synthetic data is required to be specified in the main text instead of the appendix.

6. In fact, this paper trains personalized models for each client since the learnable nodes and prototypes are customized for each client. Therefore, the personalization baselines should also be included [3].

[3] Federated Learning on Non-IID Graphs via Structural Knowledge Sharing. AAAI 2023.

**Questions:**

See above.

---

> ### Author Response · Authors · 2024-11-20
> **(1/N) Author Rebuttal**
>
> Thank you for your valuable feedback and thorough review!
>
> ### Weaknesses
> > 1. The local fitting process comprises two branches of modules and prototypical inference networks, which may bring expensive inference computational costs.
>
>
> **1) Regarding the time complexities,**
>
> __We would like to emphasize that each classifier, including the Prompt Generator and Feature Scaling modules, has the same time complexity as a standard MLP.__  Specifically, the time complexity of our classifiers is $O(d^2)$, which matches that of an MLP model. Below, we provide the detailed calculations for each module:
>
> - **Pairwise Distance between Prototypes (Equation (7))**:
>   The naive time complexity is $O(PF)$, where $P$ is the number of prototypes ($|C_v| \times s$) and $F$ is the dimension of the input ($2 \times d$, where $d$ denotes the dimension of the representation $h_v$). Since $P$ is small enough to be negligible, the complexity becomes $O(d)$.
>
> - **Distance-Based Message Generation (Equation (6))**:
>   The naive time complexity is $O(PF^2)$, where $F$ is the input dimension ($2d + 1$). With $P$ being negligible, this results in a complexity of $O(d^2)$, which is the same as that of an MLP.
>
> - **Updating the Target Node’s Representation (Equations (8-10))**:
>   Equation (8) includes an MLP and elementwise operations with subtraction, giving a total complexity of $O(d^2)$. Equations (9) and (10) involve only simple additions and are therefore negligible. Thus, the total time complexity for this update is $O(d^2)$.
>
> - **Prompt Generator**:
>   In the pretraining phase, the prompt generator requires $O(|V|d)$ complexity for Equation (11), and $O(|\mathcal{E}|d + |V|d^2)$ for Equation (12). Therefore, the total time complexity of the prompt generator is $O(|\mathcal{E}|d + |V|d^2)$, which is the same as the GNN encoder. However, it is worth noting that, during inference, it requires only $O(d^2)$, which has the same time complexity as the MLP.
>
> - **Feature Scaling**:
>  For feature scaling in Equation (5), the time complexity is $O(d)$ since the operation involves only simple element-wise additions.
>
> Consequently, the total time complexity of the classifiers, including the Prompt Generator and Feature Scaling, is $O(d^2)$. This is significantly lighter than the complexity of the GNN encoder, $O(|\mathcal{E}|d + |V|d^2)$, where $|\mathcal{E}|$ and $|V|$ denote the number of edges and nodes, respectively.
>
> - **Graph Encoder**:
>   As we utilize GraphSAGE for the GNN encoder, it requires $O(|\mathcal{E}|d + |V|d^2)$ for both forward and backward time.
>
> To sum up, our model requires $O(|\mathcal{E}|d + |V|d^2)$ complexity, which is the same as a GNN encoder. Again, it is worth noting that each classifier has the same time complexity as an MLP (i.e., $O(d^2)$), which has little influence on the total complexity of our architecture.
>
> You can refer to the updated PDF in Appendix A.5, where the revisions are highlighted in red.
>
> __2) Regarding communication overhead,__
>
> We have provided an analysis of communication overhead in Appendix A.3. Over 100 rounds, our method requires 0.65x to 2.57x the communication cost compared to other baselines, meaning our method does not always incur the highest communication cost. As shown in Table 6 in Appendix A.3, our method demonstrates faster convergence due to its use of reliable class representations, resulting in lower communication costs to achieve the same accuracy. For example, with the Cora dataset and 3 clients, our method achieves relatively low cost (second best) among the baselines.
>
> Additionally, we would like to emphasize that our method is the first to address the issue of local overfitting in the graph-FL domain, effectively handling the proposed unseen data settings (i.e., Unseen nodes, Missing class, New client), where other baselines are limited in their applicability.

---

> ### Author Response · Authors · 2024-11-20
> **(2/N) Author Rebuttal**
>
> > 2) The ablation study about the effectiveness of dividing the inference process into two head and tail branches is required. In my opinion, this design works appearing more using the ensemble of two branches, instead of considering the head or tail degree of nodes. In fact, the effect of degree is only considered by using the weighted average of predictions, . The design of the two branches themselves are not involved to the degree at all.
>
> Thank you for highlighting the effectiveness of dividing the head and tail branches.
> The primary objective of each branch is to distill knowledge from the input data into learnable synthetic data. To achieve this, we designed a prototypical network-based branch that uses learnable synthetic data to represent class-specific knowledge. By adjusting the weight (i.e., alpha) of each branch’s final prediction based on the target node's degree, we guide the gradient flow from head degree nodes primarily towards the head branch. This approach enables head degree knowledge to be distilled within the head branch’s synthetic data, and similarly, tail-degree knowledge within the tail branch.
>
> As shown in Section 3.1, tail-degree knowledge negatively impacts the performance of other clients. In Figure 3, the performance hierarchy follows the order HH > HT > TH > TT, highlighting two key insights: (1) when class headness (as indicated by the first letter) is fixed, synthetic data from head-branch nodes (as indicated by the second letter) consistently perform better, as demonstrated by HH > HT and TH > TT. This suggests that, according to the findings in Section 3.1, head-branch knowledge encapsulates more reliable information compared to tail-branch knowledge. These results confirm that the head branch effectively distills reliable knowledge, particularly head-degree knowledge, while the tail branch isolates less reliable tail-degree knowledge.
>
> This demonstrates that our framework successfully partitions degree-specific information into appropriate branches, improving the reliability of synthetic data generation.
>
> To provide additional clarity, we conducted an ablation study on the effectiveness of weight averaging based on the target node's degree. For this ablation, we fixed the alpha value at 0.5, i.e., give an equal weight to the two branches, to prevent the degree-specific knowledge from being divided across branches. This configuration results in a simple ensemble of the two branches, without consideration of degree. The results of this ablation study are as follows:
>
> |         | Cora (3 Clients) |          |          |          | Cora (5 Clients) |          |          |          | Cora (10 Clients) |          |          |          |
> |-----------------|------------------|----------|----------|----------|------------------|----------|----------|----------|-------------------|----------|----------|----------|
> | Setting         | SG               | UN       | MC       | NC       | SG               | UN       | MC       | NC       | SG                | UN       | MC       | NC       |
> | FedLoG ($\alpha=0.5$) | 0.8613           | 0.7154   | 0.5668   | 0.4852   | 0.8519           | 0.7397   | 0.4837   | 0.3982   | 0.8377            | 0.7160   | 0.3769   | 0.5285   |
> | FedLoG          | 0.8601           | 0.7341   | 0.6472   | 0.5047   | 0.8575           | 0.7413   | 0.4948   | 0.4439   | 0.8451            | 0.7406   | 0.4037   | 0.6055   |
> | Improvement (%p) | –0.12           | +1.87    | +8.04    | +1.95    | +0.56            | +0.16    | +1.11    | +4.57    | +0.74             | +2.46    | +2.68    | +7.70    |
>
> *SG: Seen Graph, UN: Unseen Node, MC: Missing Class, NC: New Client
>
> As shown in the results above, the performance on the unseen data settings (i.e., UN, MC, and NC) differs significantly from that on the Seen Graph setting. Specifically, when we eliminate degree-based weight averaging, the performance decreases significantly on unseen data settings. This difference arises because, in unseen data settings, the model relies more heavily on the reliability of the global synthetic data. Consequently, weighting the predictions from each branch based on the target node's degree effectively extracts reliable knowledge into the head branch while retaining tail-specific knowledge within the tail branch.
>
> You can refer to the updated PDF in Appendix A.11.2, where the revisions are highlighted in red.

---

> ### Author Response · Authors · 2024-11-20
> **(3/N) Author Rebuttal**
>
> > 3) The construction of global synthetic data requires the uploading of local graph nodes, which not only brings extra communication costs but also causes privacy leakage.
>
> Thank you for highlighting the communication costs and privacy concerns regarding the global synthetic data.
>
> __1) Regarding the extra communication costs,__
>
> as shown in Appendix A.3, FedLoG introduces only a minimal increase in overhead by uploading synthetic nodes instead of local graph nodes. Specifically, for the Cora dataset with a 3-client setting and 20 synthetic nodes per class, the synthetic nodes account for only 14% of the total communication overhead. In addition, FedLoG demonstrates robust performance on Seen Graph settings, maintaining competitiveness even with a small number of synthetic nodes (e.g., 1, 5, or 10 nodes per class), as shown in Figure 9.
>
> __2) Regarding privacy concerns,__
>
> we would like to first emphasize that FedLoG does not upload local graph nodes, but rather upload synthetic nodes that have a different feature distribution from the original nodes, as shown in Section 6 (Q2). This difference arises because, while both synthetic nodes and original nodes share the same graph encoder and embedding space, the information reflected in their features fundamentally differs.
>
> Original nodes are associated with explicit graph structures, and their features represent only the attributes of the nodes themselves, as the structural information is handled separately through the explicit graph structure. In contrast, synthetic nodes lack explicit graph structures, which forces their learnable features to encode generalized class-level or aggregated information. This distinction naturally results in a feature distribution for synthetic nodes that differs from that of original nodes as shown in Figure 5.
>
> Consequently, even in cases where a class has very few nodes, the synthetic nodes do not directly replicate or reveal the raw features of those nodes, thereby ensuring privacy and preventing information leakage. Furthermore, additional training with global synthetic data makes it increasingly difficult to identify or isolate the very few nodes within certain classes, further enhancing privacy protection.
>
> Last but not least, it is important to note that the synthetic data shared with the server contains only partial information about the original data, as we share only the synthetic data generated by the head branch. Even when the target node has a head degree, its prediction is collaboratively influenced by both the head and tail branches. This means that part of the target node’s information is distilled into the synthetic node in the tail branch, rather than being fully captured by the head branch. This design choice further mitigates privacy risks, as the synthetic data shared with the server inherently contains a limited representation of the original node's information.

---

> > ### Comment · Reviewer_sNMQ · 2024-11-22
> >
> > Thanks for the response. I still have a concern about the privacy. The paper says that the global synthetic data is obtained by merging the head nodes $V_{k,head}$ from all K clients. Thus, my question is that whether the head nodes $V_{k,head}$ cause privacy leakage?

---

> > > ### Author Response · Authors · 2024-11-22
> > >
> > > Thank you for your response and for raising your concerns!
> > >
> > > Firstly, we would like to clarify that **$\mathcal{V}_{\text{k,head}}$ does not represent the actual head-nodes within the clients**. As described in Section 4.1 (Local Fitting), **$\mathcal{V}_{\text{k,head}}$ refers to the learnable synthetic nodes that condense knowledge from nodes in a distinct feature space**.
> > >
> > > Consequently, the "global synthetic data is derived by merging the synthetic nodes $\mathcal{V}_{\text{k,head}}$, which are generated from the head branch, rather than from the real head nodes within the clients".
> > >
> > > If there are concerns regarding potential privacy leakage from sharing these synthetic nodes, we refer you to Section 6 of our paper (Privacy Analysis - Q2 and Q3) as well as our rebuttal addressing Weakness 3. For further clarity, we summarize these points below:
> > >
> > > ---
> > >
> > > **1. Synthetic Nodes $\mathcal{V}_{\text{k,head}}$ Have a Distinct Feature Distribution (Section 6, Q2)**
> > >
> > > As shown in Figure 5, synthetic nodes $\mathcal{V}_{\text{k,head}}$ exhibit a distinct feature distribution compared to the original nodes due to their different embedding mechanism. This distinction ensures that sensitive information, even for classes with very few original nodes, is not exposed.
> > >
> > > **2. Synthetic Nodes $\mathcal{V}_{\text{k,head}}$ Contain Only Partial Information (Rebuttal, W2)**
> > >
> > > Synthetic nodes $\mathcal{V}_{\text{k,head}}$ capture only partial information from the original nodes. Head node information is distributed across both the head and tail branches, ensuring that no single synthetic node exposes the full characteristics of the original data. This separation enhances privacy further.
> > >
> > > **3. Additional Training with Global Synthetic Data Adds Protection (Section 6, Q3)**
> > >
> > > Synthetic nodes $\mathcal{V}_{\text{k,head}}$ undergo additional training with global synthetic data, which introduces noise and randomness rather than solely relying on local node information. This process significantly reduces the risk of inferring specific original data points or launching effective gradient-based attacks.
> > >
> > > **4. Feature Scaling Factors Are Never Shared with the Server (Section 6, Q3)**
> > >
> > > Feature scaling, applied during local training, is never shared with the server. Without this information, adversaries cannot determine what global synthetic data was trained in each local client, preventing reconstruction of the original data from the shared synthetic nodes or gradients.
> > >
> > >
> > > Furthermore, if we aim for more rigid privacy enhancement beyond the current measures, we can additionally consider the following:
> > >
> > > **1. Adding Gaussian Noise to the Class Distribution (Section 6, Q1)**
> > >
> > > To further protect privacy, Gaussian noise can be added to the class distribution shared between clients. By introducing randomness, this ensures that even if an adversary attempts to infer sensitive information from the class distribution, the noise prevents accurate reconstruction of the true distribution.
> > >
> > > ---
> > >
> > > If you have any further concerns, we would be happy to discuss them with you!

---

> > > > ### Comment · Reviewer_sNMQ · 2024-11-23
> > > >
> > > > Dear authors,
> > > >
> > > > Thanks for the response. The responses address most of my concerns. Therefore, I decided to increase my score. On the other hand, I believe this paper can be improved better. First, the method incurs extra computational costs for updating synthetic nodes and extra communication costs for exchanging synthetic nodes. Second, the method of uploading synthetic nodes may still have a privacy risk even though not the original nodes are uploaded. As shown in Figure 5, the real nodes appear to be an average of synthetic nodes. Besides, I consider the description of the method may be not clear enough, which may arise from the fact that the method is too complex while many details are deferred to the appendix or too simplified.
> > > >
> > > > Thanks

---

> > > > > ### Author Response · Authors · 2024-11-23
> > > > >
> > > > > Dear Reviewer sNMQ,
> > > > >
> > > > > We sincerely appreciate your thoughtful feedback and constructive suggestions.
> > > > >
> > > > > Your points regarding computational and communication costs, as well as potential privacy risks associated with synthetic nodes, are well noted. We will explore further optimizations and privacy measures, such as stricter differential privacy mechanisms, in future work. Additionally, we will ensure key details are effectively highlighted in the main text rather than deferred to the appendix.
> > > > >
> > > > > Thank you once again for your valuable insights, which will undoubtedly help us refine and strengthen our work.

---

> ### Author Response · Authors · 2024-11-20
> **(4/N) Author Rebuttal**
>
> > 4. This technique seems to apply a mixup of data in FL, which has been used in many previous works, e.g., [1,2]. The novelty may be limited.
>
> Thank you for your valuable feedback regarding the novelty of our method.
>
> MixUp-based synthetic data generation methods [1,2,3,4] augment training datasets by mixing data samples with privacy-preserving techniques. However, these approaches operate in the raw feature space, which poses significant risks of privacy leakage, particularly when the local data size is small.
>
> Key distinctions of our approach compared to existing synthetic-based methods are as follows:
>
> __No reliance on raw features.__
>
> MixUp-based  [1,2,3,4] methods generate synthetic data by augmenting data at the raw feature level of the input, which can lead to privacy leakage, particularly when the original data is limited. In contrast, our synthetic data has a distinct feature distribution from the original data, arising from differences in the embedding approach used for synthetic and original data, particularly due to the presence or absence of the explicit structure.
>
> Furthermore, our method leverages not only the original data but also global synthetic data as an additional training set for condensation. This design significantly reduces privacy risks, especially when the local data size is small. Moreover, we only share a subset of the synthetic data (i.e., synthetic data within the head-degree branch), which not only excludes complete information about the local data to enhance privacy but is also specifically designed to capture reliable information relevant to the graph domain.
>
> __Handling the local overfitting problem.__
>
> Our method effectively addresses the local overfitting problem, which is one of the most challenging issues in federated learning. Local overfitting occurs after a few local updates with the distributed global model, causing the local model to severely struggle in predicting unseen data that involves unseen distributions, particularly for missing classes. MixUp-based approaches  [1,2,3,4]  still depend on local data for augmenting the training set, which limits their ability to generate data for missing classes. In contrast, our method generates global synthetic data even in scenarios where local data for certain classes is completely absent. This is achieved without relying on raw feature-based MixUp, ensuring both privacy and flexibility.
>
> __Optimizing training with synthetic data.__
>
> We extend beyond the generation of synthetic data by investigating how to train it effectively. Since our synthetic data has a different feature distribution from the original data but is utilized as training data (i.e., local generalization), it is essential to explore how to optimize the model training process with global synthetic data. To address this, we propose the Feature Scaling and Prompt Generator phases, as detailed in Section 4.3., to minimize the training-effect gap between original nodes and synthetic nodes.
>
> Including MixUp-based methods in the related works will further clarify our contributions in comparison to these methods. We appreciate your suggestion and will ensure that this distinction is made clear in our discussion. Thank you for pointing this out.
>
> You can refer to the updated PDF in Appendix A.6.2  where the revisions are highlighted in red.
>
> ---
>
> [1] Yoon, Tehrim, et al. "FedMix: Approximation of Mixup under Mean Augmented Federated Learning." International Conference on Learning Representations.
> [2] You, Xianyao, et al. "FedMDO: Privacy-preserving Federated Learning via Mixup Differential Objective." IEEE Transactions on Circuits and Systems for Video Technology (2024).
> [3] Oh, Seungeun, et al. "Mix2FLD: Downlink federated learning after uplink federated distillation with two-way mixup." IEEE Communications Letters 24.10 (2020): 2211-2215.
> [4] Shin, MyungJae, et al. "Xor mixup: Privacy-preserving data augmentation for one-shot federated learning." arXiv preprint arXiv:2006.05148 (2020).

---

> ### Author Response · Authors · 2024-11-20
> **(N/N) Author Rebuttal**
>
> > 5. Some key steps should be included in the maintext. For example, how to generate the edges for the synthetic data is required to be specified in the main text instead of the appendix.
>
> Thank you for your valuable feedback. To provide more clarity, we would like to note that each synthetic data point generates one prompt node through the prompt generator, using the synthetic data features as input. These prompt nodes are then simply connected to their corresponding synthetic data nodes, as described in lines 355–360. We will ensure to include the related details into the main text to further enhance clarity and understanding.
>
> You can refer to the updated PDF in lines 358-359 where the revisions are highlighted in red.
>
> ---
>
> > 6. In fact, this paper trains personalized models for each client since the learnable nodes and prototypes are customized for each client. Therefore, the personalization baselines should also be included [3].
>
> Thank you for pointing this out.
>
> Please note that FedPUB [1] is included as a representative and state-of-the-art baseline for personalized models. We would like to emphasize that personalized models often suffer from local overfitting, as they optimize for the local training label distribution. While these models may perform well on Seen Graph settings (Table 1), where the evaluation graph has a similar distribution to the training data, they struggle significantly with unseen data—particularly for missing classes—as shown in Table 2(b). This is because they tend to overfit to the local training distribution.
>
> Regarding FedStar, which you suggested adding as an additional personalized baseline, it locally trains a model specific to the node features of each graph while sharing the structural model across clients. Structural features in FedStar are not influenced by local graph features; instead, they focus on common structural patterns across graphs. Although FedStar was originally designed for graph-level classification tasks, we adapted it for node-level classification by removing the pooling layer. As you can see in the results below, FedStar also struggles with unseen data settings (i.e., UN, MC and NC), which is the main challenge our paper aims to address.
>
> | Dataset         | Cora     |          |          |          | CiteSeer  |          |          |          | PubMed   |          |          |          | Amazon Photo |          |          |          | Amazon Computers |          |          |          |
> |-----------------|----------|----------|----------|----------|-----------|----------|----------|----------|----------|----------|----------|----------|--------------|----------|----------|----------|------------------|----------|----------|----------|
> | Model           | SG       | UN       | MC       | NC       | SG        | UN       | MC       | NC       | SG       | UN       | MC       | NC       | SG           | UN       | MC       | NC       | SG               | UN       | MC       | NC       |
> | FedPUB          | 0.8476   | 0.5529   | 0.0000   | 0.3990   | 0.7455    | 0.6798   | 0.0012   | 0.3929   | 0.9064   | 0.8878   | 0.0000   | 0.4112   | 0.9399       | 0.4085   | 0.0000   | 0.3171   | 0.8339           | 0.4414   | 0.0000   | 0.5244   |
> | FedStar         | 0.8113   | 0.2180   | 0.0000   | 0.2047   | 0.7268    | 0.5283   | 0.0224   | 0.1906   | 0.8949   | 0.8541   | 0.0000   | 0.3944   | 0.9243       | 0.1779   | 0.0000   | 0.0355   | 0.8914           | 0.2394   | 0.0000   | 0.2671   |
> | **FedLoG**      | **0.8601** | **0.7341** | **0.6472** | **0.5047** | **0.7663** | **0.7624** | **0.6142** | **0.5973** | **0.9180** | **0.9044** | **0.1070** | **0.6053** | **0.9653** | **0.7065** | **0.4795** | **0.5605** | **0.9073** | **0.7677** | **0.3580** | **0.7386** |
>
> *SG: Seen Graph, UN: Unseen Node, MC: Missing Class, NC: New Client
>
> Including FedStar as an additional personalized model alongside FedPUB would help reinforce our main objective, highlighting how existing methods struggle with local overfitting and fail to generalize to unseen data that differ in distribution from the training environment. In contrast, our method effectively addresses both the Seen Graph and Unseen Data settings. Thank you for the suggestion.
>
> [1] Baek, Jinheon, et al. "Personalized subgraph federated learning." ICML, 2023.

---

### Official Review · Reviewer_1kKw · 2024-11-04

**Soundness:** 3
**Presentation:** 3
**Contribution:** 3
**Rating:** 8
**Confidence:** 3

**Summary:**

This paper studies the subgraph federated learning problem, where the author hopes to generate a global dataset that includes all categories, with s nodes for each category. The goal is to train a node classifier, and the optimization objective is to minimize the total empirical risk of all clients. Each client k's loss function includes both local data loss and global synthetic data loss. This approach allows each client to generalize across all categories, even if those categories are not present in their current client base. The author first discusses criteria for reliable data judgment and then proposes FedLog based on this criterion. FedLog consists of three steps: local training by clients, global aggregation by servers and data generation, followed by further server-side training. Through these three steps, the author solves the local overfitting problem in subgraph federated learning and conducts comprehensive experiments in three scenarios: Unseen Node, Missing Class and New Client.

**Strengths:**

In terms of methodology, the design of Feature scaling and prompt generator is very creative. Especially the introduction of the prompt generator can provide a graph structure for the generated nodes, which is a valuable idea.

In terms of writing, the content of the paper is introduced in detail, and both writing and presentation are professional.

In terms of experiments, the experimental design is comprehensive and solid, including multiple verification scenarios, and provides sufficient details to ensure reproducibility.

**Weaknesses:**

The discussion about 4.2.1 is not entirely convincing to me, even though it implicitly contains structural information, is this enough to express complex graph structures? Are there other methods of synthetic data in the Graph Learning field currently, such as graph compression, graph distillation techniques?

Considering that the current method is essentially an aggregation of features and has little to do with the graph structure, this feature aggregation approach is also common in federated learning for general classification tasks. Therefore, the paper should supplement discussions on data synthesis-based methods in the federated learning field in the related work section.

This paper chooses nodes with high degree centrality and class centrality for global data synthesis. How can we ensure that each category can generate $s$ nodes? If a certain category is a tail class or missing class among all clients, how can we guarantee generating high-quality s nodes for this category?

**Questions:**

The loss function of each client includes the loss of local data and the loss of global synthetic data. Does this design exist in previous methods based on data synthesis? Is it a standard practice?

Regarding the selection of reliability data, are there any other measurement standards besides degree and class frequency? Are there any other works with high recognition in this area?

---

> ### Author Response · Authors · 2024-11-20
> **(1/N) Author Rebuttal**
>
> Thank you for your valuable feedback and for taking the time to provide thorough reviews!
>
> > The discussion about 4.2.1 is not entirely convincing to me, even though it implicitly contains structural information, is this enough to express complex graph structures? Are there other methods of synthetic data in the Graph Learning field currently, such as graph compression, graph distillation techniques?
>
> __Regarding how the synthetic features imply graph structural information,__
>
> as mentioned in Section 4.2.1, both synthetic nodes and original nodes share the same graph encoder and embedding space. However, the information reflected in their features differs because only the original nodes have explicit graph structures. When embedding nodes with explicit graph structures, the process involves two phases: (1) feature transformation ($Z=XW$) and (2) neighbor aggregation ($AZ$), where the explicit structure is directly used.
>
> In contrast, synthetic nodes undergo only the feature transformation phase without neighbor aggregation. Consequently, the learnable features of synthetic nodes must distill aggregated information to align with the embedding space of the original nodes and act as prototypes. The reason we state that the features of synthetic nodes “implicitly” capture the original graph structure is that their features reflect the "aggregated information" of L-hop neighbors (where L denotes the number of GNN layers) rather than explicitly representing complex graph structures.
>
> In other words, for synthetic nodes in the head branch, the L-hop neighbor’s aggregated information of head-degree nodes is distilled into their features, while the same applies to tail-degree nodes in the tail branch.
>
> __Regarding other methods of synthetic data in the graph learning field, such as graph compression and graph distillation,__
>
> GCond [1], SGDD [2], and SFGC [3] are representative graph condensation methods. These approaches aim to condense original graph data into a compressed form by leveraging techniques such as gradient matching or training trajectory matching.
>
> GCond [1] is the first method to perform graph condensation by matching gradients between the original graph and the condensed graph. Although GCond relies on features to encode condensed information, it constructs the graph structure from these condensed features rather than explicitly using the original structure. SGDD [2] builds on GCond by explicitly broadcasting the original graph structure to reinforce the distillation of structural information. In contrast, SFGC [3] adopts a structure-free approach, condensing the original graph data into structure-free data using training trajectory matching.
>
> These feature-based condensation methods, such as GCond [1] and SFGC [3], demonstrate that structural information can be implicitly captured through features while still achieving competitive performance. Inspired by this, we propose condensing structural-free synthetic data using the prototypical architecture, emphasizing that implicitly representing structural information in features can remain highly effective.
>
> ---
>
> [1] Jin, Wei, et al. "Graph Condensation for Graph Neural Networks." International Conference on Learning Representations.
> [2] Yang, Beining, et al. "Does graph distillation see like vision dataset counterpart?." Advances in Neural Information Processing Systems 36 (2024).
> [3] Zheng, Xin, et al. "Structure-free graph condensation: From large-scale graphs to condensed graph-free data." Advances in Neural Information Processing Systems 36 (2024).

---

> ### Author Response · Authors · 2024-11-20
> **(2/N) Author Rebuttal**
>
> > Considering that the current method is essentially an aggregation of features and has little to do with the graph structure, this feature aggregation approach is also common in federated learning for general classification tasks. Therefore, the paper should supplement discussions on data synthesis-based methods in the federated learning field in the related work section.
>
> Thank you for your insightful feedback. We will ensure to include supplementary discussions on data synthesis-based methods in federated learning in the related work section.
>
> Generating synthetic data using aggregated knowledge from clients has emerged as a promising approach to compensate for the limitations of local training data. This method facilitates data augmentation while addressing challenges such as class imbalance and limited data availability.
>
> MixUp-based synthetic data generation methods [4,5,6,7] augment training datasets by mixing data samples with privacy-preserving techniques. However, these approaches operate in the raw feature space, which poses significant risks of privacy leakage, particularly when the local data size is small.
>
> Alternatively, generative adversarial network (GAN)-based methods, such as FedGAN [8] and FedDPGAN [9], leverage generative models to create synthetic data. These methods aim to generalize a global generator to produce synthetic data that can mitigate data imbalance while preserving privacy. However, they incur high computational costs, which limits their practical applicability.
>
> Recently, condensation-based methods [10,11,12] have been proposed to alleviate the impact of data heterogeneity. FedDC [10] condenses synthetic data based on local data and fine-tunes the global model at the server level to ensure stable convergence. FedMK [11] generates synthetic data by condensing private data into meta-knowledge, which is used as an additional training set to accelerate convergence. FedAF [12] introduces an aggregation-free paradigm, where the server directly trains the global model using condensed synthetic data.
>
> Key distinctions of our approach compared to existing synthetic-based methods are as follows:
>
> __No reliance on raw features.__
>
> MixUp-based  [4,5,6,7] and condensation-based [10,11,12] methods generate synthetic data by augmenting or condensing data at the raw feature level of the input, which can lead to privacy leakage, particularly when the original data is limited. In contrast, our synthetic data has a distinct feature distribution from the original data, arising from differences in the embedding approach used for synthetic and original data, particularly due to the presence or absence of the explicit structure. Furthermore, our method leverages not only the original data but also global synthetic data as an additional training set for condensation. This design significantly reduces privacy risks, especially when the local data size is small. Moreover, we only share a subset of the synthetic data (i.e., synthetic data within the head-degree branch), which not only excludes complete information about the local data to enhance privacy but is also specifically designed to capture reliable information relevant to the graph domain.
>
> __Handling the local overfitting problem.__
>
> Our method effectively addresses the local overfitting problem, which is one of the most challenging issues in federated learning. Local overfitting occurs after a few local updates with the distributed global model, causing the local model to severely struggle in predicting unseen data that involves unseen distributions, particularly for missing classes. MixUp-based approaches  [4,5,6,7]  still depend on local data for augmenting the training set, which limits their ability to generate data for missing classes. In contrast, our method generates global synthetic data even in scenarios where local data for certain classes is completely absent. This is achieved without relying on raw feature-based MixUp, ensuring both privacy and flexibility.
>
> __Optimizing training with synthetic data.__
>
> We extend beyond the generation of synthetic data by investigating how to train it effectively. Since our synthetic data has a different feature distribution from the original data but is utilized as training data (i.e., local generalization), it is essential to explore how to optimize the model training process with global synthetic data. To address this, we propose the Feature Scaling and Prompt Generator phases, as detailed in Section 4.3., to minimize the training-effect gap between original nodes and synthetic nodes.
>
> You can refer to the updated PDF in Appendix A.6.2  where the revisions are highlighted in red.

---

> ### Author Response · Authors · 2024-11-20
> **(3/N) Author Rebuttal**
>
> [4] Yoon, Tehrim, et al. "FedMix: Approximation of Mixup under Mean Augmented Federated Learning." International Conference on Learning Representations.
> [5] You, Xianyao, et al. "FedMDO: Privacy-preserving Federated Learning via Mixup Differential Objective.", 2024.
> [6] Oh, Seungeun, et al. "Mix2FLD: Downlink federated learning after uplink federated distillation with two-way mixup., 2020
> [7] Shin, MyungJae, et al. "Xor mixup: Privacy-preserving data augmentation for one-shot federated learning.", 2020.
> [8] Rasouli, et al. "Fedgan: Federated generative adversarial networks for distributed data.", 2020.
> [9] Zhang, Longling, et al. "FedDPGAN: federated differentially private generative adversarial networks framework for the detection of COVID-19 pneumonia." Information Systems Frontiers 23.6 (2021): 1403-1415.
> [10] Kim, et al. "Stable Federated Learning with Dataset Condensation." J. Comput. Sci. Eng. 16.1 (2022): 52-62.
> [11] Liu, Ping, et al. "Meta knowledge condensation for federated learning." arXiv preprint arXiv:2209.14851 (2022).
> [12] Wang, Yuan, et al. "An aggregation-free federated learning for tackling data heterogeneity." Proceedings of the IEEE/CVF Conference on Computer Vision and Pattern Recognition. 2024.
>
> ---
>
> > This paper chooses nodes with high degree centrality and class centrality for global data synthesis. How can we ensure that each category can generate s nodes? If a certain category is a tail class or missing class among all clients, how can we guarantee generating high-quality s nodes for this category?
>
> This is an excellent point for further discussion, and we sincerely appreciate you bringing it to our attention.
>
> If a certain category is a tail class or missing class across all clients, constructing high-quality synthetic nodes becomes particularly challenging due to the insufficient information available for that category across clients. While the quality of the global synthetic data for such categories may be relatively lower compared to head classes, we describe how our method addresses and alleviates this challenging scenario.
>
> __When a certain category is a tail class across all clients:__
>
> - __Server-Level Aggregation:__ At the server level, the global synthetic data for this tail class will be aggregated nearly uniformly across clients since the proportion of this class is consistently low for all clients. This ensures that our method captures and aggregates information for the common tail class from a broader, global perspective.
>
> - __Client-Level Aggregation:__ At the client level, the model focuses on aggregating tail class knowledge from head-degree nodes, which are more reliable for sharing with other clients. Given the lack of sufficient training data for the tail class, which makes it more susceptible to noise, generating and sharing reliable knowledge becomes even more critical.
>
> For more clarity, we evaluate the performance of the methods under a scenario where all clients share the same tail categories. To simulate this, we partition the global graph using a Dirichlet distribution, adjusting the alpha value (distinct from the alpha used in our paper to weight predictions across branches).
>
> In the context of the Dirichlet distribution, the alpha value determines the level of balance in class distributions among the partitions:
>
> - A small alpha value results in highly imbalanced class distributions.
> - A large alpha value leads to more balanced class distributions, causing the tail categories to be shared across clients.
>
> |  **[Table R1]**      $\alpha=$         | 0.05  | 0.1   | 0.5   | 1.0   | 10    | 100   | 1000  |
> |----------------------|-------|-------|-------|-------|-------|-------|-------|
> | **FedAvg**           | 0.9083| 0.7965| 0.9179| 0.8525| 0.7992| 0.8308| 0.8235|
> | **FedSAGE+**         | 0.6347| 0.8102| 0.7197| 0.8508| 0.8166| 0.8213| 0.8392|
> | **FedGCN**           | 0.8083| 0.8116| 0.8093| 0.8103| 0.8087| 0.8080| 0.8070|
> | **FedPUB**           | 0.9126| 0.8200| 0.9197| **0.8902**| 0.8031| 0.8422| 0.8353|
> | **FedNTD**           | 0.8711| 0.8180| 0.9094| 0.8852| 0.8108| 0.8403| 0.8333|
> | **FedED**            | 0.8768| 0.8082| 0.9026| 0.8820| 0.8012| 0.8194| 0.8294|
> | **FedLoG**           | **0.9255**| **0.8219**| **0.9231**| 0.8869| **0.8301**| **0.8498**| **0.8569**|
>
> As shown in Table R1, FedLoG consistently outperforms the baselines even when all clients share the same tail class (e.g., α≥10). This demonstrates that FedLoG effectively distills reliable knowledge from each local distribution into the synthetic data, maintaining robust performance even in challenging scenarios.
>
> __When a certain category is a missing class across all clients:__
>
> it becomes impossible to generate global synthetic data for that class, as there is no available information to support its generation. Please note that the primary purpose of federated learning is to collaboratively construct global knowledge from distributed client data.

---

> ### Author Response · Authors · 2024-11-20
> **(N/N) Author Rebuttal**
>
> > The loss function of each client includes the loss of local data and the loss of global synthetic data. Does this design exist in previous methods based on data synthesis? Is it a standard practice?
>
> Thank you for pointing out this detailed aspect.
>
> Regarding previous data synthesis-based methods [4][5][13], these methods also have global synthetic data in their loss functions, typically by combining synthetic data with local training data [13] or by applying mixup between synthetic data and local data [4][5]. Although our approach could also be categorized as a method that combines local data and synthetic data for training, similar to [13], the key distinction lies in the sequencing of training. Specifically, FedLoG trains the global synthetic data only after completing the training on local data.
>
> This design choice is motivated by the goal of addressing the local overfitting problem, which arises when local models are trained exclusively on their own local data, starting from a model initialized by the global model. To mitigate this issue, FedLoG adaptively customizes the global synthetic data based on the states of the local models, which are overfitted to their local training data. By leveraging global synthetic data in this way, FedLoG effectively alleviates local overfitting and ensures better generalization across clients.
>
> ---
>
> [4] Yoon, Tehrim, et al. "FedMix: Approximation of Mixup under Mean Augmented Federated Learning." International Conference on Learning Representations.
> [5] You, Xianyao, et al. "FedMDO: Privacy-preserving Federated Learning via Mixup Differential Objective." IEEE Transactions on Circuits and Systems for Video Technology (2024).
> [13] Chen, Huancheng, and Haris Vikalo. "Federated learning in non-iid settings aided by differentially private synthetic data." Proceedings of the IEEE/CVF Conference on Computer Vision and Pattern Recognition. 2023.
>
> ---
>
> > Regarding the selection of reliability data, are there any other measurement standards besides degree and class frequency? Are there any other works with high recognition in this area?
>
> Thank you for pointing out this interesting discussion point.
>
> It is important to clarify that the term “reliability of data” is not a commonly established concept but rather a term we specifically define in our paper. As stated in Section 3.1, we use this term to refer to the accuracy and consistency of information derived from decentralized nodes. While this definition provides the foundation for our work, we acknowledge that exploring other metrics to evaluate reliability in graph federated learning could be an interesting direction for future research.
>
> Our definition and motivation are guided by prior studies analyzing the informativeness or noisiness within graph data. From the perspective of node degree, [14] offers theoretical and empirical insights into degree bias in graph neural networks, showing that tail-degree nodes tend to carry less informative features and are more prone to misclassification. From the perspective of class distribution, statistical learning theory [15] demonstrates that the size of the training data strongly correlates with generalization performance. Additionally, [16] empirically shows that GNN models generalize better for head classes.
>
> Inspired by these studies, we hypothesized that data that is harder to generalize may be less reliable, and we validated this hypothesis through the experiments described in Section 3.1 in the context of federated learning.
>
> We also recognize the value of exploring other metrics for reliability beyond degree and class frequency, such as measures related to graph topology or feature distributions. This could serve as a valuable discussion point for future research. Thank you for raising this insightful perspective!
>
> [14] Subramonian, Arjun, Jian Kang, and Yizhou Sun. "Theoretical and Empirical Insights into the Origins of Degree Bias in Graph Neural Networks." arXiv preprint arXiv:2404.03139 (2024).
> [15] Vapnik, Vladimir. The nature of statistical learning theory. Springer science & business media, 2013.
> [16] Yun, Sukwon, et al. "Lte4g: Long-tail experts for graph neural networks." Proceedings of the 31st ACM International Conference on Information & Knowledge Management. 2022.

---

> > ### Comment · Reviewer_1kKw · 2024-11-25
> >
> > Thanks to the author's detail response, which clarified my concerns and proposed many related works. I have no other questions, so I decided to increase my score.

---

> > > ### Author Response · Authors · 2024-11-25
> > >
> > > We are pleased to hear that your concerns have been addressed. Thank you for acknowledging our efforts!

---

### Official Review · Reviewer_sNpJ · 2024-11-04

**Soundness:** 3
**Presentation:** 3
**Contribution:** 2
**Rating:** 6
**Confidence:** 4

**Summary:**

In this article, the proposed method, FedLoG, addresses this issue by mitigating local overfitting through global synthetic data generation, condensing reliable structural information and class representations from clients. By training on these synthetic data, FedLoG adapts to the knowledge gaps in each local dataset, enhancing local models' generalization capabilities to handle diverse, unseen data. Experimental results confirm FedLoG’s superiority in generalization over existing baselines in realistic settings.

**Strengths:**

As the paper deals with Federated learning, the major focus should be on what and which kind of data and knowledge is aggregated to train the global model. Based on this, there are following strengths observed:
1. The paper posses a good motivation and a limited but fair novelty.
2. The baselines considered are fair enough and the gain is significant.
3. The organization of the paper is quiet good.
4. the methodology containing local and global fitting with aggregations is also fairly novel.

**Weaknesses:**

1. The novelty is fair but also limited. As it looks like an ensemble of multiple techniques used in earlier literature.
2. The process contains mutliple steps which may result in overhead in terms of computation and storage.
3. There may be privacy issues while doing such proposed aggregation.
4. If any client will be having data bias, it may propagate to the global from local model as it is taking average.

**Questions:**

1. How can it be handeling the privacy concerns?
2. What complexity it possess in terms of time with respect to the other baselines? Will it also be significant or just comparable?
3. Comment 4 on weekness. How can the authors see that issue?

---

> ### Author Response · Authors · 2024-11-20
> **(1/N) Author Rebuttal**
>
> Thank you for taking the time for such thorough reviews!
>
> > W1. The novelty is fair but also limited. As it looks like an ensemble of multiple techniques used in earlier literature.
>
> Our work focuses on tackling the critical issue of local overfitting in graph federated learning (Graph FL), a challenge that has not been addressed in prior research. Graph FL is inherently unique due to properties such as node degree, structure, and the mutable nature of graph data. To address these challenges effectively, we developed an architecture specifically designed to mitigate local overfitting and ensure robust performance across diverse clients.
>
> We structured our approach around the following key considerations:
>
> __1) Selecting Reliable Data for Synthesis__
> We define reliability in federated learning as the accuracy and consistency of decentralized data, as outlined in Section 3.1. Through analysis, we found that head-class and head-degree nodes provide more reliable information for other clients. Our architecture is specifically designed to distill and utilize this critical knowledge.
>
> __2) Effective Knowledge Extraction__
> To extract and condense this reliable knowledge, we proposed a prototypical network—an approach not previously explored in graph condensation. Additionally, we implemented degree-dependent weighting to adaptively guide gradient flows, ensuring that head-degree knowledge is effectively distilled into synthetic nodes. We further introduced a class-rate-dependent aggregation strategy to enhance the quality of global synthetic data.
>
> __3) Privacy Protection of Synthetic Nodes__
> Synthetic nodes fundamentally differ from original nodes in their feature distribution, as described in Section 6 (Q2). While sharing the same embedding space, original nodes are embedded using explicit graph structures (i.e., $AXW$), while synthetic nodes are encoded solely from their features ($XW$). This distinction ensures that the learnable features of synthetic nodes do not directly reveal the raw features of original nodes, even for classes with very few nodes. As a result, this approach enhances privacy and effectively mitigates information leakage risks.
>
> __4) Optimizing Training Beyond Data Generation__
> Synthetic data differs in feature distribution from original data but must still be utilized effectively for training. To bridge this gap, we proposed Feature Scaling and Prompt Generator phases (Section 4.3), which minimize the training-effect discrepancy between synthetic and original nodes. This ensures effective local generalization and enhances the utility of synthetic data.
>
> In summary, while some aspects of our methodology draw inspiration from existing techniques, our work is guided by the motivation to address a challenging and meaningful problem. The design of our model reflects careful consideration of the unique complexities of Graph FL, guided by the core motivations of this research. We hope these considerations are recognized as a thoughtful effort to provide a practical solution that contributes to advancing this field.

---

> ### Author Response · Authors · 2024-11-20
> **(2/N) Author Rebuttal**
>
> > W2. The process contains mutliple steps which may result in overhead in terms of computation and storage.
>
> Thank you for your valuable feedback.
>
> **1) Regarding the time complexities,**
>
> __We would like to emphasize that each classifier, including the Prompt Generator and Feature Scaling modules, has the same time complexity as a standard MLP.__  Specifically, the time complexity of our classifiers is $O(d^2)$, which matches that of an MLP model. Below, we provide the detailed calculations for each module:
>
> - **Pairwise Distance between Prototypes (Equation (7))**:
>   The naive time complexity is $O(PF)$, where $P$ is the number of prototypes ($|C_v| \times s$) and $F$ is the dimension of the input ($2 \times d$, where $d$ denotes the dimension of the representation $h_v$). Since $P$ is small enough to be negligible, the complexity becomes $O(d)$.
>
> - **Distance-Based Message Generation (Equation (6))**:
>   The naive time complexity is $O(PF^2)$, where $F$ is the input dimension ($2d + 1$). With $P$ being negligible, this results in a complexity of $O(d^2)$, which is the same as that of an MLP.
>
> - **Updating the Target Node’s Representation (Equations (8-10))**:
>   Equation (8) includes an MLP and elementwise operations with subtraction, giving a total complexity of $O(d^2)$. Equations (9) and (10) involve only simple additions and are therefore negligible. Thus, the total time complexity for this update is $O(d^2)$.
>
> - **Prompt Generator**:
>   In the pretraining phase, the prompt generator requires $O(|V|d)$ complexity for Equation (11), and $O(|\mathcal{E}|d + |V|d^2)$ for Equation (12). Therefore, the total time complexity of the prompt generator is $O(|\mathcal{E}|d + |V|d^2)$, which is the same as the GNN encoder. However, it is worth noting that, during inference, it requires only $O(d^2)$, which has the same time complexity as the MLP.
>
> - **Feature Scaling**:
>  For feature scaling in Equation (5), the time complexity is $O(d)$ since the operation involves only simple element-wise additions.
>
> Consequently, the total time complexity of the classifiers, including the Prompt Generator and Feature Scaling, is $O(d^2)$. This is significantly lighter than the complexity of the GNN encoder, $O(|\mathcal{E}|d + |V|d^2)$, where $|\mathcal{E}|$ and $|V|$ denote the number of edges and nodes, respectively.
>
> - **Graph Encoder**:
>   As we utilize GraphSAGE for the GNN encoder, it requires $O(|\mathcal{E}|d + |V|d^2)$ for both forward and backward time.
>
> To sum up, our model requires $O(|\mathcal{E}|d + |V|d^2)$ complexity, which is the same as a GNN encoder. Again, it is worth noting that each classifier has the same time complexity as an MLP (i.e., $O(d^2)$), which has little influence on the total complexity of our architecture.
>
> You can refer to the updated PDF in Appendix A.5, where the revisions are highlighted in red.
>
> __2) Regarding communication overhead,__
>
> We have provided an analysis of communication overhead in Appendix A.3. Over 100 rounds, our method requires 0.65x to 2.57x the communication cost compared to other baselines, meaning our method does not always incur the highest communication cost. As shown in Table 6 in Appendix A.3, our method demonstrates faster convergence due to its use of reliable class representations, resulting in lower communication costs to achieve the same accuracy. For example, with the Cora dataset and 3 clients, our method achieves relatively low cost (second best) among the baselines.
>
> Thank you for pointing out the complexity concerns. We will ensure to include this complexity analysis, as it will significantly enhance the quality of our paper.

---

> ### Author Response · Authors · 2024-11-20
> **(3/N) Author Rebuttal**
>
> > W3. There may be privacy issues while doing such proposed aggregation.
>
> Thank you for raising this important concern regarding potential privacy issues during aggregation.
> As we address the potential privacy concerns in Section 6, we would like to reorganize how our method mitigates these issues:
>
> __1) Privacy Protection in Synthetic Data:__
>
> As illustrated in Figure 5, the distinct feature distribution of synthetic data compared to the original data mitigates privacy leakage, even for classes with very few instances in the original data. This distinction arises because the synthetic data do not directly distill the raw features of the original data; instead, they are embedded through a different mechanism. While original data are embedded with explicit graph structures, synthetic data rely solely on their learnable features, leading to inherently different feature distribution. Additionally, the synthetic data are further trained using a supplementary dataset (i.e., global synthetic data), making it even more challenging to trace back or infer information from the original data. This approach significantly reduces privacy risks, especially when the local data size is small.
>
> Last but not least, it is important to note that the synthetic data shared with the server contains only  information about the original data, as we share only the synthetic data generated by the head branch. Even when the target node has a head degree, its prediction is collaboratively influenced by both the head and tail branches. This means that part of the target node’s information is distilled into the synthetic node in the tail branch, rather than being fully captured by the head branch. This design choice further mitigates privacy risks, as the synthetic data shared with the server inherently contains a limited representation of the original node's information.
>
> __2) Adding Noise to the Class Rate:__
>
> As discussed in Section 6 (Q1), while class distribution only reflects the proportion of each class and does not directly include individual data, privacy concerns may still arise if the distribution conveys sensitive information about the client group. To address this, we validate that our method maintains robust performance even when noise is added to the class rates, as long as the overall trend of the distribution is preserved. This ensures that privacy risks are mitigated without compromising performance.
>
> __3) Feature Scaling and Its Impact on Gradients and Synthetic Data:__
>
> Each client undergoes additional training with globally shared synthetic data, adjusted using locally applied feature scaling. While this additional training affects both the gradients of the local model and the features of the synthetic data, the feature scaling factors themselves are never shared with the server. This design makes it significantly harder for adversaries to invert the gradients or synthetic data and reconstruct the original data, further enhancing privacy protection.
>
> ---
>
> > W4. If any client will be having data bias, it may propagate to the global from local model as it is taking average.
>
> We appreciate your insightful question.
>
> __It is worth noting that our problem settings are specifically designed to address the local overfitting problem, which often arises when local clients exhibit data bias.__ Appendix A.16 provides the data statistics for each client across all evaluation settings. These statistics reveal that almost all clients exhibit data bias, presenting significant challenges for federated learning. For example, Table 14 shows the class-wise data statistics for the PubMed dataset with 3 clients under the Seen Graph settings as follows:
>
> | Client   | Class | Train | Valid | Test |
> |:--------:|:-----:|:-----:|:-----:|:----:|
> | Client 0 |   0   |  277  |  222  |  209 |
> |          |   1   |   0   |   0   |   0  |
> |          |   2   | 1642  | 1227  | 1189 |
> | Client 1 |   0   |  994  |  761  |  740 |
> |          |   1   |   0   |   0   |   0  |
> |          |   2   |  594  |  408  |  414 |
> | Client 2 |   0   |   0   |   0   |   0  |
> |          |   1   | 1176  |  897  |  951 |
> |          |   2   |  736  |  536  |  538 |
>
>
> The head class for each client is class 2, class 0, and class 1, respectively, while the tail class for each client is class 1, class 1, and class 0, respectively. Our experimental settings are specifically designed to simulate biased local datasets, which represent both the most challenging and practical scenarios in federated learning.

---

> ### Author Response · Authors · 2024-11-20
> **(N/N) Author Rebuttal**
>
> In addition, for more clarity, we compare i.i.d. and non-i.i.d. partitioning to evaluate the impact of data distribution on FedLoG’s performance. Under i.i.d. settings, each client shares the same class distribution, while non-i.i.d. settings involve different class distributions across clients, representing severe data bias. To simulate these scenarios, we partition the global graph using a Dirichlet distribution by adjusting the alpha value (distinct from the alpha used in the paper to weight predictions across branches). In the Dirichlet distribution, a smaller alpha value creates highly imbalanced class distributions, leading to severe data bias, while a larger alpha value results in more balanced class distributions across clients. The results for the Cora dataset with 3 clients and alpha values in [0.05, 0.1, 0.5, 1.0, 10, 100, 1000] are presented in Table R1.
>
> |  **[Table R1]**      $\alpha=$         | 0.05  | 0.1   | 0.5   | 1.0   | 10    | 100   | 1000  |
> |----------------------|-------|-------|-------|-------|-------|-------|-------|
> | **FedAvg**           | 0.9083| 0.7965| 0.9179| 0.8525| 0.7992| 0.8308| 0.8235|
> | **FedSAGE+**         | 0.6347| 0.8102| 0.7197| 0.8508| 0.8166| 0.8213| 0.8392|
> | **FedGCN**           | 0.8083| 0.8116| 0.8093| 0.8103| 0.8087| 0.8080| 0.8070|
> | **FedPUB**           | 0.9126| 0.8200| 0.9197| **0.8902**| 0.8031| 0.8422| 0.8353|
> | **FedNTD**           | 0.8711| 0.8180| 0.9094| 0.8852| 0.8108| 0.8403| 0.8333|
> | **FedED**            | 0.8768| 0.8082| 0.9026| 0.8820| 0.8012| 0.8194| 0.8294|
> | **FedLoG**           | **0.9255**| **0.8219**| **0.9231**| 0.8869| **0.8301**| **0.8498**| **0.8569**|
>
> As shown in Table R1, FedLoG continues to outperform baselines from non-i.i.d. (i.e., alpha values 0.05 to 1.0) to i.i.d. (i.e., alpha values 10 to 1000) settings. We argue that FedLoG successfully distills reliable knowledge from each local distribution into the synthetic data. This allows clients to effectively learn absent knowledge through Local Generalization (Section 4.3), regardless of the balance or imbalance extent across the clients.
>
> We appreciate your insightful question and the opportunity to highlight the design of our experimental settings. By focusing on scenarios with biased local datasets, we aim to address real-world challenges and provide a meaningful contribution to advancing federated learning research. Thank you for raising this important point!
>
> ---
>
> > How can it be handeling the privacy concerns?
>
> We kindly refer you to our response to __W3__.
>
> ---
>
> > What complexity it possess in terms of time with respect to the other baselines? Will it also be significant or just comparable?
>
> We kindly refer you to our response to __W2__.
>
> ---
>
> > Comment 4 on weekness. How can the authors see that issue?
>
> We kindly refer you to our response to __W4__.

---

> ### Author Response · Authors · 2024-12-02
> **Gentle Reminder**
>
> Dear Reviewer sNpJ,
>
> We sincerely appreciate your feedback on our work. As the discussion period draws to a close, we would like to ensure that our responses have adequately clarified and addressed your concerns.
>
> If you have any remaining questions or need further clarification, we would be more than happy to assist before the deadline. Please don't hesitate to reach out with any additional concerns.

---

### Official Review · Reviewer_SvTi · 2024-11-07

**Soundness:** 3
**Presentation:** 4
**Contribution:** 3
**Rating:** 8
**Confidence:** 3

**Summary:**

Federated Learning (FL) on graphs can struggle with local overfitting due to changes in graph data and shifts in label distributions. Existing methods only focus on local data, which limits their ability to generalize to unseen data. The proposed FedLoG method addresses this by generating global synthetic data from reliable class representations and structural information across clients. This synthetic data helps prevent local overfitting, improving the generalization of local models to unseen data. Experimental results show that FedLoG outperforms existing approaches, and the code is publicly available.

**Strengths:**

- This paper is well-organized and has reasonable motivation.
- The proposed method, FedLoG, is interesting in improving local generalization from overfitting.
- The experiments are solid, and extensive results are shown to support the proposed method.
- The experimental results show that the proposed method can achieve the sota performances across different federated graph datasets against strong baselines.

**Weaknesses:**

- Since the proposed method relies on synthetic data, it is of particular interest whether FedLoG can protect data privacy. Can differential privacy techniques be compatible with FedLoG to further guarantee privacy issues?
- Some important literature about generalization in vanilla federated learning (not limited to federated graph learning) should be cited and discussed. For instance, FedRoD [1] bridges the gap between global generalization and local personalization. Recently, FedLAW [2] and FedDisco [3] target global generalization from aggregation perspectives, and FedETF [4], which is inspired by neural collapse, shows a potential way of achieving personalization and generalization at the same time.

---
[1] Chen, H. Y., & Chao, W. L. On Bridging Generic and Personalized Federated Learning for Image Classification. In International Conference on Learning Representations 2022.

[2] Li, Z., Lin, T., Shang, X., & Wu, C. (2023, July). Revisiting weighted aggregation in federated learning with neural networks. In International Conference on Machine Learning (pp. 19767-19788). PMLR.

[3] Ye, R., Xu, M., Wang, J., Xu, C., Chen, S., & Wang, Y. (2023, July). Feddisco: Federated learning with discrepancy-aware collaboration. In International Conference on Machine Learning (pp. 39879-39902). PMLR.

[4] Li, Z., Shang, X., He, R., Lin, T., & Wu, C. (2023). No fear of classifier biases: Neural collapse inspired federated learning with synthetic and fixed classifier. In Proceedings of the IEEE/CVF International Conference on Computer Vision (pp. 5319-5329).

**Questions:**

See weaknesses.

---

> ### Author Response · Authors · 2024-11-20
> **(1/N) Author Rebuttal**
>
> Thank you so much for your thorough and insightful feedback!
>
> > Since the proposed method relies on synthetic data, it is of particular interest whether FedLoG can protect data privacy. Can differential privacy techniques be compatible with FedLoG to further guarantee privacy issues?
>
> Thank you so much for your thorough and insightful feedback.
> Incorporating DP into our model could indeed enhance privacy guarantees and mitigate potential risks. As the features of synthetic nodes are learnable parameters, applying DP to the gradients of synthetic nodes during training can help prevent sensitive information from being revealed. Specifically, we can employ the additive noise differential privacy mechanism [1] , leveraging the Gaussian mechanism:
>
> $\mathcal{M}_{\text{Gauss}}(x, f, \epsilon, \delta) = f(x) + \mathcal{N}\left(\mu = 0, \sigma^2 = \frac{2 \ln(1.25 / \delta) \cdot (\Delta f)^2}{\epsilon^2}\right)$, which provides $(\epsilon, \delta)$-differential privacy.
>
> This mechanism ensures that the gradients do not inadvertently reveal individual data properties, thereby reinforcing privacy protection. For experiments, we set $\delta = 10^{-5}$ and $C = 1.0$, and compared the results with $\epsilon = 0.1$ and $\epsilon = 1.0$ (smaller $\epsilon$ provides stronger privacy protection). The results are summarized as follows (Amazon Computers dataset with 3 clients used):
>
> |  | Seen Graph | Unseen Node | Missing Class | New Client |
> |------------------------------|------------|-------------|---------------|------------|
> | FedLoG+DP ($\epsilon=0.1$)   | 0.9060     | 0.7456      | 0.1325        | 0.6936     |
> | FedLoG+DP ($\epsilon=1.0$)   | 0.9069     | 0.7616      | 0.1231        | 0.7019     |
> | FedLoG                        | __0.9073__     | __0.7677__      | __0.3580__        | __0.7386__     |
>
> We observe that for the Seen Graph and Unseen Node settings, which align more closely with the local data distribution, our method demonstrates robust performance even with the application of differential privacy (DP). However, for the more challenging Missing Class and New Client settings, which heavily rely on global synthetic data to compensate for locally absent knowledge, the introduction of DP has a noticeable impact on performance.
>
>  While we have not yet incorporated DP in our current work, we agree that applying this technique could provide additional privacy guarantees and broaden the applicability of our method in scenarios where privacy concerns are paramount. We appreciate this insightful suggestion and will explore it as part of future work to further strengthen our model's privacy-preserving capabilities.
>
> In our current model, FedLoG, only a subset of the synthetic data is shared (i.e., synthetic data from the head-degree branch). This design not only excludes comprehensive information about the local data, thereby enhancing privacy, but also focuses on capturing reliable and meaningful information tailored to the graph domain.
>
> [1] Dwork, Cynthia, and Aaron Roth. "The algorithmic foundations of differential privacy." Foundations and Trends® in Theoretical Computer Science 9.3–4 (2014): 211-407.

---

> ### Author Response · Authors · 2024-11-20
> **(N/N) Author Rebuttal**
>
> > Some important literature about generalization in vanilla federated learning (not limited to federated graph learning) should be cited and discussed. For instance, FedRoD [1] bridges the gap between global generalization and local personalization. Recently, FedLAW [2] and FedDisco [3] target global generalization from aggregation perspectives, and FedETF [4], which is inspired by neural collapse, shows a potential way of achieving personalization and generalization at the same time.
>
> Thank you for your suggestions and for highlighting important related works.
>
> Regarding [1], FedRoD bridges the gap between generic FL and personalized FL by leveraging a class-balanced loss and empirical risk minimization. While this approach improves generic FL, it depends on the presence of at least one data point for each class within each client. This reliance makes it less effective in scenarios where certain classes are entirely absent in some clients, a common challenge in federated learning (i.e., the Missing Class setting).
>
> FedETF [4] addresses classifier biases by enhancing the generalization of the global model and enabling personalized adaptation through local fine-tuning. To improve generalization, FedETF employs a balanced feature loss weighted by the number of samples in each class. However, its generalization phase does not adequately handle the Missing Class scenario, where certain classes have no samples at all. Furthermore, its reliance on local fine-tuning exacerbates the local overfitting problem, making local models more prone to overfitting their biased data distributions and struggling to generalize to unseen data, such as missing classes.
>
> FedLAW [2] enhances the generalization of global models by introducing a learnable weighted aggregation mechanism, where the L1 norm of the aggregation weights is constrained to be less than 1. Additionally, it incorporates the concept of client coherence to identify clients that positively contribute to generalization. Similarly, FedDisco [3] proposes a weighted aggregation method based on the discrepancy between local and global category distributions, further improving the performance of the global model. While both FedLAW [2] and FedDisco [3] primarily focus on enhancing the generalization of the global model, our work takes a different approach by addressing the local overfitting problem. Specifically, we aim to improve the generalization of local models, which are prone to overfitting their local data distributions after a few local updates from the global model, even when the global model itself is well generalized.
>
> Including these related works will clearly help improve the quality of our paper and highlight the significance of our proposed approach. Thank you for the valuable suggestion!
>
> You can refer to the updated PDF in Appendix A.6, where the revisions are highlighted in red.

---

> ### Author Response · Authors · 2024-12-02
> **Gentle Reminder**
>
> Dear Reviewer SvTi,
>
> We sincerely appreciate your feedback on our work. As the discussion period draws to a close, we would like to ensure that our responses have adequately clarified and addressed your concerns.
>
> If you have any remaining questions or need further clarification, we would be more than happy to assist before the deadline. Please don't hesitate to reach out with any additional concerns.

---

### Meta-Review · Area_Chair_aDG5 · 2024-12-20

**Metareview:**

This paper received five positive ratings, with all reviewers generally inclined to accept it. The paper introduces FedLoG, a well-motivated and innovative approach to improving local generalization in federated graph learning by addressing overfitting. The methodology, particularly the creative design of feature scaling and a prompt generator, is novel and effectively enhances adaptability across clients. The experiments are comprehensive, yielding solid results across various real-world scenarios with strong baselines, which demonstrate the method's effectiveness. The paper is clearly written, well-organized, and professionally presented, with detailed experimental setups that ensure reproducibility. FedLoG's originality lies in using synthetic global data generated from reliable node types to tackle challenges such as missing classes and mutable graph structures. This work significantly advances federated learning, offering practical solutions for privacy-sensitive applications and providing valuable insights for future research. The authors' responses have addressed most of the reviewers' concerns, leading to an increase in their scores. Therefore, the Area Chair (AC) recommends accepting the paper.

**Additional Comments On Reviewer Discussion:**

The authors' responses have addressed most of the reviewers' concerns, leading to an increase in their scores. This paper received five positive ratings, with all reviewers generally inclined to accept it.

---

### Decision · Program_Chairs · 2025-01-22

Accept (Oral)